# Online robust non-stationary estimation

**Abishek Sankararaman**    **Balakrishnan (Murali) Narayanaswamy**

{abisanka, muralibn}@amazon.com,
Amazon Web Services, Santa Clara CA, USA.

## Abstract

The real-time estimation of time-varying parameters from high-dimensional, heavy-tailed and corrupted data-streams is a common sub-routine in systems ranging from those for network monitoring and anomaly detection to those for traffic scheduling in data-centers. For estimation tasks that can be cast as minimizing a strongly convex loss function, we prove that an appropriately tuned version of the `clipped Stochastic Gradient Descent` (SGD) is simultaneously *(i)* adaptive to drift, *(ii)* robust to heavy-tailed inliers and arbitrary corruptions, *(iii)* requires no distributional knowledge and *(iv)* can be implemented in an online streaming fashion. All prior estimation algorithms have only been proven to posses a subset of these practical desiderata. A observation we make is that, neither the $\mathcal{O}\left(\frac{1}{t}\right)$ learning rate for `clipped SGD` known to be optimal for strongly convex loss functions of a *stationary* data-stream, nor the $\mathcal{O}(1)$ learning rate known to be optimal for being adaptive to drift in a *noiseless* environment can be used. Instead, a learning rate of $T^{-\alpha}$ for $\alpha < 1$ where $T$ is the stream-length is needed to balance adaptivity to potential drift and to combat noise. We develop a new inductive argument and combine it with a martingale concentration result to derive high-probability under *any learning rate* on data-streams exhibiting *arbitrary distribution shift* - a proof strategy that may be of independent interest. Further, using the classical doubling-trick, we relax the knowledge of the stream length $T$. Ours is the first online estimation algorithm that is provably robust to heavy-tails, corruptions and distribution shift simultaneously. We complement our theoretical results empirically on synthetic and real data.

## 1 Introduction

Technology improvements have made it easier than ever to collect diverse telemetry at high resolution from any cyber or physical system, for both monitoring and control [31]. This in turn has led to a data deluge, where large amounts of data must be processed at scale [7, 16]. Given the scale and velocity of the data sources, offline processing to make predictions and decisions is computationally cumbersome. For real-time applications such as performance monitoring and anomaly detection, offline/batch processing results in stale predictions [6], [61], [54]. This necessitates computationally cheap, online algorithms that make predictions and decisions on *high-dimensional* streaming data [41], [21], [58]. Further in many applications, the challenge of being restricted to online algorithms is exacerbated by *heavy-tails* [2], [3], *distribution shift* [48], [28] and *outliers/anomalies* in the data-stream [29], [42], [35, 50]. In practice, although several heuristics to circumvent these issues have been designed [1, 66, 48, 28, 35, 42], there is no systematic study of the impact of these challenges on the achievable statistical accuracy. Motivated by these problems, we study algorithms for high-dimensional online statistical estimation and rigorously establish performance guarantees which exhibit graceful degradation with distribution shift and outliers in the data-stream.

37th Conference on Neural Information Processing Systems (NeurIPS 2023).

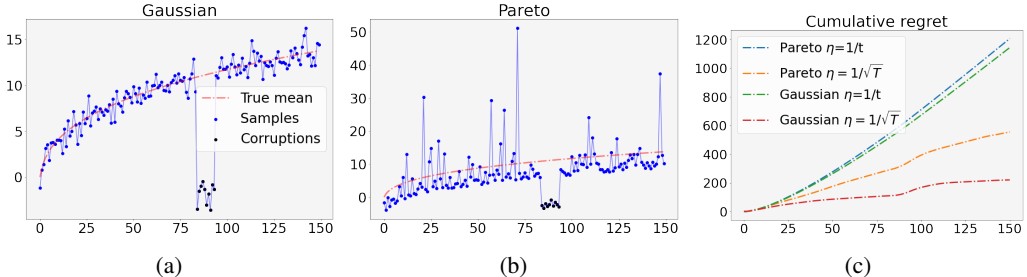

(a)  (b)  (c)

Figure 1: Figures $(a)$ and $(b)$ show a stream of independent samples (with some corruptions) from unit-variance Gaussian and Pareto distribution with shape parameter $2.001$ respectively when the underlying true mean changes with time. Figure $(c)$ shows that clipped-SGD with clip-value of $5$ incurs lower regret in estimating the true time-varying mean when the learning rate is set to $1/\sqrt{T}$ as opposed to the standard $1/(t+1)$ known to be optimal in a stationary environment.

## 1.1 Problem Setup

The online estimation problem we study is modeled as an interaction between an oblivious adversary and the estimator. The adversary before the interaction begins makes two choices - *(i)* a sequence of probability distributions $\mathbb{P}_1, \cdots, \mathbb{P}_T$ on $\mathbb{R}^d$ and *(ii)* a sequence of measurable *corruption functions* $(\mathcal{C}_t)_{t=1}^T$ where for every $t \in [T]$[1], $\mathcal{C}_t : \mathbb{R}^{d \times t} \times \mathbb{R}^{d \times t-1} \times \mathbb{R}^{p \times t-1} \to \mathbb{R}^d$, whose roles we explain below.

**Sequential interaction protocol** : At each time $t \in [T]$, the vector $X_t := Z_t + C_t$, where $Z_t \sim \mathbb{P}_t$ is sampled independently and $C_t := \mathcal{C}_t((Z_s)_{s \leq t}, (C_s)_{s < t}, (\theta_s)_{s < t})$ is the corruption vector computed using the corruption function based on the *past random samples and outputs*. $X_t$ is shown as input to the estimation algorithm. *Subsequently after observing $X_t$*, the estimation algorithm outputs an estimate $\theta_t \in \Theta$ and incurs loss $l_t := \|\theta_t - \theta_t^*\|^2$. Here, $\mathcal{L} : \mathbb{R}^d \times \mathbb{R}^p \to \mathbb{R}$ is a strongly convex loss function, $\Theta \subset \mathbb{R}^p$ a closed convex set denoting the parameter space, and $\theta_t^* := \arg\min_{\theta \in \Theta} \mathbb{E}_{Z \sim \mathbb{P}_t}[\mathcal{L}(Z, \theta)]$ is the unique[3] optimizer of the loss function with respect to the distribution $\mathbb{P}_t$, chosen by the adversary at time $t$. For every time $t$, we denote this minimizer $\theta_t^* \in \Theta$ as the *true parameter* at time $t$. The incurred loss $l_t$ is *un-observed* by the algorithm since the true parameter $\theta_t^*$ is unknown.

Formally, an estimation[4] algorithm $\mathfrak{A} := (\mathcal{A}_t)_{t=1}^T$ is a sequence of measurable functions such that for all $t \in [T]$, the estimate output $\theta_t := \mathcal{A}_t(X_t, \cdots, X_1)$, where $\mathcal{A}_t : \mathbb{R}^{d \times t} \to \mathbb{R}^p$ is a function of all inputs upto and including time $t$. After outputting an estimate $\theta_t$, the algorithm incurs an un-observed loss which is the L2 distance between the estimate and the true parameter.

**Causal property of the adversary:** Recall that at each time $t$, the corruption vector is given by $C_t = \mathcal{C}_t((Z_s)_{s \leq t}, (C_s)_{s < t}, (\theta_s)_{s < t})$, which is based only on the *past realizations* $Z_1, \cdots, Z_t$, $C_1, \cdots, C_{t-1}$ and the estimator's past outputs $\theta_1, \cdots, \theta_{t-1}$. The adversary does not have access to future randomness $Z_{t+1}, \cdots, Z_T$ while choosing the corruption vector $C_t$. This model of corruptions is *stronger* than those used in online robust estimation (cf. [19], [13]) since in our model, the adversary can choose the time instants of corruption while in previous models, the time instants of corruption are chosen randomly through i.i.d. coin flips. Our model of corruption locations are motivated by empirical observations that corruptions rarely occur randomly and are typically bunced [23, 42]. Thus we assume that the adversary has causal power to choose the location of corruptions rather than corruptions occuring at random instants.

**Regret as a Performance Measure:** The regret of the algorithm denoted by $\mathrm{Reg}_T$ is defined as

$$\mathrm{Reg}_T = \sum_{t=1}^T l_t = \sum_{t=1}^T \|\theta_t - \theta_t^*\|,$$

---

[1]We denote by $[T] := \{1, \cdots, T\}$

[2]Throughout, we denote by $\|\cdot\|$ as the L2 norm operator.

[3]Convexity implies existence and uniqueness of $\theta_t^* \in \Theta$

[4]In the literature, this is sometimes also denoted as an *optimization* algorithm. c.f. [8]

the total cumulative loss. This definition is known as *clean dynamic regret* [13], since *(i)* $\theta_t$ is compared against the true parameter $\theta_t^*$, *even if the input $X_t$ at time $t$ is corrupted*, i.e., even if $C_t \neq 0$, and *(ii)* $\theta_t$ is compared to the time-varying optimal $\theta_t^*$. Technically the regret of an algorithm $\mathfrak{A}$ is a random variable that is a function of the probability measures $(\mathbb{P}_t)_{t=1}^T$ and the corruption functions $(\mathcal{C}_t)_{t=1}^T$ and denoted as $\mathrm{Reg}_T^{(\mathfrak{A})}((\mathbb{P}_t)_{t=1}^T, (\mathcal{C}_t)_{t=1}^T)$. We will denote the regret as $\mathrm{Reg}_T$ whenever it is clear from context.

We now define the data-stream complexity parameters impacting regret : *(i)* distribution shift and (ii) corruptions.

**Distribution Shift[5]:** The cumulative distance between successive optimum points $\Phi_T := \sum_{t=2}^T \|\theta_t^* - \theta_{t-1}^*\|$ measures the complexity of distribution-shift on the data-stream. This is a natural measure since if the estimator outputs $\theta_t \in \Theta$ at time $t$, will incur an L2 loss $l_t = \|\theta_t - \theta_t^*\|$.

**Corruptions:** The corruption measure is the count $\Lambda_T := \sum_{t=1}^T \mathbf{1}(C_t \neq 0)$, of the number of times the random sample $Z_t$ is corrupted before inputting to the estimator. The count $\Lambda_T$ is a random-variable since the vector $C_t$ is a deterministic function of past random variables.

With this setup, the main desiderata we seek of an estimator is that it *(i)* is **online** : $O(d)$ run-time per data point on a stream and $O(d)$ overall memory, *(ii)* is **free of distributional knowledge**, i.e., does not require bounds on moments of $(\mathbb{P}_t)_{t\geq 1}$, distribution shift or corruptions, *(iv)* attains sub-linear in $T$ regret even when the distributions $(\mathbb{P}_t)_{t\geq 1}$ are **heavy-tailed**, and *(v)* **exhibits graceful degradation** of regret with respect to distribution shift and corruptions on the data stream.

### 1.2 Main Contributions

1. **Problem formulation and lower bounds**: We formalize the problem of online estimation under drift and corruptions and define a dimension-free target regret benchmark in Equation (1) that captures all the aforementioned desiderata. We give lower bounds in Propositions 16.1 and 2.6 that shows the necessity of assumptions such as finite diameter for the set $\Theta \subset \mathbb{R}^d$ in which the unknown parameters $\theta_t^* \in \Theta$ lie. Further discussion in Table 1 and in Section 11.

2. **High-probability upper bounds**: Theorem 5.1 proves that a tuned version of clipped-SGD achieves all aforementioned desiderata. No previous algorithm achieves all the desiderata simultaneously (Tables 1 and 2). Further in Theorem 4.3, we show that if in addition the data-stream is sub-gaussian, then our bound is optimal in the sense when instantiated with special cases of having no drift and corruptions [24] or in the setting of no corruptions and 0 variance [43], it recovers known optimal results.

3. **Algorithmic insights**: Our key finding is that neither the learning rate of $\mathcal{O}(1)$ known to be optimal for SGD to adapt to distribution shift in the absence of noise, nor the $\mathcal{O}(1/t)$ learning rate known to be optimal in a stationary environment can be used. Instead a learning rate of $T^{-\alpha}$ for $\alpha < 1$ is needed to combat noise and be adaptive to drift simultaneously. See also Figure 1. We show a lower bound in Proposition 6.1 that this insight is fundamental and not an artifact of our analysis. Knowledge of $T$ is relaxed using the doubling trick in Section 14 in the Appendix. Our result improves prior state of art in [8] that show that in the absence of corruptions, SGD *without clipping* can give sub-linear regret bounds *holding in expectation*. However, in the presence of heavy-tailed noise, high probability guarantees are more insightful compared to only in expectation [63]. Our algorithmic contribution is that a tuned version of *clipped* SGD can give high probability sub-linear regret guarantees, that hold even in the presence of non-stationarities and corruptions.

4. **Technical contributions in the proof techniques**: From a technical perspective, we employ novel proof techniques to derive high-probability under *any learning rate* on data-streams exhibiting *arbitrary distribution shift*. The prior state-of-art analysis in [36, 55] are limited to the specialized setting when the learning rate is $\eta_t = 1/(t+1)$ and the data does not exhibit drift. To be adaptive to drift requires newer techniques even in the case of sub-gaussian tails to achieve dimension-free bounds (cf. Section 4.1). We used an induction based technique to apply martingale concentrations in Lemma 19.9. We build upon this technique using contradiction arguments in Lemma 20.10 to extend to the heavy-tailed case. These arguments are of independent interest to prove bounds for other iterative algorithms.

---

[5]Following convention [22], we use the term *drift* to refer to distribution shift

| Setting | Best lower bound | Best upper bound |
|---------|------------------|------------------|
| Stationary, no-corruptions $(\Phi_T = 0, \Lambda_T = 0)$ | $\Omega[\sqrt{T}(\sqrt{\text{Trace}(\Sigma)} + \sqrt{\nu_{max}(\Sigma)\ln(1/\delta)})]$ [10] | $\mathcal{O}[\sqrt{T}(\sqrt{\text{Trace}(\Sigma)} + \sqrt{\nu_{max}(\Sigma)\ln(T/\delta)})]$ [38] |
| Stationary, corruptions $(\Phi_T = 0, \Lambda_T \geq 0)$ | Prop 2.6: $\Omega(\Lambda_T \mathcal{D})$ | Thm 5.1: $\widetilde{\mathcal{O}}(T^{\frac{5}{6}}\sigma^2 + T^{\frac{3}{4}}\mathcal{D}\Lambda_T)$ |
| Non-stationary, no-corruptions $(\Phi_T \geq 0, \Lambda_T = 0)$. | [8]: $\Omega(T^{2/3}\Phi_T^{1/3})$ | Thm 5.1: $\widetilde{\mathcal{O}}(T^{\frac{2}{3}}\Phi_T + T^{\frac{5}{6}}\sigma^2)$ [8] gives bound in expectation only. |
| Non-stationary, corruptions $(\Phi_T \geq 0, \Lambda_T \geq 0)$ | Prop 16.1: $\Omega(\Lambda_T \mathcal{D})$ | Thm 5.1: $\widetilde{\mathcal{O}}(T^{\frac{2}{3}}\Phi_T + T^{\frac{5}{6}}\sigma^2 + T^{\frac{3}{4}}\Lambda_T \mathcal{D})$ |

Table 1: Regret bounds for online estimation with heavy-tailed data and strongly convex loss. The setting of $\Phi_T = 0, \Lambda_T = 0$ (line 1) is characterized (upto log factors) in [10] and [38]. The first dimension free, high probability upper bounds are established in this paper for all other settings. For the setting of $(\Phi_T \geq 0, \Lambda_T = 0)$, [8] gives a lower bound and an upper bound holding in expectation, while ours is the first high-probability upper bound. Except line 1, there is a gap between known lower and upper bounds, closing which is an open problem (see also Section 9).

## 1.3 Motivating application

Online Anomaly Detection (AD) problems consist of a loss function $\mathcal{L}(\cdot, \cdot)$ and parameter $\theta_t$ that varies with time $t$, such that for the input $X_t$ received at time $t$, the algorithm outputs anomaly score $S_t := \mathcal{L}(X_t, \theta_t)$ ([47], [12]). The objective is to output lower scores for samples that are not corrupted, i.e., if at time $t$ the data distribution is $Z_t \sim \mathbb{P}_t$, then the optimal anomaly score $S_t^* := \mathcal{L}(X_t, \theta_t^*)$ is the one produced by the model minimizing the average anomaly score i.e., $\theta_t^* \in \arg\min_{\theta \in \Theta} \mathbb{E}_{Z \sim \mathbb{P}_t}[\mathcal{L}(Z, \theta)]$ [51]. The distance $\|\theta_t - \theta_t^*\|$ between $\theta_t$ the model used at time $t$ and the optimal $\theta_t^*$ is a measure of AD performance degradation [32, 50], motivating our study of low regret algorithms.

Special cases of such problems include **robust online mean estimation** [19] and **robust online linear regression** [20], both of which we consider in simulations in Section 7. The online mean estimation corresponds to $\mathcal{L}(X, \theta) := \frac{1}{2}\|X - \theta\|_2^2$ since $\arg\min_\theta \mathbb{E}_{Z \sim \mathbb{P}}[\|Z - \theta\|^2] = \mathbb{E}_{Z \sim \mathbb{P}}[Z]$. Online linear regression corresponds to splitting the input $X \in \mathbb{R}^{d_1+d_2}$ as $X := (X^{(1)}, X^{(2)})$, with $X^{(1)} \in \mathbb{R}^{d_1}$ and $X^{(2)} \in \mathbb{R}^{d_2}$ and using the *re-construction loss* $\mathcal{L}(X, \theta) := \frac{1}{2}\|X^{(2)} - \theta^T X^{(1)}\|_2^2$ as anomaly score. The linear regression setting of detecting anomalies by reconstructing one half of the input $X$ from the other is popularly known as self-supervised AD models [52]. Empirically, in [30], $\mathcal{L}(X, \theta) := \|(\mathbb{I}_d - \theta)X\|$ with $\theta \in \mathbb{R}^{d \times d}$ a matrix of rank $k < d$ and $\mathbb{I}_d \in \mathbb{R}^{d \times d}$ the identity matrix is used to detect anomalies. The linear model of [30] is extended to a non-linear setting in [42], using an auto-encoder [65]. In [17], a tree density model for $\mathcal{L}(X, \theta)$ is studied. These papers' focus is empirical and do not give statistical guarantees.

**Organization of the Paper:** In Section 2 we state the target benchmark regret along with the assumptions under which we seek it and appropriate lower bounds. In Section 3 we give the clipped-SGD algorithm. In Section 4, we give results under the additional assumption of sub-gaussian tails. This helps build intuition and proof techniques for the general result which we give in Section 5. We state implications of our results in Section 6, provide empirical evidence in Section 7, discuss related work in Section 8 and conclude with open problems in Section 9.

## 2 Formalizing the desiderata

Before formalizing the desiderata in Equation (1), we state some mild assumptions standard in the study of statistical estimation under which we seek guarantees [55, 19].

### 2.1 Model Assumptions

**Assumption 2.1** (Strong convexity)**.** There exists $0 < m \leq M < \infty$ *known* to the algorithm, such that for all $t$, the function $\theta \to \mathbb{E}_{Z \sim \mathbb{P}_t}[\mathcal{L}(Z, \theta)]$ is $M$ smooth and $m$ strongly convex.

This is a benign assumption that essentially states that properties of the loss function such as the convexity and smoothness are known to the estimation algorithm.

| Methods | Regret | Moment assumed | Drift tolerance | Corruption tolerance | Free of distributional knowledge |
|---|---|---|---|---|---|
| [55] | $\mathcal{O}(\sigma\sqrt{T})$ | 2 | 0 | 0 | ✗ |
| [36] | $\mathcal{O}(\sigma T^{1/p})$ | $p \in (1,2]$ | 0 | 0 | ✓ |
| [19] | $\mathcal{O}(\sqrt{dT} + (dT)^{1/4}\Lambda_T^{1/2})$ | 2 | 0 | $\mathcal{O}(T)$ | ✗ |
| Corollary. 4.8 (This paper) | $\mathcal{O}(T^{2/3}\Phi_T + T^{2/3}\sigma + T^{\frac13}\mathcal{D}^{\frac12}\Lambda_T)$ | sub-gaussian | $\mathcal{O}(T^{\frac13})$ | $\mathcal{O}(T^{\frac23})$ | ✗ |
| Theorem 4.3 (This paper) | $\mathcal{O}(T^{\alpha}\Phi_T + T^{\frac{2-\alpha}{2}}\sigma + T^{\frac{\alpha}{2}}\mathcal{D}^{\frac12}\Lambda_T)$ | sub-gaussian | $\mathcal{O}(T^{1-\alpha})$ | $\mathcal{O}(T^{\frac{2-\alpha}{2}})$ | ✗[6] |
| Corollary 5.3 (This paper) | $\mathcal{O}(T^{2/3}\Phi_T + T^{5/6}\sigma^2 + T^{\frac34}\mathcal{D}^{1/2}\Lambda_T)$ | 2 | $\mathcal{O}(T^{1/3})$ | $\mathcal{O}(T^{1/4})$ | ✓ |
| Theorem 5.1 (This paper) | $\mathcal{O}(T^{\frac{4\alpha}{3}}\Phi_T + T^{\frac{3-\alpha}{3}}\sigma^2 + T^{\frac{3\alpha}{2}}\mathcal{D}^{\frac12}\Lambda_T)$ | 2 | $\mathcal{O}(T^{\frac{3-4\alpha}{3}})$ | $\mathcal{O}(T^{\frac{2-3\alpha}{2}})$ | ✓ |

Table 2: Comparison of high-probability regret bounds of online estimation algorithms of strongly convex functions under various moment assumptions. [55] requires distance of the initial point to the unknown optimum. Theorem 4.3, Corollary 4.8, [19] and [55] requires a bound on $\sigma$. Further, [19] is not dimension-free as regret depends on $d$. We do not include [8] as their regret bounds only hold in expectation and not with high probability.

**Assumption 2.2** (Convex domain). The algorithm *knows* a closed convex set $\Theta \subset \mathbb{R}^p$ such that for all $t$, $\theta_t^* := \arg\min_{\theta\in\Theta} \mathbb{E}_{Z\sim\mathbb{P}_t}[\mathcal{L}(Z,\theta)] \in \Theta$ is the true parameter to be estimated at time $t$.

**Assumption 2.3** (Finite diameter). The diameter $\mathcal{D} := \max_{x,y\in\Theta}\|x-y\|$ of $\Theta$ is finite, i.e., $\mathcal{D} < \infty$.

This is a necessary assumption when $\Lambda_T > 0$ (Proposition 2.6). If $\Lambda_T = 0$, we relax this assumption in Corollaries 4.5 and 5.4. Let $\mathcal{R}_t(\theta) = \mathbb{E}_{Z\sim\mathbb{P}_t}[\mathcal{L}(Z,\theta)]$, denote the population risk at time $t$.

**Assumption 2.4** (Known finite upper bound on the true gradient). There exists a *known* finite $G < \infty$ such that $\sup_{\theta\in\Theta}\|\nabla\mathcal{R}_t(\theta)\| \leq G$.

This is necessary for many online optimization algorithms [36, 55, 25, 27].

**Assumption 2.5** (Existence of second moment). There exists a matrix $\Sigma \succeq 0$, *unknown* to the algorithm such that for all $t$ and $\theta \in \Theta$, the covariance of the random-vector $\nabla\mathcal{L}(Z,\theta)$ is bounded by $\Sigma$, i.e., $\mathbb{E}_{Z\sim\mathbb{P}_t}[(\nabla\mathcal{L}(Z,\theta) - \nabla\mathcal{R}_t(\theta))\cdot(\nabla\mathcal{L}(Z,\theta) - \nabla\mathcal{R}_t(\theta))^T] \preceq \Sigma$ holds for all $t$ and $\theta$.

The class of distributions admissible by this assumption is large including all sub-gaussian distributions and heavy-tailed distributions that do not have finite higher moments [25], [44]. Observe that no parameteric assumptions on the distribution family is made. Throughout, we denote by $\sigma^2$ and $\nu_{max}(\Sigma)$ as the trace and the largest eigen-value of $\Sigma$ respectively. For a positive semi-definite matrix $\Sigma \in \mathbb{R}^{p\times p}$, $\mathcal{P}(\Sigma)$ is the set of all probability measures satisfying assumptions 2.1, 2.4 and 2.5.

## 2.2 Target benchmark for algorithm design

We say that an estimator is *free of distributional knowledge* if it does not need as input bounds on either the moments of $\mathbb{P}_t$ nor on the stream complexity $\Phi_T$ and $\Lambda_T$. Our key focus - formalized below in Equation (1) - is: *can we design an online algorithm without distributional assumptions that has sub-linear regret and that degrades gracefully with drift and corruption?*

> Can we design an online algorithm $\mathfrak{A}$ without distributional knowledge, such that $\exists\, l, m, n < 1$ such that $\forall\, \Sigma \succeq 0 \in \mathbb{R}^{p\times p}$, $(\mathbb{P}_t)_{t=1}^T \in \mathcal{P}(\Sigma)$ and $(\mathcal{C}_t)_{t=1}^T$, the regret bound
>
> $$\mathrm{Reg}_T^{(\mathfrak{A})}((\mathbb{P}_t)_{t=1}^T, (\mathcal{C}_t)_{t=1}^T) \leq \mathcal{O}\left((T^l\Phi_T + T^m + T^n\mathcal{D}\Lambda_T).\mathrm{poly}\left(\sigma, \log\left(\frac{T}{\delta}\right), G, \frac{1}{m}\right)\right)$$
> (1)
>
> holds with probability at-least $1 - \delta$ ?

---

[6]$\alpha \in (0,1]$ is an input parameter.

Throughout, we use $\mathcal{O}(\cdot)$ to hide poly-logarithmic factors. Our main result is Theorem 5.1, where we prove that tuned `clipped SGD` achieves this, and our other desiderata.

### 2.3 Why is Equation (1) a good benchmark for regret bounds with drift and corruption?

The benchmark **demands robustness to heavy-tailed data,** since regret is sub-linear for any distribution with finite second moment. The benchmark **is dimension free**, since dimension terms $d, p$ do not appear in Equation (1). The finite diameter $\mathcal{D}$ only affects the bound due to corruptions and *not* the other terms. In particular, if $\Lambda_T = 0$ almost-surely, then the benchmark seeks a valid bound even if $\mathcal{D} = \infty$. In general when $\Lambda_T > 0$ however, dependence on $\mathcal{D}$ cannot be avoided as we show in Proposition 2.6 below (proof in Section 16).

**Proposition 2.6** (**Finite diameter is necessary even in the absence of drift**). *There exists a strongly convex loss function $\mathcal{L}$ such that for every $\mathcal{D} > 0$ and $d \in \mathbb{R}^d$, domain $\Theta = \{x \in \mathbb{R}^d : \|x\| \leq \mathcal{D}\}$ such that for every $T \in \mathbb{N}$ and every algorithm, there exists a probability measure $\mathbb{P}$ on $\mathbb{R}^d$ from which the un-corrupted samples are drawn from (i.e., $\Phi_T = 0$) and a sequence of corruption functions $(\mathcal{C}_t)_{t=1}^T$, such that the regret is at-least $\Omega(\Lambda_T \mathcal{D})$ with probability 1.*

Thus, even if there is no drift, finite diameter assumption cannot be avoided. Proof is given in Section 16, relies on the modeling asumption that the corruption locations can be arbitrary. This lower bound is in contrast to prior work [19] which shows that finite diameter is not necessary in the absence of corruptions and when the time instants of corruptions are random. Further in Table 1, we compare the best known lower and upper bounds for the various settings in the presence and absence of distribution shifts and corruptions. As can be seen, in all cases involving either distribution shift or corruptions, our work is the first to give high-probability estimation regret bounds.

**Equation (1) gives drift and corruption tolerance as additional performance measures:** Since the regret explicitly depends on $\Phi_T$ and $\Lambda_T$ we can define the *drift tolerance* and *corruption tolerance* as performance measures of the algorithm. Drift (Corruption) tolerance is the largest $\Phi_T$ (largest $\Lambda_T$) for which we still guarantee sub-linear in $T$ regret. Thus, if Equation (1) is satisfied by an algorithm, then the drift and corruption tolerance is $\mathcal{O}(T^{1-l})$ and $\mathcal{O}(T^{1-n})$ respectively. Thus, higher these tolerances, the better an algorithm is as they indicate that the algorithm's degradation with drifts and corruptions are more graceful. Our method is the only one to have non-zero drift tolerance (Table 2).

## 3 The clipped SGD Algorithm

In this section, we formally describe the algorithm in Algorithm 1 that achieves the desiderata. Algorithm 1 produces an estimate at time $t$ is given by $\theta_t \leftarrow \mathcal{P}_\Theta(\theta_{t-1} - \eta_t \mathrm{clip}(\nabla \mathcal{L}(X_t, \theta_{t-1}, \lambda))$, where $\mathcal{P}_\Theta$ is the projection operator on to $\Theta$ and $\mathrm{clip}(x, c) := \min(1, c/\|x\|)x, \ \forall x \in \mathbb{R}^d, c \geq 0$.

---

**Algorithm 1** `Clipped-SGD` [55]

---

1: **Input**: $(\eta_t)_{t \geq 1}, \lambda > 0, \theta_0 \in \Theta$ , $T$ time-horizon
2: **for** each time $t = 1, 2, \cdots$ , **do**
3:      Receive sample $X_t$
4:      Output $\theta_t \leftarrow \mathcal{P}_\Theta(\theta_{t-1} - \eta_t \mathrm{clip}(\nabla \mathcal{L}(X_t, \theta_{t-1}), \lambda))$
5: **end for**

---

**Intuition for why Algorithm 1 can achieve Equation (1)**: If there is no distribution shift or corruptions, clipped SGD with appropriate learning rate converges to the true parameter [55]. Now, if there are corruptions, clipping *limits* the impact on the overall regret. On the other hand, when a distribution shift occurs, the dynamics of Algorithm 1 is equivalent to restarting estimation under the new distribution, which will converge to the true parameter of this shifted distribution [55].

## 4 Regret bounds when the gradient vectors have sub-gaussian tails

Before giving the general result, we consider the special case of sub-gaussian distributions in this section. We do so because *(i)* the additional structure of sub-gaussian distributions enable us to prove stronger results in Corollaries 4.6 and 4.7 in the sequel, and *(ii)* the proof techniques from this special case enables us to build towards the proof of the general setting.

**Definition 4.1** (Sub-gaussian random vectors, [57]). A random vector $Y \in \mathbb{R}^d$ with co-variance matrix $\Sigma$ is said to be sub-gaussian, if for all $\lambda > 0$ and $u \in \mathbb{R}^d$ with $\|u\| = 1$, $\mathbb{E}[e^{\lambda \langle Y, u \rangle}] \leq e^{\frac{\lambda^2 \nu_{max}(\Sigma)}{2}}$, where $\nu_{max}(\Sigma)$ is the highest eigen-value of $\Sigma$.

**Assumption 4.2** (Sub-gaussian assumption with known upper bound). For every $t$, and $\theta \in \Theta$, the $0$ mean random vector $\nabla \mathcal{L}(Z, \theta) - \nabla \mathcal{R}_t(\theta)$, where $Z \sim \mathbb{P}_t$ is sub-gaussian with covariance matrix upper-bounded in the positive semi-definite sense by an *known* positive-semi-definite matrix $\Sigma$.

The following is the main result of this section.

---

**Theorem 4.3** (Informal version of Theorem 18.1). *Suppose Assumption 4.2 holds. For every $\delta \in (0, 1)$, if Algorithm 1 is run with constant step-size $\eta = \frac{1}{mT^\alpha}$ for $\alpha \in [0, 1]$ and clipping value $\lambda \geq G + \sigma + \mathcal{O}\left( \sqrt{\nu_{max}(\Sigma) \ln\left(\frac{T}{\delta}\right)} \right)$, then with probability at-least $1 - \delta$,*

$$
Reg_T \leq \mathcal{O}\Bigg( \underbrace{\Phi_T T^\alpha + \sqrt{\frac{T^{1+\alpha}\Phi_T \left( \sqrt{\nu_{max}(\Sigma) \ln\left(\frac{T}{\delta}\right)} + \sigma \right)}{m}}}_{\text{Regret from distribution shift}} +
$$

$$
\underbrace{\frac{T^{1-\frac{\alpha}{2}}}{m} \left( \sqrt{\nu_{max}(\Sigma) \ln\left(\frac{T}{\delta}\right)} + \sigma \right)}_{\text{Finite-sample estimation error}} + \underbrace{\Lambda_T T^{\frac{\alpha}{2}} \sqrt{m\lambda\mathcal{D}}}_{\text{Regret from corruptions}} \Bigg). \quad (2)
$$

---

The main achievement in Theorem 4.3 is in explicitly identifying how the choice of $\alpha$ trade-offs the regret contributions from distribution shift, finite sample error and corruptions.

*Remark* 4.4 (**Corruption and drift tolerance**). When Theorem 4.3 is instantiated with $\alpha \in (0, 1)$, Equation (2) yields a drift tolerance of $\mathcal{O}(T^{1-\alpha})$ and corruption tolerance of $\mathcal{O}(T^{1-\frac{\alpha}{2}})$. In particular, instantiating with $\alpha = 2/3$ yields $\mathcal{O}(T^{1/3})$ and $\mathcal{O}(T^{2/3})$ corruption and drift tolerance respectively.

We now read off several corollaries from this result.

**Corollary 4.5** (**Finite diameter assumption can be relaxed in the absence of corruptions**). *Let $\Lambda_T = 0$ almost-surely, i.e., there are no adversarial corruptions. Then, under the settings of Theorem 4.3 even when the set $\Theta$ is unbounded, the regret bound given in Equation (2) holds.*

**Corollary 4.6** (**Optimal in the stationary case:**). *Under the assumptions of Theorem 4.3, if $\Phi_T = \Lambda_T = 0$, then when Algorithm 1 is run with parameters $\lambda = +\infty$ and $\eta = 1/mT$, the regret bound $Reg_T \leq \mathcal{O}(\frac{\sqrt{T}}{m}(\sqrt{\nu_{max}(\Sigma) \ln(T/\delta)} + \sigma))$ holds with probability at-least $1 - \delta$.*

This corollary recovers the well known result [24] of convergence of SGD on strongly convex functions with sub-gaussian gradients.

**Corollary 4.7** (**Optimal in the noiseless setting**). *Under the assumptions of Theorem 4.3, if $\Lambda_T = 0$, and $\sigma = 0$, i.e., there is no noise, then running Algorithm 1 with $\lambda = +\infty$ and $\eta = 1/m$ obtains regret bound of $Reg_T = \mathcal{O}(\Phi_T)$.*

This result matches the lower bound in the noise-less setting [67] and recovers the upper-bound by previous analysis of online clipped-SGD specialized to the noiseless setting [43].

**Corollary 4.8** (**Special case of $\alpha = 2/3$**). *When Theorem 4.3 is instantiated with $\alpha = 2/3$, then w.p. at-least $1 - \delta$, a regret bound of $Reg_T \leq \mathcal{O}((T^{1/3}\Phi_T + \frac{T^{2/3}}{m} + T^{1/3}\Lambda_T)poly(\sigma, G, \ln(T/\delta)))$.*

*Remark* 4.9 (**The excess risk can be bounded using smoothness**). Oftentimes, the regret is also measured through the *excess risk*, i.e., $\sum_{t=1}^T \mathbb{E}_{Z \sim \mathbb{P}_t}[\mathcal{L}(Z, \theta_t) - \mathcal{L}(Z, \theta_t^*)]$. Since we assume that $\mathcal{L}$ is $M$ smooth (Assumption 2.1), Theorem 4.3 also bounds the excess risk regret by observing that $\mathbb{E}_{Z \sim \mathbb{P}_t}[\mathcal{L}(Z, \theta_t) - \mathcal{L}(Z, \theta_t^*)] \leq \frac{M}{2}\|\theta_t - \theta_t^*\|^2$.

## 4.1 Proof sketch for Theorem 4.3 and technical innovations

The full proof is Appendix 18. We first establish due to the condition on $\lambda$ that if an input sample is not corrupted, then the gradient will never be clipped. Then in Lemma 19.6, we expand the one-step

recursion of the clipped SGD updates by exploiting the strong-convexity. In order to account for the effect of corruptions, we expand the recursion differently if a time-step is corrupted or not. If a time-step is corrupted, then we accumulate both a *bias* term of norm at-most $\lambda$ and an appropriate *variance* term. If one were only interested in bounds *in expectation*, then the proof will follow the standard clipped SGD analysis as the expectation of the variance terms are 0.

To prove a high probability bound, a natural approach is to apply martingale concentrations to bound the variance terms as done in [24] *in the absence of drifts and corruptions*. A naive way to apply martingale concentrations would be to bound the variance error terms due to drifts and corruption by *the worst case error* by using the finite diameter $\mathcal{D}$. However, this *will not lead to the dimension free bound* of Theorem 4.3 —instead will lead to a bound in where the finite diameter $\mathcal{D}$ will also multiply the regret term due to finite sample error. In particular, this approach *will not* result in Corollary 4.5 of being able to relax the finite diameter assumption in the absence of corruptions.

We circumvent this issue by using an *inductive* argument to bound the variance terms in Lemma 19.9. Concretely, we prove the induction hypothesis that for each time $t$, error terms until time $t$ is bounded by an appropriate function of the drifts and corruptions. To establish the induction for a time $t$, we condition on the event the hypothesis holds till time $t-1$, and employ martingale concentration to show the error at time $t$ is small. Then we plug the martingale bound back into the one-step recursion and show the induction hypothesis also holds at time $t$. To get the final un-conditional result, we un-roll all the conditional events by an union bound.

## 5 Regret bounds in the general heavy-tailed case

The following is the main result of the paper, where we relax Assumption 4.2 and thus making the algorithm free of distributional knowledge and allowing for potentially heavy-tailed data.

---

**Theorem 5.1** (Informal version of Theorem 20.1). *When Algorithm 1 is run with clip value* $\lambda = 2GT^{\frac{\alpha}{3}}$, *and step-size* $\eta = \frac{1}{mT^\alpha}$ *for* $\alpha \in [0,1]$, *then for every* $\delta \in (0,1)$,

$$
Reg_T \leq \mathcal{O}\left( \underbrace{\frac{G\sigma T^{\frac{4\alpha}{3}}\Phi_T}{m^{3/2}} + \frac{\sigma T^{\frac{1}{2}+\frac{5\alpha}{6}}\sqrt{\Phi_T}}{m^{3/2}\sqrt{G}}}_{\text{Regret due to distribution shift}} + \underbrace{\frac{T^{1-\frac{\alpha}{3}}(G\sigma)^2 \ln\left(\frac{T}{\delta}\right)}{m}}_{\text{Finite-sample estimation error}} + \underbrace{\frac{T^{\frac{3\alpha}{2}}\Lambda_T G^2\sigma\sqrt{\mathcal{D}}}{m}}_{\text{Regret from corruptions}} \right),
$$
(3)

*holds with probability at-least* $1-\delta$.

---

The achievement in Theorem 5.1 is in explicitly identifying how the choice of $\alpha$ trade-offs the regret contributions from distribution shift, finite sample error and corruptions, despite heavy-tailed data.

*Remark* 5.2 (**Price for relaxing assumption 4.2 is sub-optimal bound in the stationary case:**). The setting in Theorem 5.1 is weaker compared to that of Theorem 4.3 since *(i)* no sub-gaussian tail assumptions are made, and *(ii)* no knowledge of the upper-bound matrix $\Sigma$ is assumed. The price for relaxing these assumptions is a weaker regret, specifically in the term due to finite-sample estimation error. This term scales as $\mathcal{O}(T^{1-\frac{\alpha}{3}})$ in Theorem 5.1 while it scales as $\mathcal{O}(T^{1-\frac{\alpha}{2}})$ in Theorem 4.3. In particular in the stationary case, when Equation (3) is instantiated with $\alpha = 1$, the regret bound reads as $Reg_T = \mathcal{O}(T^{2/3}\sigma^2)$ which is weaker than the optimal $\mathcal{O}(\sqrt{T}\sigma)$ for the stationary case [55, 39].

**Corollary 5.3** (**Setting** $\alpha = 1/2$). *Under the conditions of Theorem 5.1, if Algorithm 1 is run with* $\alpha = 1/2$, *then with probability at-least* $1-\delta$, *it satisfies a regret bound of* $Reg_T \leq \mathcal{O}((T^{2/3}\Phi_T + T^{11/12}\sqrt{\Phi_T} + T^{5/6} + T^{3/4}\Lambda_T)poly(\sigma, G, 1/m, \ln(T/\delta)))$.

**Corollary 5.4** (**Finite diameter assumption can be relaxed in the absence of corruptions**). *Let* $\Lambda_T = 0$ *almost-surely, i.e., there are no adversarial corruptions. Then, under the settings of Theorem 5.1 even when the set* $\Theta$ *is unbounded, the regret bound given in Equation (3) holds.*

*Remark* 5.5 (**Corruption and drift tolerance**). Theorem 5.1 when instantiated with $\alpha \in (0,1)$ yields a drift tolerance of $\mathcal{O}(T^{1-\frac{4\alpha}{3}})$ and corruption tolerance of $\mathcal{O}(T^{1-\frac{3\alpha}{2}})$. In particular, instantiating with $\alpha = 1/2$ yields $\mathcal{O}(T^{1/3})$ and $\mathcal{O}(T^{1/4})$ corruption and drift tolerance respectively.

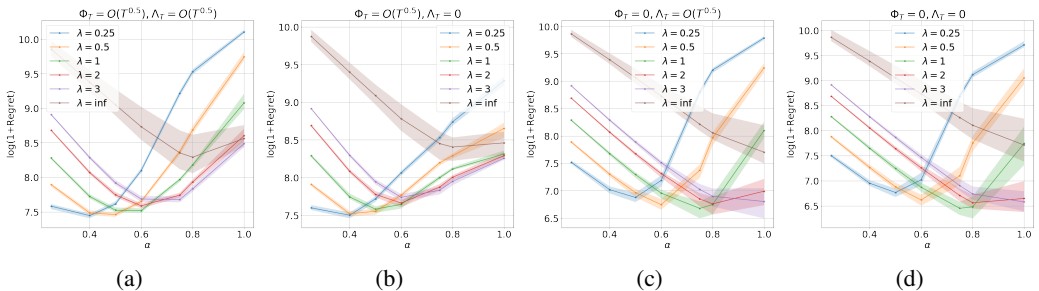

Figure 2: Plot of regret versus $\alpha$ over 10000 steps for mean-estimation. We observe that the best choice of $\alpha$ decreases as $\Phi_T$ and/or $\Lambda_T$ increases. More details in Section 7.

## 5.1 Proof sketch of Theorem 5.1 and technical innovations

The proof for this case builds upon the techniques used to prove Theorem 4.3. Unlike in the sub-gaussian case, we cannot guarantee that only corrupted samples will be clipped in the general case. This introduces an additional bias and variance terms due to clipping of potential un-corrupted samples or *inliers*. The bias terms of the inliers are handled using techniques from prior works [25, 55]. We bound the variance terms using an inductive argument similar in spirit to that of Theorem 4.3. However, identifying the *correct* hypothesis involving drifts and outliers so that the induction argument holds is challenging. The challenge is because the induction hypothesis assumed till time $t-1$ impacts the martingale error term, which in-turn impacts whether the induction hypothesis will hold at time $t$. We introduce a free parameter to the regret term and deduce the induction hypothesis to hold if a quadratic equation in this parameter does not have any real roots. This extra term contributes to regret degradation compared to the sub-gaussian case (Lemma 20.10).

# 6 Insights and remarks

**Known Time Horizon:** This is a benign assumption and can be overcome by the standard doubling trick (Section 2.3 of [11]) with additional $\log^2(T)$ factor regret (c.f. Appendix 14).

**Price for being adaptive to distribution shift:** Theorems 4.3 and 5.1 show that, in order to minimize regret due to statistical estimation, we need to set $\alpha = 1$, i.e., choose a learning rate of $O(1/T)$. This was shown to be optimal in in the absence of drift and corruptions both under sub-gaussian assumptions [24] and in heavy tail settings [55]. On the other hand, we see from Theorems 4.3 and 5.1, that in the presence of drift $\Phi_T > 0$, a learning rate $\alpha < 1$ is sufficient to ensure sub-linear regret. The following lower bound (proved in Appendix 15) shows that $\alpha < 1$ is also necessary.

**Proposition 6.1 (Lower bound showing $O(1/T)$ learning rate is not adaptive to drifts).** *There exists a loss function $\mathcal{L}$ such that for every $T$, there exists a sequence of probability measures $(\mathbb{P}_t)_{t\geq}$ with the diameter $\mathcal{D} \leq 2\log(T)$, distribution shift $\Phi_T \leq 2\log(T)$, $\Lambda_T = 0$, such that Algorithm 1 when run with $\lambda \geq 1$ and step size $\eta_t := \frac{1}{t+1}$ will incur regret at-least $\frac{T}{3}$ with probability 1.*

**Finite diameter $\mathcal{D}$ is necessary in general:** We already saw in Proposition 2.6 that $\Omega(\Lambda_T\mathcal{D})$ regret is necessary. We prove the lower-bound by considering mean-estimation in Section 16. Variants of Proposition 2.6 was observed in the literature (for ex. Proposition 1.3 [13], Lemma 6.1 in [33], Theorem $D.1$ in [14], Line 8 of Algorithm 3 in [19]). In theoretical statistics literature, $\Omega(\Lambda_T\mathcal{D})$ is considered *un-desirable* [13] and thus the models studied restrict corruptions to occur at *random times* [13, 19] rather than arbitrary as in our setup. However, in practice corruptions are rarely random and typically occur close in time for instance due to machine failures or other external confounding factors [23, 42].

**Regret bounds are dimension-free:** The problem dimensions $d, p$ do not appear in the regret bound. Moreover, the finite diameter $\mathcal{D}$ only appears in the regret term affecting through adversarial corruptions and *not* in the distribution shift and finite sample error terms. Further, Corollaries 4.5 and 5.4 give regret bounds that hold even when $\mathcal{D} = \infty$ in the absence of corruptions.

# 7 Experiments

Theorems 4.3 and 5.1 give *upper bounds* on regret showing that tuning the learning rate depending on the amount of distribution shift and number of corruptions in the stream is beneficial. We empirically observe similar phenomena in Figures 2 and 3 (in the Appendix), thus indicating our theoretical observations are fundamental and not artifact of our bounds. We consider $\mathcal{L}(X, \theta) = \frac{1}{2}\|X - \theta\|^2$ corresponding to mean-estimation in Figure 2 and linear-regression of $\mathcal{L}(X, \theta) = \frac{1}{2}\|X^{(2)} - \theta^T X^{(1)}\|^2$, where $X = (X^{(1)}, X^{(2)})$ with $X^{(1)} \in \mathbb{R}^{d-1}$, $X^{(2)} \in \mathbb{R}$, $\theta \in \mathbb{R}^{(d-1)}$ in Figure 3 in the Appendix in Section 12. We compare clipped-SGD with learning rate $1/mT^\alpha$ for variety of $\alpha$ with clipping $\lambda = 2T^{0.25}$ and use the doubling trick to remove dependence of $T$ (i.e., use Algorithm 2). For the case of $\alpha = 1$, we use the learning rate of $\eta_t = 1/(m(t+1))$ and $\lambda = 2\sqrt{T}$ as suggested in [55]. All plots are averaged over 30 runs and we plot the median along with the 5th and 95th quantile bands. We observe in Figures 2 and 3 that as $\Phi_T$ or $\Lambda_T$ increases, the optimal $\alpha$ to use decreases, validating the theoretical insight. Further details in the Appendix in Section 12. Evaluations on real-data is conducted in the Appendix in Section 13.

# 8 Related Work

FITNESS for mean estimation is proposed in [50] that requires variance as input and is adaptive to drifts and corruptions under sub-gaussian distributions with per-sample computational complexity is $O(dT)$. In contrast, our algorithm applies to any strongly convex function, does not require moment bounds, data can be heavy-tailed. The work of [8] studied regret bounds in the absence of corruptions and only give bounds in expectation. A sequence of papers in online estimation including [44, 46, 13, 25, 55, 50] have derived algorithms that are robust in different ways. However, *none of them consider the impact of distribution shift*. The works of [46, 13] study robust *linear* regression in the absence of drifts with [46] making Gaussian assumptions on both covariates and response while [13] makes Gaussian assumption only on the responses. The work of [55] show that clipped SGD attains near optimal rates for any strongly convex loss function in the *absence of drifts and corruptions*. The paper of [19] study online estimation of strongly convex loss functions with corruptions, but do not consider drift. Moreover, their regret bounds are not dimension free (cf. Table 2). Although the paper of [64] gives regret bounds for online learning, a more general setting compared to estimation, do not consider impact of drifts. High probability bounds for optimization with heavy-tailed data have been studied by [44, 25, 15, 60, 36] in recent times. However, *none of their analysis considers the impact of drift and corruptions*. More related work in Section 10.

# 9 Conclusions and Open Problems

We studied online estimation in the presence of drifts and corruptions and potentially heavy-tailed inlier data and proved regret bounds for clipped SGD for strongly convex loss functions. Ours is the first proof that an online algorithm can simultaneously be robust to heavy-tails and corruptions and adaptive to drift. A key result was to show how the choice of learning-rate trades off drift, finite sample error and corruptions. Our work leaves several interesting open problems.

- The optimal choice of $\alpha$ in Theorems 4.3 and 4.3 is a trade-off between distribution shift, finite sample error and corruptions. Can the optimal $\alpha$ be set without knowing $\Phi_T$, $\Lambda_T$ or $\sigma$ in advance?

- In the absence of corruptions, can regret $\mathcal{O}((\sqrt{T} + T^a \Phi_T^{1-a})\mathrm{poly}(G, \sigma, \log(T/\delta))$, for some $a < 1$ be achieved? Such an algorithm would simultaneously have both *(i)* $\mathcal{O}(T)$ distribution shift tolerance which is the best possible and matching the lower bound established in [50], and *(ii)* closing the gap of Remark 5.2 for the stationary case.

- Can $m = 1/2$ in Equation (1) be achieved in the general case with drift and corruptions? Theorem 5.1 shows that only $m \geq 2/3$ is achievable. For the special case of stationary stream, [55] shows $m = 1/2$ is achievable, matching the lower bound that $m \geq 1/2$ is necessary.

- (Open problem from [55]) Does there exists an online algorithm that can obtain the statistically optimal regret of $\mathcal{O}(\sqrt{T}(\sqrt{\mathrm{Tr}(\Sigma)} + \sqrt{\nu_{max}(\Sigma)\log(T/\delta)}))$ for a stationary stream?

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

# 10  More related work

**Comparison with [8]:**The seminal work of [8] started the work on non-stationary estimation in the absence of corruptions. They establish a regret lower bound of $\Omega(\Phi^{1/3}T^{2/3})$ for non-stationary estimation in the presence of non-stationarities. They also established an upper bound by showing that SGD *without clipping* yields matching regret upper bounds *in expectation*. The upper bound in Theorem 5.1 on the other hand gives high-probability regret bounds that hold even in the presence of corruptions, in addition to non-stationarities. Expressing an upper bound in high-probability is both technically challenging and algorithmically insightful. The insight we make is that for having high-probability bounds, we need to have *clipped SGD*. On the other hand, since [8] only give bounds holding in expectation, they do not need any clipping in their algorithms. The necessity of clipping in heavy-tailed settings is not an artifact of analysis, but is crucial for good empirical performance, as noted in recent works of [55]. Thus a conceptual contribution we make is that even in the absence of corruptions, a different algorithm compared to [8], namely that of clipping gradients is required to get regret bounds holding in high-probability. From a technical perspective, the proofs for high-probability bounds need different techniques as compared to [8]. For instance, we need several martingale and induction arguments to arrive at high-probability bounds in heavy tails while [8] have a much more simpler proofs just based on convexity, since they only give bounds in expectation.

**Robust Stochastic Optimization** The paper of [44] proves high-probability bounds for optimizing convex functions under heavy tailed gradients. However, they do not consider corruptions or drifts and assume finite diameter. In contrast our bounds explicitly depend on drifts and corruptions and can handle infinite diameter in the absence of corruptions. The work of [25] give high-probability bounds for *offline* clipped gradient descent based algorithms for optimizing both convex and strongly convex functions. However, they do not consider drift or corruptions. The work of [60] studies convergence of gradient descent methods in the absence of second moments, but only give bounds in expectation and do not consider drift or corruptions. The paper of [15] give high-probability bounds for stochastic optimization, but their analysis does not consider drift or corruptions. *None of these papers simultaneously consider drifts, corruptions and heavy-tailed noise*. The paper of [63] extend this to heavy-tailed noise under only the $p \in (1, 2]$th moment and non-convex functions. However, they only give results in expectation and *in a setting without drift and corruptions*.

**Robust Statistics** There is a growing literature [18, 37, 39, 10] that give near-optimal *offline* algorithms in the presence of corruptions but no drifts. Since our focus is to devise *online* algorithms that are adaptive to distribution shift, these algorithms are not directly applicable to our setting.

**Dynamic Regret Minimization** A parallel line of work concerns algorithms being adaptive to drifts for online convex optimization problems [27] where the data are arbitrary rather than sampled from a distribution [67, 65, 56, 5]. However, these algorithms assume that there are no noise in the gradients and no corruptions on the data stream.

**Gradient clipping** The work of [45] showed clipping gradients as a practical work-around to the 'exploding gradients' problem. Theoretically, [62] study the effect of clipping in offline training under restrictive noise assumptions and do not consider impact of drifts and corruptions.

# 11  Key algorithmic challenge

**All distribution changes initially look like corruptions**

In this section, we give a simple example to see why developing algorithms that are robust to drift and corruptions are challenging. Consider a simple example of estimating the mean in the absence of noise, i.e., when variance is $0$. We will show two possible underlying scenarios that yield the same observation to the forecaster. In both scenarios, the observed samples are all $0$ in the first $T - \Lambda_T$ rounds and is $R \neq 0$ in all of the last $\Lambda_T$ rounds. The first scenario is one where the true mean in the first $T - \Lambda_T$ samples is $0$ and the true mean in the last $\Lambda_T$ samples is $R$, i.e., this scenario has no adversarial corruptions and a cumulative distribution shift of $R$. In the second scenario, the true mean for all rounds is $0$, except the last $\Lambda_T$ samples are adversarially corrupted, i.e., this scenario has no distribution shift but has $\Lambda_T$ adversarially corrupted samples. If the true system is scenario 1, then the regret of a forecaster is the distance from the ground-truth $R$, while in the second scenario is the distance from the *un-corrupted* mean $0$. Thus, any algorithm will incur in at-least one of the scenarios, a regret of at-least $\frac{R\Lambda_T}{2}$. This example highlights the main tension in the problem : a

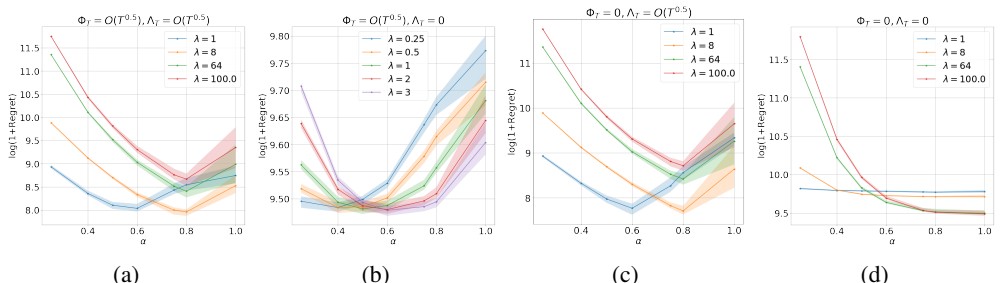

Figure 3: Over $T = 10000$ steps, we vary $\Phi_T$ and $\Lambda_T$ keeping the clipping value $\lambda = 2T^{0.25}$ for linear regression. We observe that the best choice of $\alpha$ decreases as the distribution shift $\Phi_T$ and/or corruptions $\Lambda_T$ increases.

distribution shift initially presents as outliers and the online algorithm needs samples and thus incurs regret in order to learn this to be the 'new normal'.

## 12  Additional details on simulations

We give more details on experiments reported in Figures 2 and 3 for synthetic data.

**Mean-Estimation Setup** At each time $t$, we sample a 0 mean univariate random variable $Y_t$ from the Pareto distribution with mean 0, variance $\sigma^2 := 5$ and tail parameter 3, so that it has a finite second moment. The un-corrupted $d$-dimensional vector at time $t$ is denoted by $Z_t := \frac{(Y_t - \mu_t)}{\sqrt{d}} \mathbf{v}_t$, where $\mathbf{v}_t$ is a unit vector sampled uniformly on the sphere of $d = 256$ dimensions. The true means $\mu_t$ are chosen by running a random walk, i.e., $\mu_0 = 0$ and $\mu_t := \mu_{t-1} + w_t$, where $w_t \sim \mathcal{N}\left(0, \frac{1}{T^{1-\kappa}} \mathbb{I}_d\right)$. This ensures that $\Phi_T \approx T^\kappa$. In Figures 2a, 2b, 2c, 2d , we set $\kappa \in \{0.5, 0.25, 0.15, 0\}$ respectively. The instants of corruption are chosen at integer multiples of $\lfloor \sqrt{T} \rfloor$ and the observed sample $X_t = 0$, if $t = k\lfloor \sqrt{T} \rfloor$, for some non-negative integer $k$. Otherwise, $X_t = Z_t$. For a given underlying mean vector, we run the streaming setup 30 times, and plot the average along with $95\%$ confidence interval on the regret estimates.

**Linear Regression Setup:** We generate the covariate $X_t \in \mathbb{R}^{256}$ from a generalized Pareto distribution with mean 0, variance $\sigma^2 := 1$, i.e. $m = 1$. The true parameter at time 0 was set to $\theta_0 := [d^{-0.5}, \cdots, d^{-0.5}]$. At each time $t$, the true parameter $\theta_t^* := \mu_{t-1} + w_t$ where $w_t \sim \mathcal{N}\left(0, \frac{1}{T^{1-\kappa}} \mathbb{I}_d\right)$. This ensures that $\Phi_T \approx T^\kappa$. In Figures 3a, 3b, 3c, 3d , we set $\kappa \in \{0.5, 0.25, 0.15, 0\}$ respectively. At each time $t$, the response $y_t = \langle X_t, \theta_t^* \rangle + n_t$, where $n_t$ is a Pareto distribution with 0 mean, variance $\sigma^2 = 3$ and shape parameter 2.5. The instants of corruption are chosen at integer multiples of $\lfloor \sqrt{T} \rfloor$. At instants of corruption, the covariates are sampled from a normal distribution with identity covariance and mean 100. The response variable for a corrupted time point is set to 0. For a given set of underlying mean vector, we run the streaming setup 30 times, and plot the average along with $95\%$ confidence interval on the regret estimates.

**Methods:** We compare clipped-SGD with learning rates $\eta_t := 1/mT^\alpha$ and clipping values $\lambda = 2T^\beta$ for a variety of $\alpha$ and $\beta$. Further we implement the doubling trick given in Algorithm 2 so that clipped-SGD does not require knowledge of the time-horizon $T$. In addition, we also compare against the clipped-SGD version of [55] which uses the time-varying learning rate of $1/(t)$ learning and clipping value of $2\sqrt{T}$. Note that the algorithm of [55] has strictly more information than the one we implement as it has knowledge of time horizon $T$ for setting the clipping value, while the one we propose does not and relies on the doubling trick. Further, we do not assume that the set $\Theta$ is known and consider the un-projected clipped-SGD for our algorithm, while for that of [55], we consider the smallest $L2$ ball around the horizon that contains all the $(\theta_t^*)_{t=1}^T$. Thus, the algorithm we implement is truly parameter-free as it does not have any problem specific parameters in its implementation.

**Observations:** All plots in Figure 2 show both mean-estimation and linear regression for varying amounts of distribution shift $\Phi_T$, while keeping the number of corruptions $\Lambda_T = \sqrt{T}$ and clipping value $\lambda = 2T^{0.25}$. For both mean-estimation in Figures 2a, 2b, 2c, 2d and linear regression in Figures

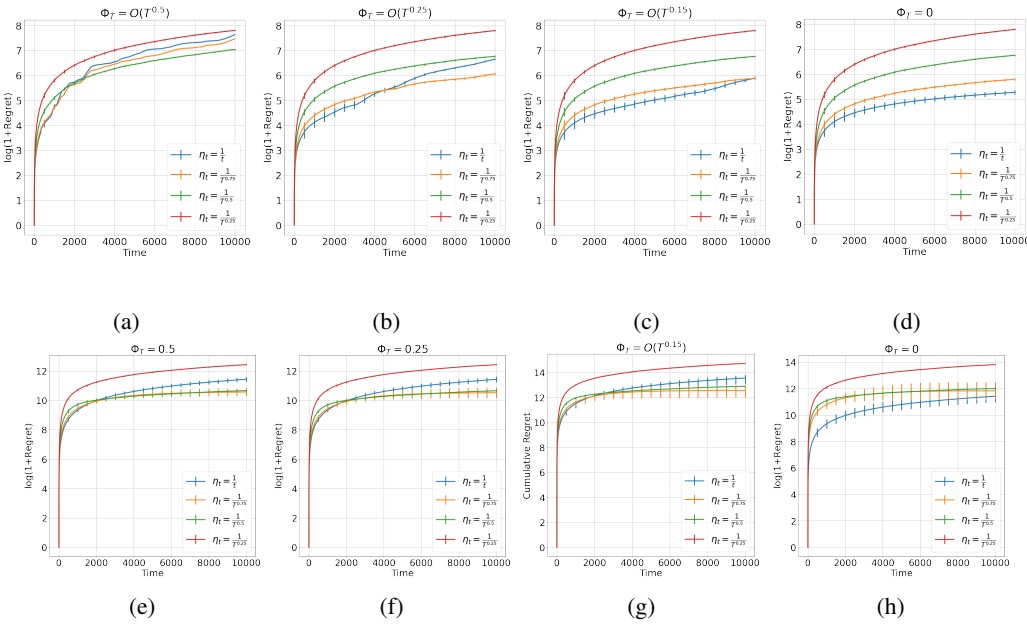

Figure 4: Over $T = 10000$ steps, we vary $\Phi_T$ while keeping the number of corruptions $\Lambda_T = \sqrt{T}$ and the clipping value $\lambda = 2T^{0.25}$ the same. The top rows $(a) - (d)$ are for the synthetic mean-estimation and the bottom figures $(e) - (h)$ are for linear regression.

3a, 3b, 3c, 3d, we show that as the amount of distribution shift $\Phi_T$ increases, the learning rate needs to be more aggressive to be adaptive to the distribution shift.

**Sample paths** In Figure 4, we plot a particular sample path over time for a variety of settings of $\alpha$.

## 13 Real-data experiments

The key result of this section is that even on real-data with real performance metrics, adapting the learning-rate to the amount of distribution shift on a data-stream is beneficial. We consider two tasks to demonstrate this : online binary classification and online anomaly detection.

| Dataset | Stream-length $T$ | Task | Dimension $d$ |
|---|---|---|---|
| ElectricityNSW [26, 4] | 45312 | Binary classification | 13 |
| MiniBoone [49, 4] | 130065 | Binary Classification | 50 |
| MNIST [34] | 11811 | Anomaly Detection | 784 |

Table 3: Real data-sets used.

### 13.1 Classification setting

**Datasets:** We consider two binary classification datasets available from the UCI repository [4] *(i)* NSW Electricity data [26] and *(ii)* MiniBoone data [49]. The NSW Electricity dataset consists of 45312 samples with each data-point having 6 numeric features and one categorical feature representing the day of week. After one-hot-encoding the categorical feature, we have 13 numeric features and a binary target. The MiniBoone dataset has 130065 samples with each data point having 50 numeric features and a binary target. We choose these two as they have been shown to be good binary classification benchmark for data-streams with drift [40]. In particular, the MiniBoone dataset is *extreme* where the first 36499 has target of 1 while all the last 93565 data points have target of 0. For both the datasets, the order of streaming is the *same order* in which they are collected. This is consistent with the evaluation protocol of streaming algorithms in the presence of drifts [40].

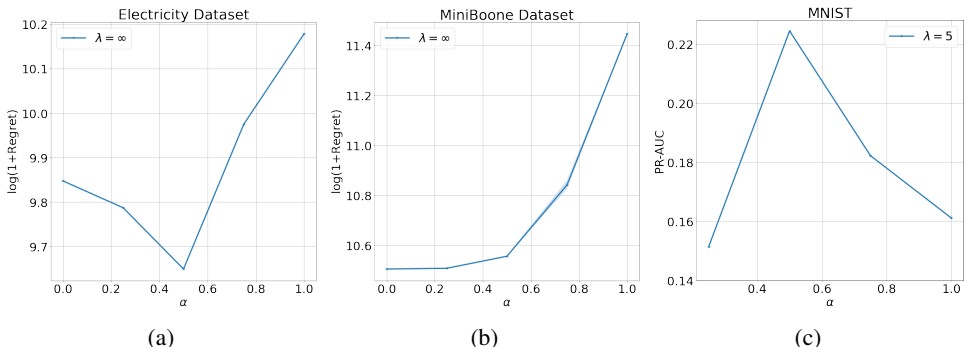

(a)  (b)  (c)

Figure 5: The plot of Regret with different choice of learning rate. In Figures $(a)$ and $(b)$, we plot the log(Regret), where lower is better, while in $(c)$ we plot the Precision-Recall area under curve, where higher is better. We see for the Electricity and MNIST dataset that $\alpha \approx 0.5$ has the best performance, while for the MiniBoone dataset in $(b)$, we see that $\alpha = 0$ has the best performance. The result for MiniBoone is not surprising as that dataset is has very little noise ($\sigma \approx 0$) compared to the Electricity dataset. Thus, the optimal choice of $\alpha$ is lower in MiniBoone compared to the Electricity data. Furthermore, across datasets, tasks and models, our theoretical insights hold demonstrating that they are not artifacts of analysis.

**Model and Methods:** For both data-sets, we consider a simple one layer Logistic regression as the model to train. We set the gradient clipping value $\lambda = 1$ for both datasets. For both these datasets, we deviate from our setup and employ the classical online learning evaluation [40]. At each time $t$, the covariate is first shown to the algorithm which then predicts the target. The prediction of the logistic regression is then threshold at $0.5$. The instantaneous loss at time $t$ is the indicator loss whether if predictor after applying the threshold matches the target. The covariate and label pair is then used to take one step of clipped SGD in order to get the model used at the next time step. We experiment with different learning rates and plot their impact in Figure 5.

**Observations:** We observe in Figure 5 that for the Electricity dataset which has more shifts and is noisy $\sigma > 0$, the optimal choice of $\alpha \approx 0.5$, while for the MiniBoone dataset has just one change point (i.e., $\Phi_T \approx 0$) and is noiseless $\sigma \approx 0$ (cf [40]). Thus the optimal $\alpha \approx 0$.

### 13.2  Online Anomaly Detection

**Dataset** : We considered the standard MNIST dataset [34]. We modified it to a stream containing drift and corruptions/anomalies as follows. We introduce abrupt change points by first streaming in all $0$'s followed by $1$'s and so on with the last of the stream being $9$'s. This way, the stream has $9$ *abrupt* change-points. In addition, within a change-point, the images are slowly rotating clockwise at a fixed angle of $1$ degree rotation to model *slowly varying* change in between two abrupt changes. Anomalies are introduced at random with probability $0.05$ by sampling a digit other than the *current* segment of time. For instance 5c, the third point on the stream is an anomaly since the digit is different from $0$ which is the inlier digit till the first break-point. Similarly, the last but one the data point on the stream is an anomaly since it is different from $9$, the inlier digit in that segment.

**Methods**: We flatten the image into a vector of $784$ and consider a simple $2$ layer auto-encoder with hidden dimension $512$ and activation function of ReLU [66]. We initialize the network to be random at the beginning of the stream and consider clipped SGD with clip-value set to $5$ for various learning rates as shown in Figure 5c. At each time $t$, the sample $X_t$ is revealed to the algorithm, a single clipped gradient step is taken on this sample to update the model and then the anomaly score is produced from the resulting model.

**Observation**: In Figure 5c, we plot the anomaly detection accuracy measured through the average precision recall score, where higher the score indicates better AD performance. We see that neither too small nor too high a value of $\alpha$ obtains the highest performance.

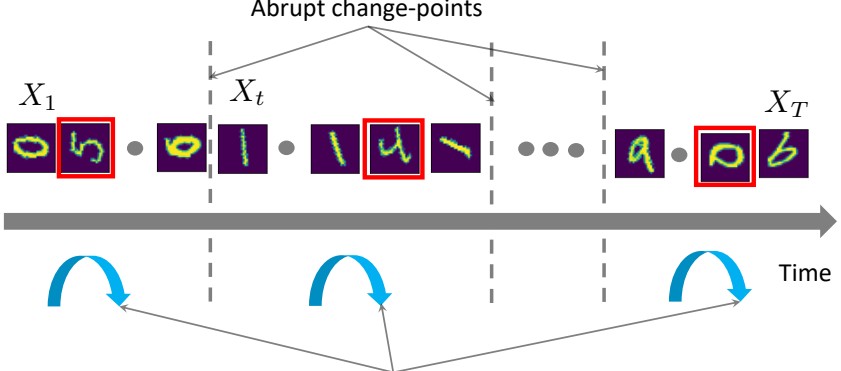

Figure 6: The data stream of MNIST digits with drifts and corruptions. The stream is ordered such that all digits labeled 0 arrive before digits labeled 1 and so-on. The abrupt change points are time instants when the next digit starts appearing. Within an abrupt change-point corresponding to a digit $i \in \{0, \cdots, 9\}$ and time-step $t$, with probability 0.95, a digit $i$ image is sampled from the MNIST dataset without replacement, rotated by angle $q_t$. With probability 0.05, an image corresponding to a digit other than $i$ is sampled, which is labeled (but unknown to the algorithm) as a true anomaly. These images are marked with the red-boundary in the figure. The rotation angle varies with time as $q_{t+1} := q_t + \frac{1}{\sqrt{t}}$, i.e., the rotations are slowly varying. The goal of the algorithm is to detect the true anomalies. In this experiment (see also Table 3), $T = 1181$ and the number of anomalies was 563.

## 14 Removing the knowledge of time horizon $T$ from Algorithm 1

In this section, we re-state the *doubling trick* from [11] to remove the dependence of the stream length $T$ on the learning rate tuning in Theorems 4.3 and 5.1.

---
**Algorithm 2** `Online-Clipped-SGD` without time horizon
---
1: **Input**: Learning rate exponent $\alpha \in [0, 1]$, clipping exponent $\beta \leq \frac{1}{2}$, $\theta_0 \in \Theta$ initialization, $m, M$
2: `TIME-HORIZON` $\leftarrow 1$, $\eta \leftarrow 1$, $\lambda \leftarrow 2m(G + 1)$
3: **for** each time $t = 1, 2, \cdots,$ **do**
4:  **if** $t >$`TIME-HORIZON` **then**
5:   `TIME-HORIZON` $\leftarrow 2*$ `TIME-HORIZON`
6:   $\eta \leftarrow \frac{1}{m\texttt{TIME-HORIZON}^\alpha}$.
7:   $\lambda \leftarrow 2m(G + 1)\texttt{TIME-HORIZON}^\beta$
8:  **end if**
9:  Receive sample $X_t$
10:  Output $\theta_t \leftarrow \mathcal{P}_\Theta\left(\theta_{t-1} - \eta\texttt{clip}(\nabla\mathcal{L}(X_t, \theta_{t-1}), \lambda)\right)$
11: **end for**
---

**Proposition 14.1** ([11]). *If for a given $\alpha, \beta, \delta, T$, clipped-SGD satisfies $\mathcal{R}_T \leq \mathfrak{R}(\alpha, \beta, T) \ln\left(\frac{T}{\delta}\right)$ with probability at-least $1-\delta$, then Algorithm 2 will satisfy regret $\mathcal{R}_T \leq \mathfrak{R}(\alpha, \beta, T)(\log_2(T))^2 \ln\left(\frac{2}{\delta}\right)$ with probability at-least $1 - \delta$.*

*Proof.* Over a time interval of $T$, there are $\log_2(T)$ times when Line 5 of Algorithm 2 is executed. Denote by the time-interval between two successive executions of Line 5 as a 'phase' of the algorithm. In the $i$th phase, the hypothesis of the proposition gives that with probability at-most $\delta$, the regret in phase $i$ denoted by $\mathcal{R}^{(i)} \geq \mathfrak{R}(\alpha, \beta, T) \ln\left(\frac{2^i}{\delta}\right)$. Thus, taking an union bound over all the $\log_2(T)$

phases, we get that with probability at-least $1 - \log_2(T)\delta$, the total regret $\mathcal{R}_T$ is bounded by

$$\mathcal{R}_T \leq \sum_{i=1}^{\log_2(T)} \mathfrak{R}(\alpha, \beta, T) \ln\left(\frac{2^i}{\delta}\right),$$

$$\leq \mathcal{R}_T \leq \mathfrak{R}(\alpha, \beta, T)(\log_2(T))^2 \ln\left(\frac{2}{\delta}\right).$$

$\square$

## 15  Proof of Proposition 6.1

The proof is by construction. For each time $t = 1, 2, \cdots$, denote by the function $f_t(x) := \frac{1}{2}(x - 1 - H_t)^2$, where $H_t := \sum_{s=1}^{t-1} \frac{1}{s}$. Let the initial point $\theta_0 = 0$ and let us set the clipping value of clipped gradient descent to be larger than or equal to 1.

We will argue by induction that $\theta_t = H_t + \frac{1}{t}$ for all $t = 1, 2, \cdots$. Thus, the loss suffered at time $t$ is $l_t := 1 - \frac{1}{t}$, since the optimal point at all times $t$ is $1 + H_t$.

For the base-case of time 1, we see that the gradient of $f_1(\cdot)$ at $x = \theta_0 = 0$ is $-1$. Thus, $\theta_1 = 1 = H_2$ and the loss at time 1 is $0 = 1 - \frac{1}{1}$. Assume the induction hypothesis is true for all $s = 1, \cdots, t$ for some $t$. Consider the time $t + 1$. The induction hypothesis tells that at time $t$, $\theta_t = H_{t+1}$. At time $t + 1$, the function revealed is $(x - 1 - H_{t+1})^2$ and thus the gradient at any $x = H_{t+1}$ is $-1$. The step size at time $t + 1$ is $\eta_{t+1} = \frac{1}{t+1}$ and thus $\theta_{t+1} = H_{t+1} + \frac{1}{t+1} = H_{t+2}$. Thus, the induction hypothesis holds.

This gives that the total cumulative regret is

$$\mathcal{R}_T = \sum_{t=1}^{T} \left(1 - \frac{1}{t}\right),$$
$$\geq T - 2\ln(T),$$
$$\geq \frac{T}{3}.$$

## 16  Proofs of Lower Bounds on Estimation Regret

Before giving the proof of Proposition 2.6, we state and prove a simpler result that finite diameter is necessary in the presence of both non-stationarities and corruptions.

**Proposition 16.1 (Finite diameter is necessary in general).** *There exists a strongly convex loss function $\mathcal{L}$ such that for every $\mathcal{D} > 0$, domain $\Theta = [-\mathcal{D}/2, \mathcal{D}/2] \subset \mathbb{R}$ and every $T \in \mathbb{N}$, such that for every algorithm, there exists a sequence of probability measures $(\mathbb{P}_t)_{t=1}^{T}$ and corruptions $(\mathcal{C}_t)_{t=1}^{T}$ where regret is at-least $\Omega(\Lambda_T \mathcal{D})$ with probability 1.*

We prove the following Proposition on mean-estimation, which will prove Proposition 16.1.

**Proposition 16.2 (Mean estimation needs finite diameter).** *Let $\Theta = [-\mathcal{D}/2, \mathcal{D}/2] \subset \mathbb{R}$, time-horizon $T$ and the number of corruptions be $\Lambda_T \leq T$ and $\mathcal{L}(Z, \theta) = \frac{1}{2}(Z - \theta)^2$. For every algorithm, there exists a sequence of probability measures $(\mathbb{P}_t)_{t=1}^{T}$ and corruption functions $(\mathcal{C}_t)_{t=1}^{T}$, such that the regret incurred is at-least $\Omega(\Lambda_T \mathcal{D})$ with probability 1.*

*Proof.* The crux of this lower-bound is that the time instants of corruptions are adversarially chosen. The proof follows by construction where the underlying ground-truth is one of two possibilities and there is no noise, i.e., the variance of the probability distributions chosen by the adversary are all 0. We will show that even against an oblivious adversary, $\Omega(\Lambda_T \mathcal{D})$ regret is un-avoidable.

Assume that the adversary has only two choices – either an environment in which the true mean at all times is $\mathcal{D}/2$ or one in which the true mean in the first $T - \Lambda_T$ samples is $\mathcal{D}/2$ and the true mean in the last $\Lambda_T$ samples is $-\mathcal{D}/2$. If the adversary picks the first scenario then it does not corrupt any sample and since there is no noise, all the $T$ observed samples in the first scenario will be equal to $\mathcal{D}/2$. If

the adversary picks the second scenario as the environment, then it will corrupt the last $\Lambda_T$ samples to show as $\mathcal{D}/2$ instead of the true $-\mathcal{D}/2$. Thus, in either choice of the adversary the observed set of $T$ samples by the forecaster is deterministic $\mathcal{D}/2$ for all $T$ samples. Thus, the forecaster cannot distinguish whether the underlying environment is from scenario 1 or 2 from observations, even if the forecaster has knowledge of the two possible situations from which the adversary is choosing.

Thus, for any deterministic choice of outputs by the algorithm, the adversary can choose one of these two situations such that the regret on the last $\Lambda_T$ samples of the stream incurs regret at-least $\frac{\Lambda_T \mathcal{D}}{4}$. $\qquad\square$

## 16.1 Proof of Proposition 2.6

Similar to Proposition 2.6, consider two scenarios for mean-estimation. In one scenario, the un-corrupted samples are all drawn from a Dirac mass at $0$, but the first $\Lambda_T$ samples are corrupted with all $d$ coordinates set to $\mathcal{D}/\sqrt{d}$. In the other scenario, there are no corruptions and the un-corrupted samples are all from Dirac mass at location with all coordinates $\mathcal{D}/\sqrt{d}$. In both situations, the first $\Lambda_T$ samples are identical. Thus, no estimator in the first $\Lambda_T$ samples can distinguish between these two scenarios and will incur regret at-least $(\Lambda_T \mathcal{D})/4$.

# 17 Useful convexity based inequalities

In this section, we collect some inequalities that we will repeatedly use throughout the proofs. Throughout the rest of the paper, for every $t \in [T]$, we denote by $\mathcal{R}_t(\theta) := \mathbb{E}_{Z \sim \mathbb{P}_t}[\mathcal{L}(Z, \theta)]$ and by $\theta_t^* := \arg\min_{\theta \in \Theta} \mathcal{R}_t(\theta)$. Convexity of $\mathcal{R}_t(\cdot)$ and the domain $\Theta$ being convex implies that $\theta_t^*$ is well-defined, exists and is unique. Recall that for all time $t$, $\mathcal{R}_t(\theta)$ is strongly convex with parameters $M, m$ respectively. Thus, we know that

$$\mathcal{R}_t(\theta_t^*) \geq \mathcal{R}_t(\theta_{t-1}) + \langle \nabla \mathcal{R}_t(\theta_{t-1}), \theta_t^* - \theta_{t-1} \rangle + \frac{m}{2}\|\theta_t^* - \theta_{t-1}\|_2^2. \tag{4}$$

Further since $\theta_t^* := \arg\min_\theta \mathcal{R}_t(\theta)$, we have that

$$\mathcal{R}_t(\theta_{t-1}) - \mathcal{R}_t(\theta_t^*) \geq \frac{m}{2}\|\theta_{t-1} - \theta_t^*\|_2^2.$$

Putting these two together, we see that

$$\langle \nabla \mathcal{R}_t(\theta_{t-1}), \theta_{t-1} - \theta_t^* \rangle \geq m\|\theta_{t-1} - \theta_t^*\|_2^2. \tag{5}$$

Also, We further use the following lemma.

**Lemma 17.1** (Lemma 3.11 from [9]). *Let $f : \mathbb{R}^d \to \mathbb{R}$ be a $M$ smooth and $m$ strongly convex function. Then for all $x, y \in \mathbb{R}^d$,*

$$\langle \nabla f(x) - \nabla f(y), x - y \rangle \geq \frac{mM}{M+m}\|x-y\|_2^2 + \frac{1}{M+m}\|\nabla f(x) - \nabla f(y)\|_2^2.$$

By substituting $x = \theta_{t-1}$, $y = \theta_t^*$ and $f(\cdot) = \mathcal{R}_t(\cdot)$ and by leveraging the fact that $\nabla \mathcal{R}_t(\theta_t^*) = 0$, we get the inequality that

$$\langle \nabla \mathcal{R}_t(\theta_{t-1}), \theta_{t-1} - \theta_t^* \rangle \geq \frac{mM}{m+M}\|\theta_{t-1} - \theta_t^*\|_2^2 + \frac{1}{M+m}\|\nabla \mathcal{R}_t(\theta_{t-1})\|_2^2.$$

Re-arranging, we see that

$$\|\nabla \mathcal{R}_t(\theta_{t-1})\|_2^2 \leq (M+m)\langle \nabla \mathcal{R}_t(\theta_{t-1}), \theta_{t-1} - \theta_t^* \rangle - mM\|\theta_{t-1} - \theta_t^*\|_2^2. \tag{6}$$

Further, using the Cauchy-schwartz inequality that $\langle \nabla \mathcal{R}_t(\theta_{t-1}), \theta_{t-1} - \theta_t^* \rangle \leq \|\nabla \mathcal{R}_t(\theta_{t-1})\|\|\theta_{t-1} - \theta_t^*\| \leq \mathcal{D}\|\nabla \mathcal{R}_t(\theta_{t-1})\|$, we get from the preceeding inequality that

$$\|\nabla \mathcal{R}_t(\theta_{t-1})\| \leq (M+m)\mathcal{D}. \tag{7}$$

## 18  Theorem statement when the gradients are sub-gaussian with known upper-bound

**Theorem 18.1** (Formal version of Theorem 4.3). *Suppose Assumption 4.2 holds. For every $\delta \in (0, 1)$, if Algorithm 1 is run with clipping value $\lambda \geq G + \sqrt{Tr(\Sigma)} + C\left(\sqrt{\nu_{max}(\Sigma)\ln\left(\frac{2T}{\delta}\right)}\right)$, where $C$ is given in Proposition 19.1 and step-size $\eta > 0$, then with probability at-least $1 - \delta$,*

$$Reg_T \leq \frac{2(l_1 + \Phi_T)}{\eta m} + T\left(C\sqrt{\nu_{max}(\Sigma)\ln\left(\frac{6T}{\delta}\right)} + \sqrt{Tr(\Sigma)}\right)\sqrt{\frac{2\eta}{m}}$$

$$+ \Lambda_T\left(\frac{2\lambda}{m} + \sqrt{\frac{4\lambda\mathcal{D}}{\eta}}\right) + 2\sqrt{\frac{T\Phi_T\left(C\sqrt{\nu_{max}(\Sigma)\ln\left(\frac{2T}{\delta}\right)} + \sqrt{Tr(\Sigma)}\right)}{\eta m^2}} \tag{8}$$

**Corollary 18.2.** *Suppose Assumption 4.2 holds. For every $\delta \in (0, 1)$, if Algorithm 1 is run with clipping value $\lambda \geq G + \sqrt{Tr(\Sigma)} + C\left(\sqrt{\nu_{max}(\Sigma)\ln\left(\frac{2T}{\delta}\right)}\right)$, where $C$ is given in Proposition 19.1 and step-size $\eta = \frac{1}{mT^\alpha}$ for $\alpha \in [0, 1)$, then with probability at-least $1 - \delta$,*

$$Reg_T \leq \mathcal{O}\underbrace{\left(\Phi_T T^\alpha + \sqrt{\frac{T^{1+\alpha}\Phi_T\left(C\sqrt{\nu_{max}(\Sigma)\ln\left(\frac{2T}{\delta}\right)} + \sqrt{Tr(\Sigma)}\right)}{m}}\right)}_{\text{Regret due to distribution shift}} +$$

$$\underbrace{\mathcal{O}\left(\frac{T^{1-\frac{\alpha}{2}}}{m}\left(\sqrt{\nu_{max}(\Sigma)\ln\left(\frac{6T}{\delta}\right)} + \sqrt{Tr(\Sigma)}\right)\right)}_{\text{Regret due to standard statistical error}} + \underbrace{\mathcal{O}\left(\Lambda_T T^{\frac{\alpha}{2}}\sqrt{m\lambda\mathcal{D}}\right)}_{\text{Regret due to adversarial corruptions}} . \tag{9}$$

The above theorem immediately yields the following corollary by substituting the learning rate as $\eta = T^{-2/3}$.

**Corollary 18.3.** *Suppose Assumption 4.2 holds. For every $\delta \in (0, 1)$, if Algorithm 1 is run with clipping value $\lambda \geq G + \sqrt{Tr(\Sigma)} + C\left(\sqrt{\nu_{max}(\Sigma)\ln\left(\frac{2T}{\delta}\right)}\right)$, where $C$ is given in Proposition 19.1 and step-size $\eta = \frac{T^{-2/3}}{m}$, then with probability at-least $1 - \delta$,*

$$Reg_T \leq \mathcal{O}\underbrace{\left(\Phi_T T^{2/3} + \sqrt{\frac{T^{5/3}\Phi_T\left(C\sqrt{\nu_{max}(\Sigma)\ln\left(\frac{2T}{\delta}\right)} + \sqrt{Tr(\Sigma)}\right)}{m}}\right)}_{\text{Regret due to distribution shift}} +$$

$$\underbrace{\mathcal{O}\left(\frac{T^{2/3}}{m}\left(\sqrt{\nu_{max}(\Sigma)\ln\left(\frac{6T}{\delta}\right)} + \sqrt{Tr(\Sigma)}\right)\right)}_{\text{Regret due to standard statistical error}} + \underbrace{\mathcal{O}\left(\Lambda_T T^{1/3}\sqrt{m\lambda\mathcal{D}}\right)}_{\text{Regret due to adversarial corruptions}} . \tag{10}$$

## 19  Proof of Theorem 18.1

The proof in this section is based on the following result on high-dimensional sub-gaussian random vectors.

**Proposition 19.1** (Exercise 6.3.5 [57]). *There exists an absolute constant $C > 80$, such that for every sub-gaussian random-vector $Z$ with $0$ mean random-vector and covariance matrix $\Sigma$ and every $\delta \in (0,1)$, the following*

$$\mathbb{P}\left[\left|\|Z\|_2 - \mathbb{E}[\|Z\|_2]\right| \geq C\sqrt{\nu_{max}(\Sigma)\ln\left(\frac{2}{\delta}\right)}\right] \leq \delta,$$

*holds, where $\nu_{max}(\Sigma)$ is the largest eigen-value of the upper-bound covariance matrix $\Sigma$.*

### 19.0.1 Notations

We follow the same proof architecture as that of [55]. We define three sequences of random variables $(\psi_t)_{t\geq 1}$ and $(\bar{\psi})_{t\geq 1}$ as follows.

$$\psi_t := \text{clip}(\nabla\mathcal{L}(Z_t, \theta_{t-1}), \lambda) - \nabla\mathcal{R}_t(\theta_{t-1}),$$
$$\bar{\psi}_t := \text{clip}(\nabla\mathcal{L}(X_t, \theta_{t-1}), \lambda) - \nabla\mathcal{R}_t(\theta_{t-1}),$$
$$\widetilde{\psi}_t := \nabla\mathcal{L}(Z_t, \theta_{t-1}) - \nabla\mathcal{R}_t(\theta_{t-1}).$$

These are random vectors since $Z_t \sim \mathbb{P}_t$ and both $C_t$ and $\theta_t$ are measurable with respect to the sigma algebra generated by $\sigma(Z_1, \cdots, Z_t, C_1, \cdots, C_{t-1})$. Clearly, for all times $t \geq 1$, on the event that $C_t = 0$, $\psi_t = \bar{\psi}_t$ holds almost-surely. Furthermore, from triangle inequality almost-surely for all time $t \geq 1$, we have

$$\|\psi_t\|_2^2 \leq \|\bar{\psi}_t\|_2^2 + 2\lambda^2 \mathbf{1}_{C_t \neq 0}. \tag{11}$$

Recall from Assumption 2.5 that for every $t$ and $\theta \in \Theta$, the covariance matrix of the random vector $\nabla\mathcal{L}(Z, \theta) - \nabla\mathcal{R}_t(\theta)$ is bounded from above in the positive semi-definite sense by $\Sigma$. Denote by $\text{Trace}(\Sigma)$ and $\nu_{\max}(\Sigma)$ as the trace and the highest eigen value respectively of the matrix $\Sigma$. Assumption 4.2 implies the following two lemmas.

Denote by the event $\mathcal{E}_{\text{no-clip}}$ as

$$\mathcal{E}_{\text{no-clip}} = \bigcap_{t=1}^{T}\left\{\nabla\mathcal{L}(Z_t, \theta_{t-1}) = \text{clip}(\nabla\mathcal{L}(Z_t, \theta_{t-1}))\right\},$$
$$= \bigcap_{t=1}^{T}\{\psi_t = \widetilde{\psi}_t\}. \tag{12}$$

### 19.0.2 Supporting Lemmas based on sub-gaussian concentrations

The main result of this section is stated at the end in Corollary 19.5. In order to state them, we build a series of useful bounds along the way. The following lemma bounds the estimation error of the true gradient using the sample $Z_t$ at each time $t$.

**Lemma 19.2.**

$$\sup_{\widetilde{\theta}_1, \cdots, \widetilde{\theta}_T \in \Theta} \mathbb{P}_{Z_1, \cdots, Z_T}\left[\sup_{t \in [T]}\|\nabla\mathcal{L}(Z_t, \widetilde{\theta}_t) - \nabla\mathcal{R}_t(\widetilde{\theta}_t)\| > C\sqrt{\nu_{max}(\Sigma)\ln\left(\frac{2T}{\delta}\right)} + \sqrt{Tr(\Sigma)}\right] \leq \delta.$$

In words, the above lemma states that for any sequence of $\widetilde{\theta}_1, \cdots, \widetilde{\theta}_T$, the probability that the norm of the difference between $\nabla\mathcal{L}(Z_t, \widetilde{\theta}_t)$ and its expectation $\nabla\mathcal{R}_t(\widetilde{\theta}_t)$ where $Z_t \sim \mathbb{P}_t$ exceeds $C\sqrt{\nu_{max}(\Sigma)\ln\left(\frac{2T}{\delta}\right)}$ is bounded above by $\delta$.

*Proof.* Fix any deterministic $\widetilde{\theta}_1, \cdots, \widetilde{\theta}_T \in \Theta$. Assumption 4.2 yields that for all time $t$, $\nabla\mathcal{L}(Z_t, \widetilde{\theta}_t) - \nabla\mathcal{R}_t(\widetilde{\theta}_t)$ is a $0$ mean sub-gaussian random vector with covariance matrix upper-bounded in the positive definite sense by $\Sigma$. Observe from Jensen's inequality that for all $t$ and $\theta \in \theta$

$$\mathbb{E}[\|\nabla\mathcal{L}(Z_t, \theta) - \nabla\mathcal{R}_t(\theta)\|] \leq \sqrt{\mathbb{E}[\|\nabla\mathcal{L}(Z_t, \theta) - \nabla\mathcal{R}_t(\theta)\|^2]} \leq \sqrt{\text{Tr}(\Sigma)}, \tag{13}$$

Now, Hoeffding's inequality in Proposition 19.1 gives that

$$\|\nabla\mathcal{L}(Z_t, \widetilde{\theta}_t) - \nabla\mathcal{R}_t(\widetilde{\theta}_t)\| \leq C\sqrt{\nu_{max}(\Sigma)\ln\left(\frac{2T}{\delta}\right)} + \mathbb{E}[\|\nabla\mathcal{L}(Z_t, \widetilde{\theta}_t) - \nabla\mathcal{R}_t(\widetilde{\theta}_t)\|],$$

$$\leq C\sqrt{\nu_{max}(\Sigma)\ln\left(\frac{2T}{\delta}\right)} + \sqrt{\mathrm{Tr}(\Sigma)}$$

holds with probability at-least $1 - \frac{\delta}{T}$. Now, taking an union bound over $t = \{1, \cdots, T\}$ gives that

$$\mathbb{P}_{Z_1,\cdots,Z_T}\left[\sup_{t\in[T]}\|\nabla\mathcal{L}(Z_t, \widetilde{\theta}_t) - \nabla\mathcal{R}_t(\widetilde{\theta}_t)\| > C\sqrt{\nu_{max}(\Sigma)\ln\left(\frac{2T}{\delta}\right)} + \sqrt{\mathrm{Trace}(\Sigma)}\right] \leq \delta.$$

Now, since $\widetilde{\theta}_1, \cdots, \widetilde{\theta}_T \in \Theta$ was arbitrary, we can take a supremum on both sides and that concludes the proof. $\square$

**Corollary 19.3.** *With probability at-least $1 - \delta$, for all times $t \in \{1, \cdots, T\}$,*

$$\|\widetilde{\psi}_t\| \leq C\sqrt{\nu_{max}(\Sigma)\ln\left(\frac{2T}{\delta}\right)} + \sqrt{Tr(\Sigma)}.$$

*Proof.* The proof follows from applying the Lemma 19.2 to the set of random locations $\theta_1, \cdots, \theta_T$. $\square$

**Lemma 19.4.** *If Algorithm 1 is run with clipping value $\lambda \geq G + \sqrt{Tr(\Sigma)} + C\left(\sqrt{\nu_{max}(\Sigma)\ln\left(\frac{2T}{\delta}\right)}\right)$, where $C$ is given in Proposition 19.1, then with probability at-least $1 - \delta$ gradient is never clipped for any un-corrupted sample, i.e., with probability at-least $1 - \delta$, event $\mathcal{E}_{no\text{-}clip}$ holds.*

*Proof of Lemma 19.4.* From the definition, we know that the gradients are never clipped for the un-corrupted samples if $\|\nabla\mathcal{L}(\theta_{t-1}, Z_t)\| \leq \lambda$ holds for all $t$.

From Corollary 19.3, we know that with probability at-least $1 - \delta$, for all $t \in \{1, \cdots, T\}$, we have

$$\|\nabla\mathcal{L}(Z_t, \theta_{t-1}) - \nabla\mathcal{R}_t(\theta_{t-1})\| \leq C\sqrt{\nu_{max}(\Sigma)\ln\left(\frac{2T}{\delta}\right)} + \sqrt{\mathrm{Tr}(\Sigma)}. \tag{14}$$

From the triangle inequality we know that

$$\|\nabla\mathcal{L}(Z, \theta_{t-1})\| \leq \|\nabla\mathcal{L}(Z, \theta_{t-1}) - \nabla\mathcal{R}_t(\theta_{t-1})\| + \|\nabla\mathcal{R}_t(\theta_{t-1})\|,$$
$$\leq \|\nabla\mathcal{L}(Z, \theta_{t-1}) - \nabla\mathcal{R}_t(\theta_{t-1})\| + \sup_{\theta\in\Theta}\|\nabla\mathcal{R}_t(\theta)\|,$$
$$= \|\nabla\mathcal{L}(Z, \theta_{t-1}) - \nabla\mathcal{R}_t(\theta_{t-1})\| + G. \tag{15}$$

Substituting the bound from Equation (14) into (15), we get that for every $t \in [T]$,

$$\|\nabla\mathcal{L}(Z, \theta_{t-1})\| \leq G + \sqrt{\mathrm{Tr}(\Sigma)} + C\left(\sqrt{\nu_{max}(\Sigma)\ln\left(\frac{2T}{\delta}\right)}\right) \leq \lambda, \tag{16}$$

holds with probability at-least $1 - \delta$.

$\square$

This immediately yields the following corollary.

**Corollary 19.5.** *If Algorithm 1 is run with clipping value $\lambda \geq G + \sqrt{Tr(\Sigma)} + C\left(\sqrt{\nu_{max}(\Sigma)\ln\left(\frac{2T}{\delta}\right)}\right)$, where $C$ is given in Proposition 19.1, then with probability at-least $1 - 2\delta$, for all times $t \in [T]$,*

$$\|\psi_t\| \leq C\sqrt{\nu_{max}(\Sigma)\ln\left(\frac{2T}{\delta}\right)} + \sqrt{Tr(\Sigma)}.$$

*Proof.* On the event that both Corollary 19.3 and Lemma 19.4 holds, we have the result. Both those events fail to hold with probability at-most $2\delta$. □

Denote by the event

$$\mathcal{E}_\psi = \bigcap_{t=1}^{T} \left\{ \|\psi_t\| \leq C\sqrt{\nu_{max}(\Sigma) \ln\left(\frac{2T}{\delta}\right)} + \sqrt{\text{Tr}(\Sigma)} \right\}. \tag{17}$$

Corollary 19.5 states that $\mathbb{P}[\mathcal{E}_\psi] \geq 1 - 2\delta$.

## 19.1 Lemmas based on expanding the gradient descent recursion

**Lemma 19.6.** *Under event $\mathcal{E}_\psi$, for every time $t \in [T]$,*

$$\|\theta_t^* - \theta_t\|_2^2 \leq (1 - \eta m)^{t-1}\|\theta_1 - \theta_1^*\|_2^2 + \sum_{s=1}^{t-1}(1 - \eta m)^{t-s}\|\theta_s^* - \theta_{s+1}^*\|_2^2 + \frac{2\eta B^2}{m}$$

$$-2\eta\sum_{s=1}^{t-1}(1 - \eta m)^{t-s-1}\langle\theta_s - \theta_s^*, \psi_{s+1}\rangle + 2\eta B\sum_{s=1}^{t-1}(1 - \eta m)^{t-s-1}\|\theta_s^* - \theta_{s+1}^*\|$$

$$+ 4\eta^2\lambda^2\sum_{s=2}^{t-1}(1 - \eta m)^{t-s}\mathbf{1}_{C_s \neq 0} + 4\eta\lambda\mathcal{D}\sum_{s=2}^{t-1}(1 - \eta m)^{t-s}\mathbf{1}_{C_s \neq 0}, , \tag{18}$$

*where $B = C\sqrt{\nu_{max}(\Sigma) \ln\left(\frac{2T}{\delta}\right)} + \sqrt{Tr(\Sigma)}$ holds.*

### 19.1.1 Proof of Lemma 19.6

*Proof.* Consider any time $t$. We have

$$\|\theta_t - \theta_t^*\|_2^2 = \|\mathcal{P}_\Theta(\theta_{t-1} - \eta\text{clip}(\mathcal{L}(X_t, \theta_{t-1}), \lambda)) - \theta_t^*\|_2^2, \tag{19}$$

$$\overset{(a)}{\leq} \|\theta_{t-1} - \eta\text{clip}(\mathcal{L}(X_t, \theta_{t-1}), \lambda) - \theta_t^*\|_2^2, \tag{20}$$

$$= \|\theta_{t-1} - \eta(\bar{\psi}_t + \nabla\mathcal{R}_t(\theta_{t-1})) - \theta_t^*\|_2^2,$$

$$= \|\theta_{t-1} - \theta_t^*\|_2^2 + \eta^2\|\bar{\psi}_t + \nabla\mathcal{R}_t(\theta_{t-1})\|_2^2 - 2\eta\langle\theta_{t-1} - \theta_t^*, \bar{\psi}_t + \nabla\mathcal{R}_t(\theta_{t-1})\rangle,$$

$$\overset{(b)}{\leq} \|\theta_{t-1} - \theta_t^*\|_2^2 + 2\eta^2\|\bar{\psi}_t\|_2^2 + 2\eta^2\|\nabla\mathcal{R}_t(\theta_{t-1})\|_2^2 - 2\eta\langle\theta_{t-1} - \theta_t^*, \bar{\psi}_t + \nabla\mathcal{R}_t(\theta_{t-1})\rangle, \tag{21}$$

Step $(a)$ follows since $\Theta$ is a convex set, $\|\mathcal{P}_\Theta(\theta_t) - \theta_t^*\| \leq \|\theta_t - \theta_t^*\|$, since $\theta_t^* \in \Theta$. In step $(b)$, we use the fact that $\|a + b\|_2^2 \leq 2\|a\|_2^2 + 2\|b\|_2^2$, for all $a, b \in \mathbb{R}^d$. Substituting Equation (6) into (21), we get that

$$\|\theta_t^* - \theta_t\|_2^2 \leq \|\theta_{t-1} - \theta_t^*\|_2^2 + 2\eta^2\|\bar{\psi}_t\|_2^2 - 2\eta\langle\theta_{t-1} - \theta_t^*, \bar{\psi}_t\rangle$$
$$+ 2\eta^2\left((M + m)\langle\nabla\mathcal{R}_t(\theta_{t-1}), \theta_{t-1} - \theta_t^*\rangle - mM\|\theta_{t-1} - \theta_t^*\|_2^2\right) - 2\eta\langle\nabla\mathcal{R}_t(\theta_{t-1}), \theta_{t-1} - \theta_t^*\rangle.$$

Re-arranging the equation above yields

$$\|\theta_t^* - \theta_t\|_2^2 \leq (1 - 2\eta^2 mM)\|\theta_{t-1} - \theta_t^*\|_2^2 + 2\eta^2\|\bar{\psi}_t\|_2^2 - 2\eta\langle\theta_{t-1} - \theta_t^*, \bar{\psi}_t\rangle$$
$$- 2\eta(1 - \eta((M + m))\langle\nabla\mathcal{R}_t(\theta_{t-1}), \theta_{t-1} - \theta_t^*\rangle.$$

Further substituting Equation (5) into the display above yields that

$$\|\theta_t^* - \theta_t\|_2^2 \leq (1 - 2\eta m + 2\eta^2 m^2)\|\theta_{t-1} - \theta_t^*\|_2^2 + 2\eta^2\|\bar{\psi}_t\|_2^2 - 2\eta\langle\theta_{t-1} - \theta_t^*, \bar{\psi}_t\rangle,$$
$$\leq (1 - \eta m)\|\theta_{t-1} - \theta_t^*\|_2^2 + 2\eta^2\|\bar{\psi}_t\|_2^2 - 2\eta\langle\theta_{t-1} - \theta_t^*, \bar{\psi}_t\rangle,$$

where the inequality comes from the fact that if $\eta m < 1 \implies 2\eta m - 2\eta^2 m^2 > \eta m$. We simplify the display above using the inequality in Equation (45) as,

$$\|\theta_t^* - \theta_t\|_2^2 \le (1-\eta m)\|\theta_{t-1} - \theta_{t-1}^*\|_2^2 + (1-\eta m)\|\theta_{t-1}^* - \theta_t^*\|_2^2$$
$$+ 2\eta^2\|\psi_t\|_2^2 + 4\eta^2\lambda^2 \mathbf{1}_{C_t \neq 0} - 2\eta\langle\theta_{t-1} - \theta_t^*, \psi_t\rangle + 2\eta\langle\theta_{t-1} - \theta_t^*, \psi_t - \bar\psi_t\rangle. \quad (22)$$

Using the Cauchy-Schartz inequality that $\langle\theta_{t-1} - \theta_t^*, \psi_t - \bar\psi_t\rangle \le \|\theta_{t-1} - \theta_t^*\|\|\bar\psi_t - \psi_t\| \le 2\lambda\|\theta_{t-1} - \theta_t^*\|\mathbf{1}_{C_t \neq 0}$, where the last inequality comes from the fact that for all time $t$, $\|\bar\psi_t - \psi_t\| \le 2\lambda\mathbf{1}_{C_t \neq 0}$ almost-surely. Plugging this into Equation (22) yields

$$\|\theta_t^* - \theta_t\|_2^2 \le (1-\eta m)\|\theta_{t-1} - \theta_{t-1}^*\|_2^2 + (1-\eta m)\|\theta_{t-1}^* - \theta_t^*\|_2^2$$
$$+ 2\eta^2\|\psi_t\|_2^2 + 4\eta^2\lambda^2 \mathbf{1}_{C_t \neq 0} - 2\eta\langle\theta_{t-1} - \theta_t^*, \psi_t\rangle + 4\eta\lambda\|\theta_{t-1} - \theta_t^*\|\mathbf{1}_{C_t \neq 0}. \quad (23)$$

Using the fact that the diameter of the set $\Theta$ is $\mathcal{D}_\Theta$ now yields that

$$\|\theta_t^* - \theta_t\|_2^2 \le (1-\eta m)\|\theta_{t-1} - \theta_{t-1}^*\|_2^2 + (1-\eta m)\|\theta_{t-1}^* - \theta_t^*\|_2^2$$
$$+ 2\eta^2\|\psi_t\|_2^2 + 4\eta^2\lambda^2 \mathbf{1}_{C_t \neq 0} - 2\eta\langle\theta_{t-1} - \theta_t^*, \psi_t\rangle + 4\eta\lambda\mathcal{D}\mathbf{1}_{C_t \neq 0}.$$

Unrolling the recursion yields,

$$\|\theta_t^* - \theta_t\|_2^2 \le (1-\eta m)^{t-1}\|\theta_1 - \theta_1^*\|_2^2 + \sum_{s=1}^{t-1}(1-\eta m)^s\|\theta_{t-s}^* - \theta_{t-s+1}^*\|_2^2 + 2\eta^2\sum_{s=1}^{t-1}(1-\eta m)^{s-1}\|\psi_{t-s+1}\|_2^2 +$$

$$4\eta^2\lambda^2\sum_{s=1}^{t-1}(1-\eta m)^{s-1}\mathbf{1}_{C_{t-s+1}\neq 0} - 2\eta\sum_{s=1}^{t-1}(1-\eta m)^{s-1}\langle\theta_{t-s} - \theta_{t-s+1}^*, \psi_{t-s+1}\rangle + 4\eta\lambda\mathcal{D}\sum_{s=1}^{t-1}(1-\eta m)^{s-1}\mathbf{1}_{C_{t-s+1}\neq 0}.$$

Using the fact that $\sum_{s=1}^{t-1}(1-\eta m)^{s-1} \le \frac{1}{\eta m}$, we get that

$$\|\theta_t^* - \theta_t\|_2^2 \le (1-\eta m)^{t-1}\|\theta_1 - \theta_1^*\|_2^2 + \sum_{s=1}^{t-1}(1-\eta m)^s\|\theta_{t-s}^* - \theta_{t-s+1}^*\|_2^2$$

$$+ 2\eta^2\sum_{s=1}^{t-1}(1-\eta m)^{s-1}\underbrace{\|\psi_{t-s+1}\|_2^2}_{\text{Term I}} \underbrace{-2\eta\sum_{s=1}^{t-1}(1-\eta m)^{s-1}\langle\theta_{t-s} - \theta_{t-s+1}^*, \psi_{t-s+1}\rangle}_{\text{Term II}}$$

$$+ 4\eta^2\lambda^2\sum_{s=1}^{t-1}(1-\eta m)^{s-1}\mathbf{1}_{C_{t-s+1}\neq 0} + 4\eta\lambda\mathcal{D}\sum_{s=1}^{t-1}(1-\eta m)^{s-1}\mathbf{1}_{C_{t-s+1}\neq 0}. \quad (24)$$

Changing the variables in the summation, the above inequality can be written as

$$\|\theta_t^* - \theta_t\|_2^2 \le (1-\eta m)^{t-1}\|\theta_1 - \theta_1^*\|_2^2 + \sum_{s=1}^{t-1}(1-\eta m)^{t-s}\|\theta_s^* - \theta_{s+1}^*\|_2^2$$

$$+ 2\eta^2\sum_{s=2}^{t-1}(1-\eta m)^{t-s}\underbrace{\|\psi_s\|_2^2}_{\text{Term I}} \underbrace{-2\eta\sum_{s=1}^{t-1}(1-\eta m)^{t-s-1}\langle\theta_s - \theta_{s+1}^*, \psi_{s+1}\rangle}_{\text{Term II}}$$

$$+ 4\eta^2\lambda^2\sum_{s=2}^{t-1}(1-\eta m)^{t-s}\mathbf{1}_{C_s\neq 0} + 4\eta\lambda\mathcal{D}\sum_{s=2}^{t-1}(1-\eta m)^{t-s}\mathbf{1}_{C_s\neq 0}. \quad (25)$$

Simplifying term I from Corollary 19.5 since event $\mathcal{E}_\psi$ holds by assumption of the Lemma, we get that with probability at-least $1 - 2\delta$,

$$\|\theta_t^* - \theta_t\|_2^2 \le (1-\eta m)^{t-1}\|\theta_1 - \theta_1^*\|_2^2 + \sum_{s=1}^{t-1}(1-\eta m)^{t-s}\|\theta_s^* - \theta_{s+1}^*\|_2^2$$

$$+ \underbrace{\frac{2\eta B^2}{m} - 2\eta\sum_{s=1}^{t-1}(1-\eta m)^{t-s-1}\langle\theta_s - \theta_s^*, \psi_{s+1}\rangle}_{\text{Term II}} - 2\eta\sum_{s=1}^{t-1}(1-\eta m)^{t-s-1}\langle\theta_s^* - \theta_{s+1}^*, \psi_{s+1}\rangle$$

$$+ 4\eta^2\lambda^2 \sum_{s=2}^{t-1}(1-\eta m)^{t-s}\mathbf{1}_{C_s \neq 0} + 4\eta\lambda\mathcal{D}\sum_{s=2}^{t-1}(1-\eta m)^{t-s}\mathbf{1}_{C_s \neq 0}\cdot, \tag{26}$$

where $B = C\sqrt{\nu_{max}(\Sigma)\ln\left(\frac{2T}{\delta}\right)} + \sqrt{\text{Tr}(\Sigma)}$. Further, simplifying using Cauchy Schwartz inequality, we get the stated result of Lemma 19.6.

$$\|\theta_t^* - \theta_t\|_2^2 \leq (1-\eta m)^{t-1}\|\theta_1 - \theta_1^*\|_2^2 + \sum_{s=1}^{t-1}(1-\eta m)^{t-s}\|\theta_s^* - \theta_{s+1}^*\|_2^2$$

$$+ \frac{2\eta B^2}{m}\underbrace{-2\eta\sum_{s=1}^{t-1}(1-\eta m)^{t-s-1}\langle\theta_s - \theta_s^*, \psi_{s+1}\rangle}_{\text{Term II}} + 2\eta B\sum_{s=1}^{t-1}(1-\eta m)^{t-s-1}\Phi_s$$

$$+ 4\eta^2\lambda^2\sum_{s=2}^{t-1}(1-\eta m)^{t-s}\mathbf{1}_{C_s \neq 0} + 4\eta\lambda\mathcal{D}\sum_{s=2}^{t-1}(1-\eta m)^{t-s}\mathbf{1}_{C_s \neq 0}\cdot, \tag{27}$$

□

## 19.2  Martingale Analysis of Term II

Denote by the constant

$$U_t \leq \left[(1-\eta m)^{t-1}\|\theta_1 - \theta_1^*\|_2^2 + \sum_{s=1}^{t-1}(1-\eta m)^{t-s}\|\theta_s^* - \theta_{s+1}^*\|_2^2 + 2\eta B\sum_{s=1}^{t-1}(1-\eta m)^{t-s-1}\Phi_s\right.$$

$$\left. + \frac{2\eta B^2}{m} + 4\eta^2\lambda^2\sum_{s=2}^{t-1}(1-\eta m)^{t-s}\mathbf{1}_{C_s \neq 0} + 4\eta\lambda\mathcal{D}\sum_{s=2}^{t-1}(1-\eta m)^{t-s}\mathbf{1}_{C_s \neq 0}\right]. \tag{28}$$

The following proposition holds by simple algebraic manipulation.

**Proposition 19.7.** *For every $t \in [2, T]$ and $s \leq t$,*

$$(1-\eta m)^{t-s}U_s \leq U_t.$$

Denote by the sequence $(\xi_s)_{s=1}^T$ of random vectors as

$$\xi_s = \begin{cases} \theta_s^* - \theta_s & \text{if } \|\theta_s^* - \theta_s\|^2 \leq 2U_s \\ 0 & \text{otherwise} \end{cases} \tag{29}$$

Denote by the event

$$\mathcal{E}_\xi = \bigcap_{t=1}^T \left\{\sum_{s=1}^{t-1}(1-\eta m)^{t-s}\langle\xi_s, \psi_{s+1}\rangle \leq \sqrt{\frac{20U_t\nu_{max}(\Sigma)\ln\left(\frac{2T}{\delta}\right)}{\eta m}}\right\} \tag{30}$$

**Lemma 19.8.**

$$\mathbb{P}[\mathcal{E}_\xi] \geq 1 - \delta. \tag{31}$$

Recall the definition of event $\mathcal{E}_\psi$ in Equation (17).

Corollary 19.5 give that $\mathbb{P}[\mathcal{E}_\psi] \geq 1 - 2\delta$.

**Lemma 19.9.** *Under the events $\mathcal{E}_\xi$ and $\mathcal{E}_\psi$,*

$$\|\theta_t^* - \theta_t\|_2^2 \leq 2U_t, \tag{32}$$

*holds for all $t \in [T]$.*

## 19.3 Proof of Theorem 18.1

*Proof.* Corollary 19.5 gives that $\mathbb{P}[\mathcal{E}_\psi^C] \leq 2\delta$. Lemma 19.8 gives that $\mathbb{P}[\mathcal{E}_\xi^c] \leq \delta$. An union bound argument thus gives that $\mathbb{P}[\mathcal{E}_\psi^C \cup \mathcal{E}_\xi^C] \leq 2\delta$. Thus, we have that both events $\mathcal{E}_\psi$ and $\mathcal{E}_\xi$ hold with probability at-least $1 - 3\delta$, i.e.,

$$\mathbb{P}[\mathcal{E}_\psi \cap \mathcal{E}_\xi] \geq 1 - 3\delta.$$

Under the event $\mathcal{E}_\psi \cap \mathcal{E}_\xi$, Lemma 19.9 gives that

$$\|\theta_t - \theta_t^*\| \leq \sqrt{2U_t}.$$

Now, applying the fact that $\sqrt{\sum_i x_i} \leq \sum_i \sqrt{x_i}$, we can simplify $\sqrt{U_t}$ as

$$
\begin{aligned}
\sqrt{U_t} \leq \sqrt{2}\Bigg[ & (1-\eta m)^{\frac{t-1}{2}}\|\theta_1 - \theta_1^*\|_2 + \sum_{s=1}^{t-1}(1-\eta m)^{\frac{s}{2}}\|\theta_{t-s}^* - \theta_{t-s+1}^*\| \\
& + \left(C\sqrt{\nu_{max}(\Sigma)\ln\left(\frac{2T}{\delta}\right)} + \sqrt{\mathrm{Tr}(\Sigma)}\right)\sqrt{\frac{2\eta}{m}} \\
& + \sqrt{2\eta\left(C\sqrt{\nu_{max}(\Sigma)\ln\left(\frac{2T}{\delta}\right)} + \sqrt{\mathrm{Tr}(\Sigma)}\right)\sum_{s=1}^{t-1}(1-\eta m)^{\frac{t-s-1}{2}}\sqrt{\|\theta_s^* - \theta_{s+1}^*\|}} \\
& + 2\eta\lambda\sum_{s=1}^{t-1}(1-\eta m)^{\frac{s-1}{2}}\mathbf{1}_{C_{t-s+1}\neq 0} + \sqrt{4\eta\lambda\mathcal{D}}\sum_{s=1}^{t-1}(1-\eta m)^{\frac{s-1}{2}}\mathbf{1}_{C_{t-s+1}\neq 0}\Bigg]. \quad (33)
\end{aligned}
$$

Observe the following deterministic identities that can be proven by switching the order of summations.

$$\sum_{s\geq 1}(1-\eta m)^{\frac{s-1}{2}} \leq \frac{2}{\eta m}, \quad (34)$$

$$\sum_{t=1}^{T}\sum_{s=1}^{t-1}(1-\eta m)^{\frac{s-1}{2}}\mathbf{1}_{C_{t-s+1}\neq 0} \leq \frac{2\Lambda_T}{\eta m}, \quad (35)$$

$$\sum_{t=1}^{T}\sum_{s=1}^{t-1}(1-\eta m)^{\frac{s}{2}}\|\theta_{t-s}^* - \theta_{t-s+1}^*\| \leq \frac{2\Phi_T}{\eta m}, \quad (36)$$

$$\sum_{t=1}^{T}\sum_{s=1}^{t-1}(1-\eta m)^{\frac{t-s-1}{2}}\sqrt{\|\theta_s^* - \theta_{s+1}^*\|} \leq \frac{2\sum_{s=1}^{t}\sqrt{\|\theta_s^* - \theta_{s+1}^*\|}}{\eta m} \leq \frac{2\sqrt{T\Phi_T}}{\eta m}. \quad (37)$$

The first inequality above follows from the fact that for all $x \in (0,1)$, we have $\frac{1}{1-\sqrt{1-x}} \leq \frac{2}{x}$ and the last inequality follows from Cauchy-Schwartz inequality that $\sum_{i=1}^{T}\sqrt{x_i} \leq \sqrt{T\sum_{i=1}^{T}x_i}$. Recall that the regret $\mathrm{Reg}_T := \sum_{t=1}^{T}\|\theta_t - \theta_t^*\|$. Thus we get from Equations (32) and (33) that with probability at-least $1 - 3\delta$,

$$
\begin{aligned}
\mathrm{Reg}_T \leq & \frac{2(l_1 + \Phi_T)}{\eta m} + T\left(C\sqrt{\nu_{max}(\Sigma)\ln\left(\frac{2T}{\delta}\right)} + \sqrt{\mathrm{Tr}(\Sigma)}\right)\sqrt{\frac{2\eta}{m}} \\
& + \Lambda_T\left(\frac{2\lambda}{m} + \sqrt{\frac{4\lambda\mathcal{D}}{\eta}}\right) + 2\sqrt{\frac{T\Phi_T\left(C\sqrt{\nu_{max}(\Sigma)\ln\left(\frac{2T}{\delta}\right)} + \sqrt{\mathrm{Tr}(\Sigma)}\right)}{\eta m^2}}, \quad (38)
\end{aligned}
$$

holds with probability at-least $1 - 3\delta$. By a change of variables by re-parametrizing $\delta \to \frac{\delta}{3}$ yields the stated result.

$\square$

## 19.4 Proofs of Lemma 19.9

*Proof.* We prove this lemma by induction on $t$. For the base case of $t = 1$, Equation (32) holds with probability 1 trivially by definition. Now for the induction hypothesis, assume Equation (32) holds for all $s \in \{1, \cdots, t-1\}$ for some $t > 1$. Since we are on the event $\mathcal{E}_\xi$, we Equation (27) holds. Thus,

$$\|\theta_t^* - \theta_t\|_2^2 \leq (1 - \eta m)^{t-1} \|\theta_1 - \theta_1^*\|_2^2 + \sum_{s=1}^{t-1} (1 - \eta m)^{t-s} \|\theta_s^* - \theta_{s+1}^*\|_2^2$$

$$+ \frac{2\eta B^2}{m} \underbrace{-2\eta \sum_{s=1}^{t-1} (1 - \eta m)^{t-s-1} \langle \theta_s - \theta_s^*, \psi_{s+1} \rangle}_{\text{Term II}} + 2\eta B \sum_{s=1}^{t-1} (1 - \eta m)^{t-s-1} \Phi_s$$

$$+ 4\eta^2 \lambda^2 \sum_{s=2}^{t-1} (1 - \eta m)^{t-s} \mathbf{1}_{C_s \neq 0} + 4\eta \lambda \mathcal{D} \sum_{s=2}^{t-1} (1 - \eta m)^{t-s} \mathbf{1}_{C_s \neq 0}. \tag{39}$$

From the induction hypothesis, we have that $\|\theta_s^* - \theta_s\|_2^2 \leq 2U_s$ holds for all $s \in \{1, \cdots, t-1\}$. Thus,

$$\sum_{s=1}^{t-1} (1 - \eta m)^{t-s} \langle \xi_s, \psi_{s+1} \rangle = \sum_{s=1}^{t-1} (1 - \eta m)^{t-s} \langle \theta_s - \theta_s^*, \psi_{s+1} \rangle.$$

Further since we are on the event $\mathcal{E}_\xi$, we have that

$$\sum_{s=1}^{t-1} (1 - \eta m)^{t-s} \langle \xi_s, \psi_{s+1} \rangle \leq \sqrt{\frac{20 U_t \nu_{max}(\Sigma) \ln\left(\frac{2T}{\delta}\right)}{\eta m}}.$$

Thus, plugging this back into Equation (39), we get that

$$\|\theta_t^* - \theta_t\|_2^2 \leq U_t + \sqrt{\frac{80 \eta U_t \nu_{max}(\Sigma) \ln\left(\frac{2T}{\delta}\right)}{m}}.$$

From Equation (28), we see that $\frac{2\eta B^2}{m} \leq U_t$. Substituting this in the above equation, we see that

$$\|\theta_t^* - \theta_t\|_2^2 \leq U_t + \frac{U_t \sqrt{40 \nu_{max}(\Sigma) \ln\left(\frac{2T}{\delta}\right)}}{B},$$

$$\leq 2U_t$$

The second inequality follows since $B = C\sqrt{\nu_{max}(\Sigma) \ln\left(\frac{2T}{\delta}\right)} + \sqrt{\text{Tr}(\Sigma)}$, where $C \geq 40$ as given in Proposition 19.1. Thus, $\frac{\sqrt{40 \nu_{max}(\Sigma) \ln\left(\frac{2T}{\delta}\right)}}{B} \leq 1$.

$\square$

## 19.5 Proof of Lemma 19.8

We first reproduce an useful result.

**Lemma 19.10** (Sub-gaussian Martingale Azuma-Hoeffding inequality [53]). *Suppose $Y_1, \cdots, Y_T$ is a martingale sequence with respect to a filtration $(\mathcal{F}_t)_{t=1}^T$, i.e., $\mathbb{E}[Y_t | \mathcal{F}_{t-1}] = 0$ for all $t$. Further, suppose there exists deterministic non-negative numbers $(\sigma_t)_{t=1}^T$ such that for every $\lambda \in \mathbb{R}$ and $t \in [T]$, we have $\mathbb{E}[\exp(\lambda Y_t) | \mathcal{F}_{t-1}] \leq \exp\left(\frac{\lambda^2 \sigma_t^2}{2}\right)$ almost-surely. Then for every $a > 0$,*

$$\mathbb{P}\left[\left|\sum_{t=1}^T Y_t\right| \geq a\right] \leq 2\exp\left(\frac{-3a^2}{28 \sum_{t=1}^T \sigma_t^2}\right)$$

In order to prove Lemma 19.8, we will show that for each $t \in [T]$, the event

$$\mathcal{E}_\xi^{(t)} = \left\{ \sum_{s=1}^{t-1} (1 - \eta m)^{t-s} \langle \xi_s, \psi_{s+1} \rangle \leq 2BU_t \sqrt{t} \ln\left(\frac{T^2}{\delta}\right) \right\}, \tag{40}$$

holds with probability at-least $1 - \frac{\delta}{T}$. Then an union bound will conclude the proof of Lemma 19.8.

To do so, first observe that the sum $\sum_{s=1}^{t-1} (1 - \eta m)^{t-s} \langle \xi_s, \psi_{s+1} \rangle$ is a martingale difference with respect to the filtration $(\mathcal{F}_t)_{t=1}^T$. This follows from two facts.

**Fact I** $\xi_s \in \mathcal{F}_s$ is measurable with respect to the filtration generated by all the random vectors $Z_1, \cdots, Z_s$ and the corruption vectors $C_1, \cdots, C_s$. This follows since the adversary is causal and does not have access to future randomness to decide on corruption levels.

**Fact II** $\mathbb{E}[\psi_{s+1}|\mathcal{F}_s] = 0$ is un-biased.

**Fact III**, The sequence of random variables $((1 - \eta m)^{t-s} \langle \xi_s, \psi_{s+1} \rangle)_{s=1}^t$ satisfies the premise of Lemma 19.10 with $\sigma_s \leq (1 - \eta m)^{2(t-s)} \|\xi_s\|^2 \nu_{max}(\Sigma) \leq 2(1 - \eta m)^{t-s} U_t \nu_{max}(\Sigma)$, for all $1 \leq s \leq t$.

The third fact follows from the definition that conditional on $\mathcal{F}_s$, the covariance matrix of $\psi_{s+1}$ is upper-bounded by $\Sigma$ and $\|\xi_s\|^2 \leq 2U_s$ for all $s$, and $(1 - \eta m)^{t-s} U_s \leq U_t$ holds for all $s \leq t$.

In order to apply Lemma 19.10, observe that the sum of variances $\sum_{s=1}^t \sigma_s^2 = 2U_t \nu_{max}(\Sigma) \sum_{s=1}^t (1 - \eta m)^{t-s} \leq \frac{2U_t \nu_{max}(\Sigma)}{\eta m}$. Applying Lemma 19.10, we get that

$$\mathbb{P}\left[ \sum_{s=1}^t (1 - \eta m)^{t-s} \langle \xi_s, \psi_{s+1} \rangle \geq \sqrt{\frac{20 U_t \nu_{max}(\Sigma) \ln\left(\frac{2T}{\delta}\right)}{\eta m}} \right] \leq \delta.$$

### 19.6 Proof of Corollary 4.5

The finite diameter that $\mathcal{D} < \infty$ fact is only used in Lemma 19.6. From observing the proof of Lemma 19.6, it holds even if $\mathcal{D} = \infty$, albeit the bound is vacuous if $\Lambda_T > 0$. Thus, the entire proof holds verbatim even if $\mathcal{D} = \infty$ and gives non-trivial regret guarantees when $\Lambda_T = 0$.

## 20 Theorem statement and Proofs in the general case from Section 5

**Theorem 20.1** (Formal version of Theorem 5.1). *For every $\delta \in (0,1)$, if Algorithm 1 is run with clipping value $\lambda \geq 2G$, and step-size $\eta > 0$, then with probability at-least $1 - \delta$,*

$$Reg_T \leq \left( \frac{91\sigma}{\lambda\sqrt{m\eta}} + \frac{2\lambda\sqrt{m}}{\sigma} + 2 \right) \left[ \frac{2(l_1 + \Phi_T)}{\eta m} + 4\sigma T \sqrt{\frac{\eta}{\lambda m}} + 7\sigma T \frac{\eta}{m} + 6\eta\lambda T \sqrt{\ln\left(\frac{2T}{\delta}\right)} \right.$$

$$\left. + \frac{6T\eta^{3/4}\sqrt{\lambda\sigma\sqrt{\ln\left(\frac{2T}{\delta}\right)}}}{2m^{1/4}} + 3\sigma\sqrt{\frac{T\Phi_T}{\eta\lambda m^2}} + 2\sqrt{\frac{\lambda T\Phi_T}{\eta}} + \frac{4\lambda\Lambda_T}{\eta m} + 4\Lambda_T \sqrt{\frac{\lambda\mathcal{D}}{\eta m^2}} \right]. \tag{41}$$

**Corollary 20.2.** *For every $\delta \in (0,1)$, if Algorithm 1 is run with clipping value $\lambda = (2GT^\beta)$, and step-size $\eta = \frac{1}{mT^\alpha}$ for $\alpha \in [0,1)$, then with probability at-least $1 - \delta$,*

$$Reg_T \leq \underbrace{\mathcal{O}\left( \left( GT^{\alpha+\beta} + \frac{T^{\frac{3\alpha}{2}-\beta}}{G} \right) \frac{\sigma\Phi_T}{m^{3/2}} + \frac{\sigma}{m^{3/2}} \sqrt{\frac{\Phi_T T^{1+2\alpha-\beta}}{G}} \right)}_{\text{Regret due to distribution shift}} +$$

$$\underbrace{\mathcal{O}\left( \left( \sqrt{G}T^{1-\left(\frac{\alpha-\beta}{2}\right)} + T^{1-\frac{\alpha}{2}} + G^2 T^{1+2\beta-\alpha} \right) \frac{\sigma^2}{m} \ln\left(\frac{T}{\delta}\right) \right)}_{\text{Regret due to finite-sample estimation error}} +$$

$$\mathcal{O}\left(\Lambda_T \frac{\sigma\sqrt{\mathcal{D}}}{m}\left(G^2 T^{2\beta+\alpha} + T^{\frac{3\alpha}{2}} + \frac{T^{\alpha-\frac{\beta}{2}}}{\sqrt{G}} + G^{3/2}T^{\frac{3\beta}{2}+\alpha}\right)\right). \quad (42)$$

$$\underbrace{\phantom{\mathcal{O}\left(\Lambda_T \frac{\sigma\sqrt{\mathcal{D}}}{m}\left(G^2 T^{2\beta+\alpha} + T^{\frac{3\alpha}{2}} + \frac{T^{\alpha-\frac{\beta}{2}}}{\sqrt{G}} + G^{3/2}T^{\frac{3\beta}{2}+\alpha}\right)\right)}}_{\text{Regret due to adversarial corruptions}}$$

**Corollary 20.3.** *For every $\delta \in (0,1)$, if Algorithm 1 is run with clipping value $\lambda = 2GT^{\frac{\alpha}{3}}$, and step-size $\eta = \frac{1}{mT^\alpha}$ for $\alpha \in [0,1]$, then with probability at-least $1-\delta$,*

$$Reg_T \leq \underbrace{\mathcal{O}\left(\frac{G\sigma T^{\frac{4\alpha}{3}}\Phi_T}{m^{3/2}} + \frac{\sigma T^{\frac{1}{2}+\frac{5\alpha}{6}}\sqrt{\Phi_T}}{m^{3/2}\sqrt{G}}\right)}_{\text{Regret due to distribution shift}} + \underbrace{\mathcal{O}\left(\frac{T^{1-\frac{\alpha}{3}}(G\sigma)^2 \ln\left(\frac{T}{\delta}\right)}{m}\right)}_{\text{Regret due to finite-sample estimation error}} + \underbrace{\mathcal{O}\left(\frac{T^{\frac{3\alpha}{2}}\Lambda_T G^2 \sigma\sqrt{\mathcal{D}}}{m}\right)}_{\text{Regret due to adversarial corruptions}}.$$

$$(43)$$

**Corollary 20.4.** *For every $\delta \in (0,1)$, if Algorithm 1 is run with clipping value $\lambda \geq \max(2G, T^{1/5})$, and step-size $\eta = \frac{T^{-1/2}}{m}$, then with probability at-least $1-\delta$,*

$$Reg_T \leq \underbrace{\mathcal{O}\left(T^{5/6}(\Phi_T + \sqrt{\Phi_T})\right)}_{\text{Regret due to distribution shift}} + \underbrace{\mathcal{O}\left(T^{5/6}\mathcal{D}\lambda^{\frac{1}{2}}\left(\frac{Tr(\Sigma)}{m}\right)^{\frac{1}{4}}\ln\left(\frac{T}{\delta}\right)^{\frac{1}{2}}\right)}_{\text{Regret due to finite-sample estimation error}}$$

$$+ \underbrace{\mathcal{O}\left(\Lambda_T \frac{\sqrt{\mathcal{D}}}{m}\left(T^{2\beta+\alpha} + T^{\frac{3\alpha}{2}} + T^{\alpha-\frac{\beta}{2}} + T^{\frac{3\beta}{2}+\alpha}\right)\right)}_{\text{Regret due to adversarial corruptions}}. \quad (44)$$

## 20.1 Proof of Theorem 20.1

We follow a similar proof architecture as that of Theorem 18.1. We define two sequences of random variables $(\psi_t)_{t\geq 1}$ and $(\bar{\psi})_{t\geq 1}$ as follows.

$$\psi_t := \text{clip}(\nabla\mathcal{L}(Z_t, \theta_{t-1}), \lambda) - \nabla\mathcal{R}_t(\theta_{t-1}),$$

and by

$$\bar{\psi}_t := \text{clip}(\nabla\mathcal{L}(X_t, \theta_{t-1}), \lambda) - \nabla\mathcal{R}_t(\theta_{t-1}).$$

These are random vectors since $Z_t \sim \mathbb{P}_t$ and both $C_t$ and $\theta_t$ are measurable with respect to the sigma algebra generated by $\sigma(Z_1, \cdots, Z_t, C_1, \cdots, C_{t-1})$. Clearly, for all times $t \geq 1$, on the event that $C_t = 0$, $\psi_t = \bar{\psi}_t$ holds almost-surely. Furthermore, from triangle inequality almost-surely for all time $t \geq 1$, we have

$$\|\psi_t\|_2^2 \leq \|\bar{\psi}_t\|_2^2 + 2\lambda^2 \mathbf{1}_{C_t \neq 0}. \quad (45)$$

**Expanding the one-step recursion**

Similar to the analysis carried out in the proof of Theorem 18.1, we consider any time $t$ and write the recursion as

$$\|\theta_t - \theta_t^*\|_2^2 = \|\mathcal{P}_\Theta(\theta_{t-1} - \eta\text{clip}(\mathcal{L}(X_t, \theta_{t-1}), \lambda)) - \theta_t^*\|_2^2.$$

Now, this can be expanded *verbatim* as done in Section 19.1 to yield an identical replica of Equation 24.

$$\|\theta_t^* - \theta_t\|_2^2 \leq (1-\eta m)^{t-1}\|\theta_1 - \theta_1^*\|_2^2 + \sum_{s=1}^{t-1}(1-\eta m)^s\|\theta_{t-s}^* - \theta_{t-s+1}^*\|_2^2$$

$$+ 2\eta^2 \sum_{s=1}^{t-1}(1-\eta m)^{s-1}\|\psi_{t-s+1}\|_2^2 - 2\eta\sum_{s=1}^{t-1}(1-\eta m)^{s-1}\langle\theta_{t-s}-\theta^*_{t-s+1}, \psi_{t-s+1}\rangle$$

$$+ 4\eta^2\lambda^2\sum_{s=1}^{t-1}(1-\eta m)^{s-1}\mathbf{1}_{C_{t-s+1}\neq 0} + 4\eta\lambda\mathcal{D}\sum_{s=1}^{t-1}(1-\eta m)^{s-1}\mathbf{1}_{C_{t-s+1}\neq 0}. \qquad (46)$$

Denote by $\psi_t := \psi_t^{(b)} + \psi_t^{(v)}$, where $\psi_t^{(b)} := \mathbb{E}_{Z_t}[\psi_t|\mathcal{F}_{t-1}]$ and $\psi_t^{(v)} := \psi_t - \psi_t^{(b)}$. Using this in the display above and using that fact that $\|a+b\|_2^2 \leq 2\|a\|_2^2 + 2\|b\|_2^2$, we get

$$\|\theta^*_t - \theta_t\|_2^2 \leq (1-\eta m)^{t-1}\|\theta_1-\theta^*_1\|_2^2 + \sum_{s=1}^{t-1}(1-\eta m)^s\|\theta^*_{t-s}-\theta^*_{t-s+1}\|_2^2$$

$$+ 4\eta^2\sum_{s=1}^{t-1}(1-\eta m)^{s-1}\|\psi_{t-s+1}^{(b)}\|_2^2 + 4\eta^2\sum_{s=1}^{t-1}(1-\eta m)^{s-1}\|\psi_{t-s+1}^{(v)}\|_2^2$$

$$- 2\eta\sum_{s=1}^{t-1}(1-\eta m)^{s-1}\langle\theta_{t-s}-\theta^*_{t-s+1}, \psi_{t-s+1}^{(b)}\rangle$$

$$- 2\eta\sum_{s=1}^{t-1}(1-\eta m)^{s-1}\langle\theta_{t-s}-\theta^*_{t-s+1}, \psi_{t-s+1}^{(v)}\rangle$$

$$+ 4\eta^2\lambda^2\sum_{s=1}^{t-1}(1-\eta m)^{s-1}\mathbf{1}_{C_{t-s+1}\neq 0} + 4\eta\lambda\mathcal{D}\sum_{s=1}^{t-1}(1-\eta m)^{s-1}\mathbf{1}_{C_{t-s+1}\neq 0}. \qquad (47)$$

Further simplifying by adding and subtracting $\mathbb{E}_{Z_t}[\|\psi_t^{(v)}\|_2^2|\mathcal{F}_{t-1}]$ to be above display, we get

$$\|\theta^*_t - \theta_t\|_2^2 \leq (1-\eta m)^{t-1}\|\theta_1-\theta^*_1\|_2^2 + \sum_{s=1}^{t-1}(1-\eta m)^s\|\theta^*_{t-s}-\theta^*_{t-s+1}\|_2^2$$

$$+ 4\eta^2\sum_{s=1}^{t-1}(1-\eta m)^{s-1}\|\psi_{t-s+1}^{(b)}\|_2^2 + 4\eta^2\sum_{s=1}^{t-1}(1-\eta m)^{s-1}\mathbb{E}_{Z_{t-s+1}}[\|\psi_{t-s+1}^{(v)}\|_2^2|\mathcal{F}_{t-s}]$$

$$+ 4\eta^2\sum_{s=1}^{t-1}(1-\eta m)^{s-1}(\|\psi_{t-s+1}^{(v)}\|_2^2 - \mathbb{E}_{Z_{t-s+1}}[\|\psi_{t-s+1}^{(v)}\|_2^2|\mathcal{F}_{t-s}])$$

$$- 2\eta\sum_{s=1}^{t-1}(1-\eta m)^{s-1}\langle\theta_{t-s}-\theta^*_{t-s}, \psi_{t-s+1}^{(b)}\rangle - 2\eta\sum_{s=1}^{t-1}(1-\eta m)^{s-1}\langle\theta^*_{t-s}-\theta^*_{t-s+1}, \psi_{t-s+1}^{(b)}\rangle$$

$$- 2\eta\sum_{s=1}^{t-1}(1-\eta m)^{s-1}\langle\theta_{t-s}-\theta^*_{t-s}, \psi_{t-s+1}^{(v)}\rangle - 2\eta\sum_{s=1}^{t-1}(1-\eta m)^{s-1}\langle\theta^*_{t-s}-\theta^*_{t-s+1}, \psi_{t-s+1}^{(v)}\rangle$$

$$+ 4\eta^2\lambda^2\sum_{s=1}^{t-1}(1-\eta m)^{s-1}\mathbf{1}_{C_{t-s+1}\neq 0} + 4\eta\lambda\mathcal{D}\sum_{s=1}^{t-1}(1-\eta m)^{s-1}\mathbf{1}_{C_{t-s+1}\neq 0}. \qquad (48)$$

**Lemma 20.5.** *If* $\lambda \geq 2\sup_{\theta\in\Theta}\|\nabla\mathcal{R}_t(\theta)\| = 2G$, *the following inequalities hold.*

$$\|\psi_t^{(v)}\| \leq 2\lambda \qquad (49)$$

$$\|\psi_t^{(b)}\|_2 \leq \frac{4\sigma^2}{\lambda} \qquad (50)$$

$$\mathbb{E}_{Z_t}[\|\psi_t^{(v)}\|_2^2|\mathcal{F}_{t-1}] \leq 10\sigma^2 \qquad (51)$$

Simplifying Equation (48) using bounds in Lemma 20.5, we get

$$\|\theta^*_t - \theta_t\|_2^2 \leq (1-\eta m)^{t-1}\|\theta_1-\theta^*_1\|_2^2 + \sum_{s=1}^{t-1}(1-\eta m)^s\|\theta^*_{t-s}-\theta^*_{t-s+1}\|_2^2$$

$$+ \frac{16\eta^2\sigma^2}{\lambda}\sum_{s=1}^{t-1}(1-\eta m)^{s-1} + 40\eta^2\sigma^2\sum_{s=1}^{t-1}(1-\eta m)^{s-1}$$

$$+ 4\eta^2\sum_{s=1}^{t-1}(1-\eta m)^{s-1}(\|\psi_{t-s+1}^{(v)}\|_2^2 - \mathbb{E}_{Z_{t-s+1}}[\|\psi_{t-s+1}^{(v)}\|_2^2|\mathcal{F}_{t-s+1}])$$

$$+ 2\eta\sum_{s=1}^{t-1}(1-\eta m)^{s-1}\|\theta_{t-s} - \theta_{t-s}^*\|\|\psi_{t-s+1}^{(b)}\| + 2\eta\sum_{s=1}^{t-1}(1-\eta m)^{s-1}\|\theta_{t-s}^* - \theta_{t-s+1}^*\|\|\psi_{t-s+1}^{(b)}\|$$

$$- 2\eta\sum_{s=1}^{t-1}(1-\eta m)^{s-1}\langle\theta_{t-s} - \theta_{t-s}^*, \psi_{t-s+1}^{(v)}\rangle - 2\eta\sum_{s=1}^{t-1}(1-\eta m)^{s-1}\langle\theta_{t-s}^* - \theta_{t-s+1}^*, \psi_{t-s+1}^{(v)}\rangle$$

$$+ 4\eta^2\lambda^2\sum_{s=1}^{t-1}(1-\eta m)^{s-1}\mathbf{1}_{C_{t-s+1}\neq 0} + 4\eta\lambda\mathcal{D}\sum_{s=1}^{t-1}(1-\eta m)^{s-1}\mathbf{1}_{C_{t-s+1}\neq 0}. \tag{52}$$

Further applying the bound that $\|\psi_t^{(b)}\| \leq \frac{4\sigma^2}{\lambda}$

$$\|\theta_t^* - \theta_t\|_2^2 \leq (1-\eta m)^{t-1}\|\theta_1 - \theta_1^*\|_2^2 + \sum_{s=1}^{t-1}(1-\eta m)^s\|\theta_{t-s}^* - \theta_{t-s+1}^*\|_2^2$$

$$+ \frac{16\eta^2\sigma^2}{\lambda}\sum_{s=1}^{t-1}(1-\eta m)^{s-1} + 40\eta^2\sigma^2\sum_{s=1}^{t-1}(1-\eta m)^{s-1}$$

$$+ 4\eta^2\sum_{s=1}^{t-1}(1-\eta m)^{s-1}(\|\psi_{t-s+1}^{(v)}\|_2^2 - \mathbb{E}_{Z_{t-s+1}}[\|\psi_{t-s+1}^{(v)}\|_2^2|\mathcal{F}_{t-s}])$$

$$+ \frac{8\sigma^2\eta}{\lambda}\sum_{s=1}^{t-1}(1-\eta m)^{s-1}\|\theta_{t-s} - \theta_{t-s}^*\| + \frac{8\sigma^2\eta}{\lambda}\sum_{s=1}^{t-1}(1-\eta m)^{s-1}\|\theta_{t-s}^* - \theta_{t-s+1}^*\|$$

$$- 2\eta\sum_{s=1}^{t-1}(1-\eta m)^{s-1}\langle\theta_{t-s} - \theta_{t-s}^*, \psi_{t-s+1}^{(v)}\rangle \underbrace{- 2\eta\sum_{s=1}^{t-1}(1-\eta m)^{s-1}\langle\theta_{t-s}^* - \theta_{t-s+1}^*, \psi_{t-s+1}^{(v)}\rangle}$$

$$+ 4\eta^2\lambda^2\sum_{s=1}^{t-1}(1-\eta m)^{s-1}\mathbf{1}_{C_{t-s+1}\neq 0} + 4\eta\lambda\mathcal{D}\sum_{s=1}^{t-1}(1-\eta m)^{s-1}\mathbf{1}_{C_{t-s+1}\neq 0}. \tag{53}$$

We further simplify the underscored term above as

$$-2\eta\sum_{s=1}^{t-1}(1-\eta m)^{s-1}\langle\theta_{t-s}^* - \theta_{t-s+1}^*, \psi_{t-s+1}^{(v)}\rangle \leq 2\eta\sum_{s=1}^{t-1}(1-\eta m)^{s-1}\|\theta_{t-s}^* - \theta_{t-s+1}^*\|\|\psi_t^{(v)}\|,$$

$$\overset{49}{\leq} 4\eta\lambda\sum_{s=1}^{t-1}(1-\eta m)^{s-1}\|\theta_{t-s}^* - \theta_{t-s+1}^*\|.$$

Using the inequality above into Equation (53) along with the bound that $4\eta^2\lambda^2\sum_{s=1}^{t-1}(1-\eta m)^{s-1}\mathbf{1}_{C_{t-s+1}\neq 0} \leq \frac{4\eta\lambda^2\Lambda_T}{m}$, where $\Lambda_T := \sum_{t=1}^T \mathbf{1}_{C_t\neq 0}$, we get

$$\|\theta_t^* - \theta_t\|_2^2 \leq (1-\eta m)^{t-1}\|\theta_1 - \theta_1^*\|_2^2 + \sum_{s=1}^{t-1}(1-\eta m)^s\|\theta_{t-s}^* - \theta_{t-s+1}^*\|_2^2$$

$$+ \frac{16\eta\sigma^2}{\lambda m} + \frac{40\eta\sigma^2}{m} + \underbrace{4\eta^2\sum_{s=1}^{t-1}(1-\eta m)^{s-1}(\|\psi_{t-s+1}^{(v)}\|_2^2 - \mathbb{E}_{Z_{t-s+1}}[\|\psi_{t-s+1}^{(v)}\|_2^2|\mathcal{F}_{t-s}])}_{\text{Term I}}$$

$$+ \frac{8\sigma^2\eta}{\lambda}\sum_{s=1}^{t-1}(1-\eta m)^{s-1}\|\theta_{t-s} - \theta_{t-s}^*\| + \frac{8\sigma^2\eta}{\lambda}\sum_{s=1}^{t-1}(1-\eta m)^{s-1}\|\theta_{t-s}^* - \theta_{t-s+1}^*\|$$

$$\underbrace{-2\eta \sum_{s=1}^{t-1}(1-\eta m)^{s-1}\langle \theta_{t-s} - \theta^*_{t-s}, \psi^{(v)}_{t-s+1}\rangle}_{\text{Term II}} + 4\eta\lambda \sum_{s=1}^{t-1}(1-\eta m)^{s-1}\|\theta^*_{t-s} - \theta^*_{t-s+1}\|$$

$$+ 4\eta^2\lambda^2 \sum_{s=1}^{t-1}(1-\eta m)^{s-1}\mathbf{1}_{C_{t-s+1}\neq 0} + 4\eta\lambda\mathcal{D}\sum_{s=1}^{t-1}(1-\eta m)^{s-1}\mathbf{1}_{C_{t-s+1}\neq 0}. \tag{54}$$

Denote by the event $\mathcal{E}^{(1)}$ as

$$\mathcal{E}^{(1)} = \bigcap_{t=1}^{T}\left\{ 4\eta^2 \sum_{s=1}^{t-1}(1-\eta m)^{s-1}(\|\psi^{(v)}_{t-s+1}\|_2^2 - \mathbb{E}_{Z_{t-s+1}}[\|\psi^{(v)}_{t-s+1}\|_2^2|\mathcal{F}_{t-s}]) \leq \right.$$

$$\left. 32\eta^2\lambda^2 \ln\left(\frac{2T}{\delta}\right) + \frac{32\eta^{3/2}\lambda\sigma\sqrt{\ln\left(\frac{2T}{\delta}\right)}}{\sqrt{2m}}\right\}$$

**Lemma 20.6.**

$$\mathbb{P}[\mathcal{E}^{(1)}] \geq 1 - \delta.$$

For every $t \in \{1, \cdots, T\}$, denote by the constant

$$U_t = \left[(1-\eta m)^{t-1}\|\theta_1 - \theta^*_1\|_2^2 + \sum_{s=1}^{t-1}(1-\eta m)^s\|\theta^*_{t-s} - \theta^*_{t-s+1}\|_2^2\right.$$

$$+ \frac{16\eta\sigma^2}{\lambda m} + \frac{40\eta\sigma^2}{m} + 32\eta^2\lambda^2 \ln\left(\frac{2T}{\delta}\right) + \frac{32\eta^{3/2}\lambda\sigma\sqrt{\ln\left(\frac{2T}{\delta}\right)}}{\sqrt{2m}} + \frac{8\sigma^2\eta}{\lambda}\sum_{s=1}^{t-1}(1-\eta m)^{s-1}\|\theta^*_{t-s} - \theta^*_{t-s+1}\|$$

$$+ 4\eta\lambda\sum_{s=1}^{t-1}(1-\eta m)^{s-1}\|\theta^*_{t-s} - \theta^*_{t-s+1}\| + 4\eta^2\lambda^2\sum_{s=1}^{t-1}(1-\eta m)^{s-1}\mathbf{1}_{C_{t-s+1}\neq 0}$$

$$\left. + 4\eta\lambda\mathcal{D}\sum_{s=1}^{t-1}(1-\eta m)^{s-1}\mathbf{1}_{C_{t-s+1}\neq 0}\right]. \tag{55}$$

**Proposition 20.7.** *For every* $1 \leq s < t \leq T$,

$$(1-\eta m)^{s-1}U_{t-s} \leq U_t. \tag{56}$$

For each $t \in \{1, \cdots, T\}$, denote by the event $\mathcal{E}^{(2)}_t$ as

$$\mathcal{E}^{(2)}_t = \left\{-2\eta\sum_{s=1}^{t-1}(1-\eta m)^{s-1}\langle\theta_{t-s} - \theta^*_{t-s}, \psi^{(v)}_{t-s+1}\rangle \leq 4\eta\lambda\sqrt{QU_t}\left(1 + \sqrt{1 + \frac{2}{\eta m}}\right)\right\}$$

Observe from definition, $\mathbb{P}[\mathcal{E}^{(2)}_1] = 1$. In order to bound the other events, the following lemma holds.

**Lemma 20.8.** *For each* $t \in \{2, \cdots, T\}$

$$\mathbb{P}[\mathcal{E}^{(2)}_t|\mathcal{E}^{(2)}_{t-1}, \cdots, \mathcal{E}^{(2)}_1] \geq 1 - \frac{\delta}{T}.$$

Denote by the event

$$\mathcal{E}^{(2)} := \bigcap_{t=1}^{T}\mathcal{E}^{(2)}_t. \tag{57}$$

**Corollary 20.9.**

$$\mathbb{P}[\mathcal{E}^{(2)}] \geq 1 - \delta,$$

*where event $\mathcal{E}^{(2)}$ is given in Equation (57).*

**Lemma 20.10.** *Under events $\mathcal{E}^{(1)}$ and $\mathcal{E}^{(2)}$, for all times $t \in \{1, \cdots, T\}$,*

$$l_t^2 \leq QU_t,$$

*where $Q = \frac{206\sigma^2}{\lambda^2 m\eta} + \frac{4\lambda^2}{\sigma^2} + 2$.*

We prove this lemma by induction. The base case of $t = 1$ holds trivially by definition.

By the induction hypothesis, assume that for some $t > 1$, $l_s^2 \leq QU_s$ holds for all $s < t$. Then, from Equation (54) we have that

$$l_t^2 \leq U_t + \frac{8\sigma^2\eta}{\lambda} \sum_{s=1}^{t-1} (1 - \eta m)^{s-1} \|\theta_{t-s} - \theta_{t-s}^*\| + 4\eta\lambda\sqrt{QU_t} \left(1 + \sqrt{1 + \frac{2}{\eta m}}\right),$$

$$\overset{(a)}{\leq} U_t + \frac{8\sigma^2\eta}{\lambda} \sum_{s=1}^{t-1} (1 - \eta m)^{\frac{s-1}{2}} \sqrt{(1 - \eta m)^s QU_{t-s}} + 4\eta\lambda\sqrt{QU_t} \left(1 + \sqrt{1 + \frac{2}{\eta m}}\right),$$

$$\overset{(56)}{\leq} U_t + \frac{8\sigma^2\eta}{\lambda} \sqrt{QU_t} \sum_{s=1}^{t-1} (1 - \eta m)^{\frac{s-1}{2}} + 4\eta\lambda\sqrt{QU_t} \left(1 + \sqrt{1 + \frac{2}{\eta m}}\right),$$

$$\leq U_t + \frac{64\sigma^2}{m\lambda} \sqrt{QU_t} + 4\eta\lambda\sqrt{QU_t} \left(1 + \sqrt{1 + \frac{2}{\eta m}}\right),$$

$$\overset{(b)}{\leq} U_t + \frac{64\sigma^2}{m\lambda} \sqrt{QU_t} + 16\lambda\sqrt{\frac{\eta QU_t}{m}}$$

Step $(a)$ follows from the induction hypothesis. Step $(b)$ follows since $\frac{2}{\eta m} > 1$.

**Claim** If $Q \geq \left(\frac{206\sigma^2}{\lambda^2 m\eta} + \frac{4\lambda^2}{\sigma^2}\right) + 2$, then $U_t + \frac{64\sigma^2}{m\lambda}\sqrt{QU_t} + 16\lambda\sqrt{\frac{\eta QU_t}{m}} \leq QU_t$.

We prove this claim by contradiction. Assume that $Q \geq \left(\frac{206\sigma^2}{\lambda^2 m\eta} + \frac{4\lambda^2}{\sigma^2}\right) + 2$ and that $U_t + \frac{64\sigma^2}{m\lambda}\sqrt{QU_t} + 16\lambda\sqrt{\frac{\eta QU_t}{m}} > QU_t$. Re-arranging, we see that

$$\sqrt{QU_t} \left(\frac{64\sigma^2}{m\lambda} + 16\lambda\sqrt{\frac{\eta}{m}}\right) > (Q - 1)U_t,$$

$$\sqrt{Q} \left(\frac{64\sigma^2}{m\lambda} + 16\lambda\sqrt{\frac{\eta}{m}}\right) > (Q - 1)\sqrt{U_t},$$

$$\sqrt{Q} \left(\frac{64\sigma^2}{m\lambda} + 16\lambda\sqrt{\frac{\eta}{m}}\right) > (Q - 1)\sqrt{\frac{40\eta\sigma^2}{m}}.,$$

The last inequality follows from Equation (55), where $U_t \geq \frac{40\eta\sigma^2}{m}$. Re-aranging the last inequality, we see that

$$\left(\frac{64\sigma}{\lambda\sqrt{40m\eta}} + \frac{16\lambda}{\sqrt{40\sigma}}\right)\sqrt{Q} > (Q - 1).$$

It is easy to verify that the above inequality and the fact that $Q = \left(\frac{206\sigma^2}{\lambda^2 m\eta} + \frac{4\lambda^2}{\sigma^2}\right) + 2$ cannot hold simultaneously. This can be checked by squaring both sides and solving for the root of the quadratic equation in $Q$.

## 20.2 Proof of Theorem 20.1

We know from Lemma 20.6 and Corollary 20.9, that with probability at-least $1 - 2\delta$, events $\mathcal{E}^{(1)} \cap \mathcal{E}^{(2)}$ holds. Lemma 20.10 gives that for all $t \in \{1, \cdots, T\}$, $\|\theta_t - \theta_t^*\|^2 \le QU_t$, where $Q$ is given in Lemma 20.10 and $U_t$ is given in Equation (55). Now applying the inequality that $\sqrt{x + y} \le \sqrt{x} + \sqrt{y}$, we get that

$$\|\theta_t - \theta_t^*\| \le \left( \frac{15\sigma}{\lambda\sqrt{m\eta}} + \frac{2\lambda}{\sigma} + 2 \right) \left[ (1 - \eta m)^{\frac{t-1}{2}} \|\theta_1 - \theta_1^*\| + \sum_{s=1}^{t-1} (1 - \eta m)^{\frac{s}{2}} \|\theta_{t-s}^* - \theta_{t-s+1}^*\| \right.$$

$$+ \frac{4\sqrt{\eta}\sigma}{\sqrt{\lambda m}} + \frac{7\sigma\sqrt{\eta}}{\sqrt{m}} + 6\eta\lambda \sqrt{\ln\left(\frac{2T}{\delta}\right)} + \frac{6\eta^{3/4}\sqrt{\lambda\sigma\sqrt{\ln\left(\frac{2T}{\delta}\right)}}}{2m^{1/4}} + \frac{3\sigma\sqrt{\eta}}{\sqrt{\lambda}} \sum_{s=1}^{t-1} (1-\eta m)^{\frac{s-1}{2}} \sqrt{\|\theta_{t-s}^* - \theta_{t-s+1}^*\|}$$

$$+ 2\sqrt{\eta\lambda} \sum_{s=1}^{t-1} (1 - \eta m)^{\frac{s-1}{2}} \sqrt{\|\theta_{t-s}^* - \theta_{t-s+1}^*\|} + 2\eta\lambda \sum_{s=1}^{t-1} (1 - \eta m)^{\frac{s-1}{2}} \mathbf{1}_{C_{t-s+1} \ne 0}$$

$$\left. + 2\sqrt{\eta\lambda\mathcal{D}} \sum_{s=1}^{t-1} (1 - \eta m)^{\frac{s-1}{2}} \mathbf{1}_{C_{t-s+1} \ne 0} \right]$$

Recalling that $\mathrm{Reg}_T := \sum_{t=1}^T \|\theta_t - \theta_t^*\|$, we have by simplifying using the series expansions in Equations (34, 35, 36, 37), we get that

$$\mathrm{Reg}_T \le \left( \frac{15\sigma}{\lambda\sqrt{m\eta}} + \frac{2\lambda}{\sigma} + 2 \right) \left[ \frac{2(l_1 + \Phi_T)}{\eta m} + 4\sigma T \sqrt{\frac{\eta}{\lambda m}} + 7\sigma T \frac{\eta}{m} + 6\eta\lambda T \sqrt{\ln\left(\frac{2T}{\delta}\right)} \right.$$

$$\left. + \frac{6T\eta^{3/4}\sqrt{\lambda\sigma\sqrt{\ln\left(\frac{2T}{\delta}\right)}}}{2m^{1/4}} + 3\sigma\sqrt{\frac{T\Phi_T}{\eta\lambda m^2}} + 2\sqrt{\frac{\lambda T\Phi_T}{\eta}} + \frac{4\lambda\Lambda_T}{\eta m} + 4\Lambda_T \sqrt{\frac{\lambda\mathcal{D}}{\eta m^2}} \right].$$

## 20.3 Proof of Lemma 20.8

*Proof.* Fix a time $t \in \{1, \cdots, T\}$. We wish to bound the sum $\sum_{s=1}^{t-1} 2\eta(1 - \eta m)^{s-1} \langle \theta_{t-s}^* - \theta_{t-s}, \psi_{t-s+1}^{(v)} \rangle$. In order to do so, we define the following sequence of random vectors

Denote by the sequence $(\xi_s)_{s=1}^T$ of random vectors as

$$\xi_s = \begin{cases} \theta_s^* - \theta_s & \text{if } \|\theta_s^* - \theta_s\|^2 \le QU_s, \\ 0 & \text{otherwise} \end{cases} \tag{58}$$

where $Q$ is defined in Lemma 20.10. From the induction hypothesis, on the event $\mathcal{E}_{t-1}^{(2)}, \cdots, \mathcal{E}_1^{(2)}$, we have

$$\sum_{s=1}^{t-1} 2\eta(1 - \eta m)^{s-1} \langle \theta_{t-s}^* - \theta_{t-s}, \psi_{t-s+1}^{(v)} \rangle = \sum_{s=1}^{t-1} 2\eta(1 - \eta m)^{s-1} \langle \xi_{t-s}, \psi_{t-s+1}^{(v)} \rangle,$$

holding almost-surely. We will now bound the sum $\sum_{s=1}^{t-1} 2\eta(1 - \eta m)^{s-1} \langle \xi_{t-s}, \psi_{t-s+1}^{(v)} \rangle$ using Freedman's martinglae inequality. We know that,

$$|2\eta(1 - \eta m)^{s-1} \langle \xi_{t-s}, \psi_{t-s+1}^{(v)} \rangle| \le 2\eta(1 - \eta m)^{s-1} \|xi_{t-s}\| \|\psi_{t-s+1}^{(v)}\|,$$
$$\le 4\eta\lambda\sqrt{QU_t}.$$

Further, the sum of conditional variances can be bounded as

$$\sum_{s=1}^{t-1} |2\eta(1 - \eta m)^{s-1} \langle \xi_{t-s}, \psi_{t-s+1}^{(v)} \rangle|^2 \le \sum_{s=1}^{t-1} |16\lambda^2\eta^2(1 - \eta m)^{s-1} QU_t,$$

$$\leq \frac{16\lambda^2\eta QU_t}{m}.$$

Applying Freedman's inequality, we see that

$$\sum_{s=1}^{t-1} 2\eta(1-\eta m)^{s-1}\langle\theta_{t-s}^* - \theta_{t-s}, \psi_{t-s+1}^{(v)}\rangle \leq 4\eta\lambda\sqrt{QU_t} + \sqrt{16\eta^2\lambda^2 QU_t \ln^2(T/\delta) + \frac{32\lambda^2\eta QU_t}{m}\ln(T/\delta)},$$

$$\leq 4\eta\lambda\sqrt{QU_t}\left(1 + \sqrt{1 + \frac{2}{\eta m}}\right).$$

$\square$

## 20.4 Useful Martingale concentration inequality

**Lemma 20.11** (Freedman's inequality[59]). *Suppose $Y_1, \cdots, Y_T$ is a bounded martingale with respect to a filtration $(\mathcal{F}_t)_{t=0}^T$ with $\mathbb{E}[Y_t|\mathcal{F}_{t-1}] = 0$ and $\mathbb{P}[|Y_t| \leq B] = 1$ for all $t \in \{1, \cdots, T\}$. Denote by $V_s := \sum_{n=1}^s Var(Y_n|\mathcal{F}_{n-1})$ be the sum of conditional variances. Then, for every $a, v > 0$,*

$$\mathbb{P}\left(\exists n \in [1, T] \text{ such that } \sum_{t=1}^n Y_t \geq a \text{ and } V_n \leq v\right) \leq \exp\left(\frac{-a^2}{2(v + Ba)}\right). \tag{59}$$

Re-arranging the above inequality, we see that if

$$a \geq B\ln\left(\frac{2T}{\delta}\right) + \sqrt{\left(B\ln\left(\frac{2T}{\delta}\right)\right)^2 + 2v\ln\left(\frac{2T}{\delta}\right)}, \tag{60}$$

then the RHS of Equation (59) is bounded above by $\frac{\delta}{2}$.

## 20.5 Proof of Lemma 20.6

*Proof of Lemma 20.6.* Fix a $t \in \{1, \cdots, T\}$. For $s \in \{1, \cdots, t-1\}$, denote by the random variable $Y_s^{(t)} := 4\eta^2(1-\eta m)^{s-1}(\|\psi_{t-s+1}^{(v)}\|_2^2 - \mathbb{E}_{Z_{t-s+1}}[\|\psi_{t-s+1}^{(v)}\|_2^2|\mathcal{F}_{t-s}])$. Observe that the sequence $(Y_s^{(t)})_{s=1}^{t-1}$ is a martingale difference sequence with respect to the filtration $(\mathcal{G}_s)_{s=1}^{t-1}$, where $\mathcal{G}_s := \mathcal{F}_{t-s}$. Furthermore, Lemma 20.5 gives that with probability 1, $|Y_s^{(t)}| \leq 4\eta^2(4\lambda^2 + 4\lambda^2) \leq 32\eta^2\lambda^2$. We can bound the conditional variance as

$$\sum_{s=1}^{t-1} Var(Y_s^{(t)}|\mathcal{G}_s) \leq 16\eta^4 \sum_{s=1}^{t-1}(1-\eta m)^{2(s-1)}\mathbb{E}_{Z_{t-s}}[(\|\psi_{t-s+1}^{(v)}\|_2^2 - \mathbb{E}_{Z_{t-s+1}}[\|\psi_{t-s+1}^{(v)}\|_2^2|\mathcal{F}_{t-s}])^2|\mathcal{F}_{t-s}],$$

$$\overset{49}{\leq} 16\eta^4 8\lambda^2 \sum_{s=1}^{t-1}(1-\eta m)^{2(s-1)}\mathbb{E}_{Z_{t-s}}[|\|\psi_{t-s+1}^{(v)}\|_2^2 - \mathbb{E}_{Z_{t-s+1}}[\|\psi_{t-s+1}^{(v)}\|_2^2|\mathcal{F}_{t-s}]||\mathcal{F}_{t-s}],$$

$$\leq 128\eta^4\lambda^2 \sum_{s=1}^{t-1}(1-\eta m)^{2(s-1)}(2\mathbb{E}_{Z_{t-s}}[\|\psi_{t-s+1}^{(v)}\|_2^2|\mathcal{F}_{t-s}]),$$

$$\overset{51}{\leq} 2560\eta^4\lambda^2\sigma^2 \sum_{s=1}^{t-1}(1-\eta m)^{2(s-1)},$$

$$\leq \frac{2560\eta^3\lambda^2\sigma^2}{m}.$$

Now, putting $B := 32\eta^2\lambda^2$ and $v = \frac{2560\eta^3\lambda^2\sigma^2}{m}$, we get from Equation (60) that with probability at-least $1 - \delta/2$,

$$\text{Term I} \leq 32\eta^2\lambda^2\ln\left(\frac{2T}{\delta}\right) + \sqrt{\left(32\eta^2\lambda^2\ln\left(\frac{2T}{\delta}\right)\right)^2 + \frac{5120\eta^3\lambda^2\sigma^2}{m}\ln\left(\frac{2T}{\delta}\right)},$$

$$\leq 32\eta^2\lambda^2 \ln\left(\frac{2T}{\delta}\right) + \frac{32\eta^{3/2}\lambda\sigma\sqrt{\ln\left(\frac{2T}{\delta}\right)}}{\sqrt{2m}}.$$

$\square$

## 20.6 Proof of Lemma 20.5 reproduced from [25]

This follows from the analysis of [25] and reproduced here for completeness.

**Proof of Equation (49)**

$$
\begin{aligned}
\|\psi_t^{(v)}\| &= \|\psi_t - \mathbb{E}_{Z_t}[\psi_t|\mathcal{F}_{t-1}]\|, \\
&\overset{(a)}{=} \|\mathrm{clip}(\nabla\mathcal{L}(Z_t,\theta_{t-1}),\lambda) - \mathbb{E}_{Z_t}[\mathrm{clip}(\nabla\mathcal{L}(Z_t,\theta_{t-1}),\lambda)|\mathcal{F}_{t-1}], \\
&\leq \|\mathrm{clip}(\nabla\mathcal{L}(Z_t,\theta_{t-1}),\lambda)\| + \|\mathbb{E}_{Z_t}[\mathrm{clip}(\nabla\mathcal{L}(Z_t,\theta_{t-1}),\lambda)|\mathcal{F}_{t-1}]\|, \\
&\leq \lambda + \lambda
\end{aligned}
$$

Equality $(a)$ follows from the fact that $\theta_{t-1} \in \mathcal{F}_{t-1}$.

**Proof of Equation (50)**

Denote by the event $\xi_t := \mathbf{1}_{\|\nabla\mathcal{L}(Z_t,\theta_{t-1})\geq\lambda}$ and by $\chi_t := \mathbf{1}_{\|\nabla\mathcal{L}(Z_t,\theta_{t-1})-\nabla\mathcal{R}_t(\theta_{t-1})\|>\frac{\lambda}{2}}$. Since we have $\lambda > 2G$, we have the inequality that

$$\xi_t \leq \chi_t \ a.s. \tag{61}$$

Then, we have that

$$\mathrm{clip}\nabla\mathcal{L}(_t,\theta_{t-1})),\lambda) = \nabla\mathcal{L}(Z_t,\theta_{t-1})(1-\xi_t) + \frac{\lambda}{\|\nabla\mathcal{L}(Z_t,\theta_{t-1})\|}\nabla\mathcal{L}(Z_t,\theta_{t-1})\xi_t.$$

Now, expanding out $\psi_t^{(b)}$, we get

$$
\begin{aligned}
\|\psi_t^{(b)}\|_2 &= \|\mathbb{E}_{Z_t}[\psi_t|\mathcal{F}_{t-1}]\|_2, \\
&= \|\mathbb{E}_{Z_t}[\mathrm{clip}(\nabla\mathcal{L}(Z_t,\theta_{t-1}),\lambda) - \mathcal{R}_t(\theta_{t-1})|\mathcal{F}_{t-1}]\|_2, \\
&= \left\|\mathbb{E}_{Z_t}\left[\nabla\mathcal{L}(Z_t,\theta_{t-1}) + \left(\frac{\lambda}{\|\nabla\mathcal{L}(Z_t,\theta_{t-1})\|}-1\right)\nabla\mathcal{L}(Z_t,\theta_{t-1})\xi_t - \mathcal{R}_t(\theta_{t-1})\Big|\mathcal{F}_{t-1}\right]\right\|_2, \\
&\overset{(a)}{=} \left\|\mathbb{E}_{Z_t}\left[\left(\frac{\lambda}{\|\nabla\mathcal{L}(Z_t,\theta_{t-1})\|}-1\right)\nabla\mathcal{L}(Z_t,\theta_{t-1})\xi_t\Big|\mathcal{F}_{t-1}\right]\right\|_2, \\
&\leq \mathbb{E}_{Z_t}\left[\left|\frac{\lambda}{\|\nabla\mathcal{L}(Z_t,\theta_{t-1})\|}-1\right|\|\nabla\mathcal{L}(Z_t,\theta_{t-1})\|\xi_t\Big|\mathcal{F}_{t-1}\right], \\
&\leq \mathbb{E}_{Z_t}\left[1-\left(\frac{\lambda}{\|\nabla\mathcal{L}(Z_t,\theta_{t-1})\|}\right)\|\nabla\mathcal{L}(Z_t,\theta_{t-1})\|\xi_t\Big|\mathcal{F}_{t-1}\right], \\
&\leq \mathbb{E}_{Z_t}[\|\nabla\mathcal{L}(Z_t,\theta_{t-1})\|\xi_t|\mathcal{F}_{t-1}], \\
&\overset{61}{\leq} \mathbb{E}_{Z_t}[\|\nabla\mathcal{L}(Z_t,\theta_{t-1})\|\chi_t|\mathcal{F}_{t-1}], \\
&\leq \mathbb{E}_{Z_t}[\|\nabla\mathcal{L}(Z_t,\theta_{t-1}) - \nabla\mathcal{R}_t(\theta_{t-1})\|\chi_t|\mathcal{F}_{t-1}] + \|\nabla\mathcal{R}_t(\theta_{t-1})\|\mathbb{E}_{Z_t}[\chi_t|\mathcal{F}_{t-1}], \\
&\leq \sqrt{\mathbb{E}_{Z_t}[\|\nabla\mathcal{L}(Z_t,\theta_{t-1}) - \nabla\mathcal{R}_t(\theta_{t-1})\|_2^2|\mathcal{F}_{t-1}]\mathbb{E}_{Z_t}[\chi_t^2|\mathcal{F}_{t-1}]} + \|\nabla\mathcal{R}_t(\theta_{t-1})\|\mathbb{E}_{Z_t}[\chi_t|\mathcal{F}_{t-1}], \\
&\leq \sigma\sqrt{\mathbb{E}_{Z_t}[\chi_t|\mathcal{F}_{t-1}]} + \frac{\lambda}{2}\mathbb{E}_{Z_t}[\chi_t|\mathcal{F}_{t-1}]. \tag{62}
\end{aligned}
$$

Now computing the probability

$$\mathbb{E}_{Z_t}[\chi_t|\mathcal{F}_{t-1}] = \mathbb{P}[\|\nabla\mathcal{L}(Z_t,\theta_{t-1}) - \nabla\mathcal{R}_t(\theta_{t-1})|\mathcal{F}_{t-1}\| \geq \frac{\lambda}{2}],$$

$$= \mathbb{P}[\|\nabla\mathcal{L}(Z_t, \theta_{t-1}) - \nabla\mathcal{R}_t(\theta_{t-1})\|^2 \geq \frac{\lambda^2}{4}|\mathcal{F}_{t-1}],$$

$$\leq \frac{4\sigma^2}{\lambda^2}. \tag{63}$$

The last in-equality is from Chebychev's inequality. Substituting Equation (63) into Equation (62) yields the result.

**Proof of Equation (51)**

$$\mathbb{E}[\|\psi_t^{(v)}\|_2^2|\mathcal{F}_{t-1}] \overset{(a)}{=} \mathbb{E}_{Z_t}[\|\text{clip}(\mathcal{L}(Z_t, \theta_{t-1}), \lambda) - \mathbb{E}_{Z_t}[\text{clip}(\mathcal{L}(Z_t, \theta_{t-1}), \lambda)|\mathcal{F}_{t-1}]\|_2^2|\mathcal{F}_{t-1}],$$

$$\overset{(b)}{\leq} \mathbb{E}[\|\text{clip}(\mathcal{L}(Z_t, \theta_{t-1}), \lambda) - \nabla\mathcal{R}_t(\theta_{t-1})\|_2^2|\mathcal{F}_{t-1}]. \tag{64}$$

Inequality $(a)$ follows from the fact that $\theta_{t-1} \in \mathcal{F}_{t-1}$ and step $(b)$ follows from the fact that for any random vector $Z$ and filtration $\mathcal{F}$ and $Y \in \mathcal{F}$, $\mathbb{E}_Z[\|Z - \mathbb{E}[Z|\mathcal{F}]\|_2^2|\mathcal{F}] \leq \mathbb{E}[\|Z - Y\|_2^2|\mathcal{F}]$ holds almost-surely. To simplify notation, we denote by $\mathcal{E}_t := \mathcal{E}[\cdot|\mathcal{F}_{t-1}]$. We now bound Equation (64) to conclude.

$$\mathbb{E}_t[\|\text{clip}(\mathcal{L}(Z_t, \theta_{t-1}), \lambda) - \nabla\mathcal{R}_t(\theta_{t-1})\|_2^2] = \mathbb{E}_t[\|\nabla\mathcal{L}(Z_t, \theta_{t-1}) - \nabla\mathcal{R}_t(\theta_{t-1})\|_2^2(1 - \xi_t)^2] +$$

$$+ \mathbb{E}_t\left[\left\|\frac{\lambda\nabla\mathcal{L}(Z_t, \theta_{t-1})}{\|\nabla\mathcal{L}(Z_t, \theta_{t-1})\|} - \nabla\mathcal{R}_t(\theta_{t-1})\right\|_2^2 \xi_t\right],$$

$$\overset{(a)}{\leq} \sigma^2 + \mathbb{E}_t\left[\left(2\left\|\frac{\lambda\nabla\mathcal{L}(Z_t, \theta_{t-1})}{\|\nabla\mathcal{L}(Z_t, \theta_{t-1})\|}\right\|_2^2 + 2\|\nabla\mathcal{R}_t(\theta_{t-1})\|_2^2\right)\xi_t^2\right],$$

$$\overset{(b)}{\leq} \sigma^2 + \frac{5}{2}\lambda^2\mathbb{E}_t[\xi_t^2],$$

$$\overset{(c)}{\leq} \sigma^2 + \frac{5}{2}\lambda^2\mathbb{E}_t[\xi_t^2],$$

$$\overset{63}{\leq} \sigma^2 + \frac{5\lambda^2}{2}\cdot\frac{4\sigma^2}{\lambda^2}.$$

Inequality $(a)$ follows from the fact that for any two vectors $a, b$, $\|a + b\|_2^2 \leq 2\|a\|_2^2 + 2\|b\|_2^2$, and the assumption that $\sup_{\theta\in\Theta} \mathbb{E}_t[\|\mathcal{L}(Z, \theta) - \mathcal{R}_t(\theta)\|_2^2] \leq \sigma^2$.

## 20.7 Proof of Corollary 5.4

The finite diameter that $\mathcal{D} < \infty$ fact is only used in Equation (46) which holds even if $\mathcal{D} = \infty$, albeit the bound is vacuous if $\Lambda_T > 0$. Thus, the entire proof holds verbatim even if $\mathcal{D} = \infty$ and gives non-trivial regret guarantees when $\Lambda_T = 0$.

