# OpenReview forum: "Online robust non-stationary estimation"
_NeurIPS.cc/2023/Conference — NeurIPS 2023 poster_

### Official Review · Reviewer_MGHA · 2023-07-05

**Soundness:** 3 good
**Presentation:** 3 good
**Contribution:** 3 good
**Rating:** 7
**Confidence:** 3

**Summary:**

This paper considers an online estimation setting where the learner observes a sequence of samples, which are drawn from a (previously determined but unknown) sequence of probability distributions, and with some fraction of samples arbitrarily corrupted. In each round, the learner makes a decision, and regret is measured based on how far this decision is from that which would minimize their expected loss if no corruption occurred. The loss function is strongly convex, and the goal is to obtain total regret bounds which are sublinear in the time horizon $T$ (but may scale with some standard parameters like diameter of the decision space, (unknown) amount of distribution shift, (unknown) amount of corruptions, (known) gradient norm upper bound, (unknown) gradient covariance upper bound). The authors show that a tuned version of clipped SGD achieves the desired regret bounds, with some partial lower bounds. The proofs use combine a novel inductive argument with martingale concentration techniques to provide high probability regret bounds under arbitrary distribution shift, and the results are verified with some simple experiments pertaining to mean estimation and linear regression.

**Strengths:**

- To the best of my knowledge, and as claimed by the authors, no work has addressed this issue of online, (outlier/heavy-tail) robust, and non-stationary convex optimization, and this setting seems important for many applications.
- There are several settings where their regret bounds are tight, and many improvements over the state-of-the-art
- The proposed algorithm is quite simple, and they empirically validate the choice of tuning suggested by their theory
- The paper is generally well-written, with thorough explanations of the problem setup, prior work, and the different components of their regret bounds (though I struggled a bit with understanding the analysis of their general regret bound)

**Weaknesses:**

- It would seem more natural to measure regret via the excess risk
$\sum_{t=1}^T \mathbb{E}\_{Z \sim P_t}[\mathcal{L}(Z,\theta_t)] - \inf_{\theta \in \Theta}\mathbb{E}_{Z \sim P_t}[\mathcal{L}(Z,\theta)].$
Do your results transfer to this metric? If not, why? Cor. 4.6-style results for the stationary setting without corruptions are usually stated with respect to this benchmark, as far as I am aware.
- In the stationary setting with corruptions, it is not clear to me that the diameter-dependence is necessary (and this is considered quite undesirable in the robust mean estimation literature). The authors' lower bound in Section 16 looks to require non-stationarity. Can a lower bound be provided without this requirement?
- There is a claim of dimension independence (modulo the diameter dependence), but I view dependence on the trace of $\Sigma$ as implicitly depending on the dimension in many cases of interest, so this seems a bit misleading
- It is rather unclear to me whether these results are fundamentally interesting, or whether the solution is somewhat standard and only new because this combination of problem settings hasn't been explicitly studied before. In particular, clipped SGD is a standard solution in this space, though their tuning analysis appears novel.
	- If the analysis for the general case is of fundamental interest, I suggest that more space is spent in the appendix (or added page of a final version) describing the induction details - I found the appendix a bit hard to follow and did not verify correctness

Minor Nits:
- On line 48, the superscript for the footnote looks like a power. Right after that, the way the loss function is introduced read to me as if it was already used.
- In the abstract and elsewhere, "high-probability" is used as an adjective without an accompanying noun (presumable "regret bounds").
- Footnote 4 appears before its reference in Table 1
- "an" -> "a" on line 202
- "at-least" -> "at least" in Theorem 4.3
- Notation "m" for strong convexity constant conflicts with power of $T$ in Eq. 1

**Questions:**

In addition to the questions raised in the "weaknesses" section, I was curious if this approach could be adapted to incorporate differential privacy (since gradient clipping + noise is a common approach to private learning).

**Limitations:**

Assumptions are made clear, and I don't anticipate any negative impacts.

---

> ### Author Rebuttal · Authors · 2023-08-08
>
> **Excess risk regret is a direct corollary of our result:**
> As the loss function $\mathcal{L}$ is M smooth (Assumption 1 in our draft), we have that $\mathbb{E}[\mathcal{L}(Z,\theta_t​)−\mathcal{L}(Z,\theta^*_t​)]\leq M \|| \theta_t​− \theta_t^* \||^2$. Thus, a regret bound on the norm  $\||\theta_t​− \theta_t^* \||^2$ translates to a regret bound as measured through the excess risk. We thank the reviewer for this pointer and will add this fact as a corollary.
>
> **Finite diameter assumption is necessary in the presence of corruptions, even if there is no distribution shift**
>
> ***Proof Sketch*** Similar to Prop 2.6, consider two scenarios for mean-estimation. In one scenario,  the un-corrupted samples are all drawn from a Dirac mass at $0$, but the first $\Lambda_T$ samples are corrupted with all $d$ coordinates set to $\mathcal{D}/\sqrt{d}$. In the other scenario, there are no corruptions and the un-corrupted samples are all from Dirac mass at location with all coordinates  $\mathcal{D}/\sqrt{d}$. In both situations, the first $\Lambda_T$ samples are identical. Thus, no estimator for the first $\Lambda_T$ samples can distinguish between these two scenarios and will incur regret at-least $\Omega(\Lambda_T\mathcal{D})$.
>
> We will add a proposition in the revision showing the necessity of finite diameter in the presence of corruptions, even if there is no distribution shift. This does not contradict  [18] since in their model, corruptions occur at random instants while corruptions occur at arbitrary instants in ours.
>
> *In summary,*  (i) finite diameter is necessary in the presence of corruptions, whether or not there is distribution shift, and (ii) an infinite diameter can be handled in the absence of corruptions even if there is distribution shift. See also the attached pdf.
>
> **Regret depending on $\text{Trace}(\Sigma)$ trace of the covariance matrix is a classical ``definition” of dimension free regret in statistics literature:**
> We follow standard terminology that states that a bound is dimension free if it depends on the trace of the co-variance matrix and not on dimension times the maximum eigen-value of the covariance matrix, (c.f. [35, 36] of our attached draft). To be concrete, for mean-estimation of a $d$ dimensional vector, a regret bound that depends on $\text{Trace}(\Sigma)$ is deemed to be “dimension-free”  (c.f. [35, 36] ). On the other hand, a  bound that depends on $d\nu_{max}​(\Sigma)$ where $\nu_{max}(\Sigma)$ is the highest eigen-value of the co-variance matrix is NOT dimension free. A bound on the trace is more favorable in high-dimensional settings (cf [35,36]) since by definition we always have $d\nu_{max}​(\Sigma) \geq \text{Trace}(\Sigma)$. We will add this definition and discussion in the revision stating that our bounds only depend on $\text{Trace}(\Sigma)$ and not on $d\nu_{max}​(\Sigma)$  thereby making our results dimension-free.
>
> **Our setting and insights are conceptually new and interesting! Our work is the first to extend online robust estimation beyond iid/stationary assumption:** We strongly believe our setting and results are interesting (and new)! To the best of our knowledge this is the first work to understand how to estimate in a streaming setting when there are heavy-tails, distribution shifts, high-dimensional observations and adversarial corruptions. As we mention in our introduction, there is a plethora of work in the applied literature where  heuristics are proposed for dealing with streaming settings with all these characteristics. Our work is the first to formalize the question, set benchmarks and desiderata and present an analysis of an achievable algorithm and lower bounds.
>
> From a technical perspective, there are new insights this work provides. For example, we show in Proposition 2.6 (and in this response) that finite diameter is now a necessary in the presence corruptions.
> The proofs  are non-trivial and are *NOT* a corollary of existing results. Analyzing gradient based methods for heavy-tailed *stationary settings without corruptions* is itself an actively emerging field of literature (see [24, 34, 43, 52, 57, 60]). Our work extends this line of work by providing the first analysis in the presence of distribution shifts and corruptions, that is based on different martingale concentration arguments combined with novel induction arguments.
>
> Thus, we respectfully disagree on the claim that “*the work is not interesting since it only combines problem settings not studied before*”. A key surprising result is that a simple/practical algorithm such as clipped SGD with the right tuning is able to be robust to drifts, corruptions, heavy tails and lead to dimension free results. Our work provides insight that the learning rate should straddle the $O(1)$ known to be optimal in the absence of noise to be adaptive to distribution shift and the $O(1/t)$ known to be optimal in the stationary setting when there is no drifts. Thus, we also respectfully disagree with the claim that “*clipped SGD is a standard/known solution for these problems*”.
>
> ***That said however***, this is a first work in this space and our paper leaves several fundamental open questions as we list in the conclusions which are exciting avenues of future work.  We will also take up the reviewers suggestion and add more discussion (for example highlight Lemma 19.9 and 20.10) in the additional page that will be available for the camera ready.
>
> **Writing errors and corrections:**
> We thank the reviewer for a thorough and careful read! We will make these corrections and other writing fixes in the revision.
>
> **Connections with privacy:** Unfortunately, we don't have anything interesting to say. Aggarwal et.al. consider privacy implications in the stationary non-stochastic setting. Studying the price of privacy in a non-stationary setup with drifts , heavy-tails and corruptions is exciting future work.
>
> *Aggarwal et.al. The Price of Differential Privacy for Online Learning, ICML 2017*

---

> > ### Comment · Reviewer_MGHA · 2023-08-11
> >
> > Thanks for your clarifications. Your included lower bound without distribution shift was very helpful for my understanding. It's pretty interesting that there is such a contrast between randomly vs adversarially placed corruptions.
> >
> > I am considering increasing my score, but I want to spend more time understanding some of the other reviewer's concerns.

---

> > > ### Comment · Reviewer_MGHA · 2023-08-15
> > >
> > > I have decided to increase my score to a 7.
> > >
> > > By the way, my confusion about "dimension-free" was that with robust mean estimation, this term is usually reserved for dependence on $\lambda_{\mathrm{max}}(\Sigma)$ instead of $tr(\Sigma)$, since the latter often scales with dimension in standard settings. But, especially given your lower bound, I think your usage is fine, as long as you include your discussion above in a remark.

---

> > > > ### Author Response · Authors · 2023-08-15
> > > >
> > > > Dear reviewer,
> > > >
> > > >  Thank you for your detailed comments and thorough read of the paper leading to insightful questions! We will include these discussions, the lower bounds and precisely define "dimension free" and clarify its difference from the context of robust mean estimation literature.

---

### Official Review · Reviewer_6VQW · 2023-07-07

**Soundness:** 3 good
**Presentation:** 1 poor
**Contribution:** 2 fair
**Rating:** 4
**Confidence:** 3

**Summary:**

This works studies robust sequential estimation under a non-stationary environment. A loss function is fixed in advance. The data generating process is non-stationary over time, and hence the optimal parameter \theta^*_t, which minimizes the expected loss over the distribution at time t, is changing over time. A policy returns in each round an estimated parameter \theta_t. The goal is to minimize the regret, defined as sum of differences between \theta^*_t and \theta. The central question is, is there a policy that is free of distributional knowledge (i.e., moments of the data-generating distributions or stream complexity). They answered this question by presenting a gradient-based algo with sublinear regret. These upper bounds match the known lower bounds in the no-noise and no-drift setting.

**Strengths:**

Presented sublinear regret bounds which matches known lower bounds for the no-drift or no-corruption setting.

**Weaknesses:**

1. I am not able to find significant novelty in either the problem formulation or the results. Maybe there is a good practical reason to consider this particular formulation but at least in this submission, the authors did not sell it well.

2. Writing: In general this paper is not written well.
-	There are plenty of typos and gramatic mistakes, even in the abstract and formulation section. E.g. “A observation … “ in the abstract. Line 133 “formalize”. E.g. “upto” -> up to. Line 99, “to derive high-probability under any rate..”
-	Consistency: do not use a concept before defining it. E.g. in the abstract, “neither the O(1/t)....” What is "t"? It seems lower case t is not the same as "T"
-	Vague language. Just to name a few, in the Abstract: “A observation … can be used” – I don’t understand this line. In the def of regret, what norm are we using? Line 96, “the data stream is subgaussian”, do you mean the distribution in each round is subgaussian, or the the entire stream?
-	Do not start a sentence with a mathematical notation; see e.g. “X_t is shown as …”
-	The tone switches between being very informal to very formal abruptly (and why use the word “diserderata” so frequently?)

**The above issues combined suggest that the submission has been written in a rush.** I suggest the authors carefully polish the paper.

3. Lacking of discussion to previous work. It seems that this problem can be reduced to the problem studied in “non-stationary stochastic optimization” (Oper. Res. ‘14) by Besbes et al. Both papers proposed gradient-based algo and used the total-variance distance (“\Phi_T”) to measure the non-stationarity.  I am wondering what results do previous results in OCO imply, and how the results

**Questions:**

It seems that this problem can be reduced to the problem studied in “non-stationary stochastic optimization” (Oper. Res. ‘14) by Besbes et al. Does this work already imply sublinear regret for some special case of this work? I understand in this submission the corruption is assumed to be adversarial, but this does not seem to be an essential consideration.

---

> ### Author Rebuttal · Authors · 2023-08-08
>
> **Fundamental improvements in the problem setting and results compared to Besbes et.al. :**
>
> We thank the reviewer for pointing out missing a reference and comparison to Besbes et.al. which we will add in the revision. Here, we highlight two conceptual contributions we make in the paper compared to Besbes et.al.
>
> **1. In expectation bounds given by Besbes et.al versus high-probability bounds in our paper.** Even in the absence of corruptions, we give high-probability regret bounds while the work of Besbes et. al only give regret bound in expectation. This jump from in expectation to high-probability bound is both technically challenging and algorithmically insightful. The insight we make is that for having high-probability bounds, we need to have "clipped SGD". On the other hand, since Besebes et.al. only give bounds holding in expectation, they can get away without having to do clipping. The necessity of clipping in heavy-tailed settings is not an artifact of analysis, but is crucial for good empirical performance, as noted in recent works of [24]. Thus a conceptual contribution we make is that even in the absence of corruptions, a different algorithm compared to Besbes et.al., namely that of clipping gradients is required to get regret bounds holding in high-probability. From a technical perspective, the proofs for high-probability bounds need different techniques as compared to Besbes et.al. For instance, we need several martingale and induction arguments to arrive at high-probability bounds in heavy tails while Besbes et.al. have a much more simpler proofs just based on convexity, since they only give bounds in expectation.
>
> **2. Impact of corruptions which we study, is not secondary and is a fundamental algorithmic challenge!** There is a huge line of work in statistics and algorithms that deals with design of algorithms in the presence of adversarial corruptions. (For example the book of Algorithmic Robust Statistics by Ilias Diakonikolas and Daniel M. Kane and the survey of [35] cited in our manuscript). Given this huge literature and sub-field of learning in the presence of corruptions, our result shows rather surprisingly that  a simple algorithm such as clipped SGD can yield algorithms that are simultaneously robust to corruptions and drifts in the online setting. In light of this, we respectfully push back on the claim of the reviewer that corruptions is only a secondary aspect of the problem.
>
> In light of this, we respectfully dis-agree that *our work is a direct corollary of the work of Besbes et.al.*
>
> **Reiterating the novelty in our work from the paper here in the response:**
>
> **Novelty in the problem setting :** As mentioned in the introduction on Page 1 of our manuscript, online estimation on data streams with high-dimensions, heavy tails and corruptions is a fundamental and important sub-routine in several applications. The conceptual improvement in our problem setting compared to prior work on online estimation is to go beyond the un-corrupted data assumption and consider the impacts of corruptions. The impact of corruptions is both critical to applications (as we show in Page 1), and is also technically challenging (cf. the book of Algorithmic Robust Statistics by Diakonikolas and Kane). However, the impact of corruptions on estimation has mostly been studied in the offline setting. Our setup is the *first* to jointly consider the effects of corruptions and distribution shift simultaneously for online estimation.
>
> **Novelty in the results :** Ours is the first algorithm that is provably robust to outliers and can adapt to distribution shifts in high-dimensional heavy tailed data streams. No other algorithm or analysis can achieve all of these simultaneously. Furthermore, we make conceptual contributions in the paper. For instance, we show in Proposition 2.6 that finite diameter is a necessary criteria to have meaningful performance in the presence of non-stationarities and corruptions. Our work also provides insight that the learning rate should straddle the $O(1)$ known to be optimal in the absence of noise to be adaptive to distribution shift and the $O(1/t)$ known to be optimal in the stationary setting when there is noise.
>
> **Novelty in the analysis :** Providing high-probability finite sample bounds for stationary data streams in the absence of corruptions and drifts are challenging and are only recently being understood (see [24, 34, 43, 52, 57, 60]). All of these papers only present an analysis *in the stationary setting without corruptions*. Our work contributes to this line of work by providing the first analysis in the presence of distribution shifts and outliers that is based on different martingale concentration arguments combined with induction arguments.
>
> **Improvements to the writing:**
>
>  We thank the reviewer for a careful review and identifying issues in presentation. We propose to make these changes in the revision.

---

### Official Review · Reviewer_stYK · 2023-07-19

**Soundness:** 2 fair
**Presentation:** 2 fair
**Contribution:** 2 fair
**Rating:** 4
**Confidence:** 2

**Summary:**

The paper studies the problem of online estimation in a setup that generalizes the stochastic i.i.d. input assumption. The authors consider a setting where the input distribution is allowed to change over time (a certain number of times), and furthermore the input is allowed to be adversarially corrupted (a certain number of times). The authors analyze the standard clipped SGD algorithm to tackle both of these issues simultaneously (Being a simple and implementable algorithm, clipped SGD has many favorable properties in practice). In essence, the paper establishes that the clipped SGD algorithm is "Lipschitz" with respect to distribution drift and contaminations.


**Strengths:**

The paper studies an important problem setting in online convex optimization that gracefully generalizes the standard stochastic i.i.d. input assumption.


**Weaknesses:**

1. At a high level, my reservations with the paper are that the paper proposes a goal in terms of drift and corruption tolerance (on page 4) that seems a bit arbitrary. I was unable to see if even one of these in isolation is understood and what the correct rates are in those settings. In particular, in all of the explicit examples that I could find in the paper, all the lower bounds in terms of $\Delta_T$ (or $\Phi_t$) did not have any multiplicative term with $T$; see, for example, Proposition 2.6 and Section 11.

1. (How the distribution shift is measured) The paper defines the quantity $\Phi_t$ to be the number of drifts in the input sequence. However, all of the results in the paper have regret scaling with $\Phi_T$ times a polynomial in $T$. Is this necessary for algorithms that achieve vanishing regret? What are the best upper bounds and lower bounds for the regret in terms of $\Phi_T$ (without any outliers)? (See the first point for more context)

2. (How the clean error is measured) For outliers, the paper defines the quantity $\Delta_t$ to count the number of outliers in the input sequence. However, the results then depend on $\Delta_T$ multiplied by the diameter of the set and a polynomial in $T$. I am not sure if multiplicative dependence on $T$ is necessary for stochastic inputs  (See the points above for more context). Are there lower bounds?

3. (What counts as dimension free? and Comparison with existing work) The paper lists their results as achieving dimension-independent errors, but they have suboptimal dependence on these quantities in the regret bounds. For example, heavy-tailed mean estimation (where the claimed results are somewhat immediate since they multiplicatively depend on the trace of the covariance matrix). The paper [18] is said to have dimension-dependent errors but I was unable to find the entry corresponding to [18] in Table 1 in the paper [18]. Their result on isotropic covariance matrices naturally uses $d$ samples.



*All references are based on the version uploaded to the supplementary material.*

---
## Recommendation

This is perhaps because I am not from this subfield but I am unable to appreciate the technical results of the paper (more comments below). Thus, I recommend weak reject, and I would be happy to change my score if the authors/other reviews convince me otherwise.


**Questions:**

See above

---

> ### Author Rebuttal · Authors · 2023-08-08
>
> We thank the reviewer for making a thorough read and providing feedback on the paper. Part of the reviewer's questions are also addressed in the table in the attached pdf. Below here, we respond in text with the reviewer’s questions highlighted in ***bolded italics*** with our response below.
>
> ***“At a high level, my reservations with the paper are that the paper proposes a goal in terms of drift and corruption tolerance (on page 4) that seems a bit arbitrary. I was unable to see if even one of these in isolation is understood and what the correct rates are in those settings.“***
>
> Indeed, settings where only either there are distribution shifts or only corruptions is not yet completely understood  as we mention in the conclusion section. We have included a pdf in this rebuttal showing the best known bounds for the various scenarios and what new results our paper contributes. Summarizing from the attached pdf, our paper is the first to give lower and upper bounds in the case when there are corruptions, both in the presence and absence of distribution shift. In the absence of corruptions, prior works have characterized upper and lower bounds only in the absence of distribution shift. In the presence of distribution shifts ours is the first high probability upper bound.  Previous works only gave regret upper bounds *only holding in expectation* for the setting with distribution drift but absence of corruptions. See the table in the attached pdf for specific details.
>
> Proving high-probability results for heavy-tailed data is technically non-trivial. Just providing a high-probability bound heavy-tailed settings *without distribution shifts and corruptions* is itself new and an actively emerging field of literature (see [24, 34, 43, 52, 57, 60]). Our work contributes and extends this line of work by providing the first analysis in the presence of distribution shifts and corruptions, that is based on different martingale concentration arguments combined with induction arguments, which we believe are interesting in their own right.
>
> Practically, the setting with drifts and corruptions are important in applications where several heuristics are proposed (see the Introduction on page 1). Our work is the first to put such estimation tasks on a formal footing to identify upper and lower bounds in the different regimes of presence and absence of drifts and corruptions.
>
> ***“In all of the explicit examples that I could find in the paper, all the lower bounds in terms of $\Phi_T$ (or $\Lambda_T$ ) did not have any multiplicative term with ; see, for example, Proposition 2.6 and Section 11.”***
>
> *Lower bounds for drift:* As we mention in the attached pdf, Besbes et.al.‘s Non-stationary Stochastic Optimization, Operations Research, 2015 and [47], show that $(\Phi_T)^{1/3}T^{2/3}$, is a lower bound for the expected regret in the absence of corruptions. Further, Besbes et.al., show that  this bound can be achieved *in expectation*. However, since we are seeking regret bounds holding in high-probability, our upper bounds have a gap from the lower bound. Concretely, we can only establish upper bounds of the form $T^{l}\Phi_T$​ for some l<1 (Thm 5.1). In the conclusions section of our paper, we list as an open question (second bullet point) of whether there exists an algorithm that can obtain high-probability regret bound of the form $T^{l}\Phi_T^{1-l}$ for some $l\in(0,1)$, which will then close the gap to the lower bound of Besbes et.al.
>
> *Lower bounds for corruption:* Our work is the first to give non-trivial upper and lower bounds on the regret in the presence of corruptions on the online stream (see also attached pdf).  The contribution in our lower bound in Proposition 2.6 and the attached pdf shows that one cannot aim for standard statistical aspirations of infinite diameter, i.e., $\mathcal{D}<\infty$ is needed. However, as the reviewer correctly identifies, our lower bound does not have any dependence on the time-horizon and thus we conjecture it to be loose. This is an artifact of our proof technique where we only consider settings with 0 variance. We believe that more sophisticated arguments with non-zero variance settings, can recover a polynomial in T term in the lower bound. Improving either the lower bound or the high probability upper bound in the case of corruptions is a challenging future work.
>
>
> ***“What counts as dimension-free?”***
> We follow standard terminology that states that a bound is dimension free if it depends on the trace of the co-variance matrix and not on dimension times the maximum eigen-value of the covariance matrix, (c.f. [35, 36] of our attached draft). To be concrete, for mean-estimation of a $d$ dimensional vector, a regret bound that depends on $\text{Trace}(\Sigma)$ is deemed to be “dimension-free”  (c.f. [35, 36] ). On the other hand, a  bound that depends on $d\nu_{max}​(\Sigma)$ where $\nu_{max}(\Sigma)$ is the highest eigen-value of the co-variance matrix is NOT dimension free. A bound on the trace is more favorable in high-dimensional settings (cf [35,36]) since by definition we always have $d\nu_{max}​(\Sigma) \geq \text{Trace}(\Sigma)$. We will add this definition and discussion in the revision stating that our bounds only depend on $\text{Trace}(\Sigma)$ and not on $d\nu_{max}​(\Sigma)$  thereby making our results dimension-free. We clarify the dimension-free definition in the revision
>
> ***Entry corresponding to [18] in Table 1:*** Please look at the third row from the top.

---

> > ### Comment · Reviewer_stYK · 2023-08-17
> >
> > Thank you for the response.
> >
> > **Tightness of bounds**
> > After reading the rebuttal, I am rather surprised by the omission of the paper Besbes-Gur-Zeevi-13, here onwards [BGZ13], in the literature survey; thanks to the reviewer 6VQW for pointing it out. [BGZ13] does help put this paper in context and deserves a prominent discussion in the paper.
> >
> > That being said, the results in the paper are rather unsatisfactory compared to [BGZ13]. Their bounds are always sublinear whenever $\Phi_T$ is sublinear, as opposed to the present paper.  (Similarly one expects sublinear regret whenever the corruption level is sublinear). I understand that their regret bound holds only in expectation but making their bounds high probability should be relatively easy by making clipping (high probability bounds are usually nontrivial in high dimensional settings where one wants a finer control on $\textrm{trace}(\Sigma)$ and $\|\Sigma\|_2$).
> >
> > **First row in the table in rebuttal** How is the first row obtained from the results in Catoni12 and Lugosi-Mendelson-19? Those results are for offline algorithms.
> >
> > **Entry corresponding to [18] in Table 1**
> > My question regarding [18] in Table 1 was how the third row was obtained from the results in [18]. I do not see such result in [18].

---

> > > ### Author Response · Authors · 2023-08-17
> > >
> > > We thank the reviewer for their time and energy and providing very valuable feedback! We clarify the questions raised in the response above where we paraphrase the question/concern in ***bold-italics*** followed by our response.
> > >
> > >
> > > 1. ***Our results are weak since we do not have sub-linear regret whenever drift is sub-linear***: This question is stated as open-problem number 2 in Section 9 of our paper. The best lower bounds from [BGZ13] and [47] hint that it might be possible to find an algorithm to satisfy open problem 2. However, solving the open problem is  technically challenging requiring new ideas and thus outside the scope of the present paper.
> > >
> > > 2. ***High-probability should be relatively easy***: We dis-agree with this claim. Paraphrasing from our first rebuttal in this reply chain —  *proving high-probability results for heavy-tailed data is technically non-trivial. Simple settings without distribution shifts and corruptions is itself an emerging field (see [24, 34, 43, 52, 57, 60]). Our work contributes to this by providing the first analysis in the presence of distribution shifts and corruptions, that is based on different martingale concentration and induction arguments, which we believe are interesting in their own right*.
> > >
> > > 3. ***High probability bounds are only nontrivial in high dimensional settings***: Our results are  *high-dimensional* as we give explicit characterization in terms of trace and largest eigen values of the covariance  matrix. Theorem 5.1 does not assume that a known upper bound on the second moment unlike previous works such as [18,47]. Nevertheless, regret only depends on $\text{Trace}(\Sigma)$ and not on the dimension $d$.
> > >
> > > 4. ***First row of rebuttal***: The offline results state that at any time $t \in \{1,\cdots, T\}$ where there are $t$ samples to estimate the mean, a bound on the instantaneous regret is provided. Summing the instantaneous regret over time $t=1$ through to $t=T$, gives a regret bound for online mean-estimation.
> > >
> > > 5. ***On the results in [18]***: Equation (8) in Theorem 4.2 of [18] can be translated into a regret bound, since it gives a formula for the instantaneous regret at time $n$. Summing that over time yields a cumulative regret bound.  However, we want to point out that Theorem 4.2 in [18] is established under a weaker condition where the time instants of corruption are random and not adversarially chosen (see line 61 of our submission). We will repeat this caveat in table 1 in our revision.

---

### Official Review · Reviewer_ijft · 2023-07-23

**Soundness:** 4 excellent
**Presentation:** 4 excellent
**Contribution:** 4 excellent
**Rating:** 8
**Confidence:** 1

**Summary:**

The paper studys online estimation problems on data stream exhibits challenging properties, including distribution drift, heavy tails, and outlier/anonmalies corruptions. Formally, at each time step, (given all the data that has arrived) the algorithm needs to output an estimation on certain unknown parameter so as to minimize the cumulative regret.

Consider the task of mean estimation as an illustrative example: At each time step, the algorithm receives a data point drawn from an unknown distribution and must estimate the mean of that distribution. The core challenges here are threefold:
1. Distribution drifting: The mean of the distribution from which the data point is drawn can change over time.
2. Heavy tail: The data's distribution might possess unbounded 3rd or higher-order moments.
3. Outliers: Observed data points could be significantly distorted or corrupted.

Interestingly, the paper reveals that a modified version of the clipped Stochastic Gradient Descent (SGD) can attain sublinear regret, even when all three of the aforementioned challenges are present. A particularly notable insight from the research is the necessity of an intermediate learning rate for optimal performance. While an $O(1)$ rate is ideal for addressing distribution drift and an $O(1/t)$ rate is best suited for noise, managing both simultaneously requires a rate of $O(1/T^\alpha)$, where $\alpha$ lies between 0 and 1.

While these findings are derived under certain assumptions regarding the parameter domain and loss functions, the authors further fortify their claims by proving several lower bounds. This demonstrates the indispensability of the stated assumptions.


**Strengths:**

The paper effectively articulates the problem setting and its significance. Even with my limited familiarity with the subject, it's evident that the problem is complex and the results presented are notably substantive.


**Weaknesses:**

N/A

**Questions:**

(The paper under review is outside my area of expertise, and I had not initially opted to review it. Unfortunately, I wasn't presented with alternative submissions for evaluation)

While I struggled to grasp the entirety of the analysis due to unfamiliarity with the methodologies used, I found the distinction between drift and corruption intriguing. It's a bit surprising to me that, it is possible for an algorithm to even distinguish between *drift* and *corruption*, and it required very different rate ($O(1)$ vs $O(1/t)$) to handle them optimally. The two definitions appear to me to be essentially analogous. The paper's main result seem to confirm this intuition, but the analysis still treats the two issues separately. A more detailed discussion from the authors on this particular point would be beneficial.

---

> ### Author Rebuttal · Authors · 2023-08-08
>
> We thank the reviewer for a thorough read and providing feedback on the paper.
>
> **Distinguishing between drift and corruption:** Indeed,  the reviewer’s intuition is spot on. Our lower bound in Proposition 2.6 and in the sketch in the attached pdf is based on the fact that drift and corruption in a precise sense are indistinguishable. Further, the analysis of our algorithm does not impose/prove that our algorithm can distinguish between drift and corruption. Rather the analysis only shows that *for any data-stream* with total distribution shift $\Phi_T$ and corruptions $\Lambda_T$, the regret of clipped-SGD when appropriately tuned, is bounded by an explicit formula of $\Phi_T$ and $\Lambda_T$ as given in Theorem 5.1.
>
> We hope this clarifies the intuition the reviewer is seeking.

---

### Official Review · Reviewer_nCPY · 2023-07-26

**Soundness:** 3 good
**Presentation:** 3 good
**Contribution:** 3 good
**Rating:** 7
**Confidence:** 3

**Summary:**

This paper studies the online robust estimation problem in possibly non-stationary environments. Under the assumption that the loss function is strongly convex, the authors propose an online clipped stochastic gradient descent algorithm with tunable clipping parameter that is able to achieve both adaptiveness in distribution shift and robustness to heavy-tailed inliers and arbitrary corruptions. Moreover, the algorithm does not require distributional knowledge. Theoretical results and experiments are provided.

**Strengths:**

This paper presents an online estimation algorithm that for the first time provably robust to heavy-tails, corruptions and distribution shift simultaneously. The soundness of the paper is supported by their theoretical results and experiments.

**Weaknesses:**

The presentation of this paper lacks some organization and the problem setup part is not clear at the beginning. For example, $X_t$ is said to be the corrupted input, which is the summation of the sample $Z_t$ And the corruption $C_t$, then the authors want to estimate $\theta_t$, which is not reflected in $X_t = Z_t+C_t$ at all. Moreover, some of the contents in the appendix part (simulations on real data) could be moved to the main body.

**Questions:**

N/A

**Limitations:**

Yes

---

> ### Author Rebuttal · Authors · 2023-08-08
>
> We thank the reviewer for a thorough read and providing feedback on the paper.
>
> **Regarding model definition:**  The unknown vector $\theta^{*}_{t}$ is the minimizer of the expected loss function $\mathcal{L}$, where the expectation is with respect to the random vector $Z_t$​. Concretely,  $\theta\_{t}^{\*}$ $ =  \arg\min\_{\theta \in \theta}\mathbb{E}\_{Z_t}[\mathcal{L}(Z_t; \theta)]$. However, the estimaotr cannot directly observe $Z_t$, but can only observe $X_t := Z_t + C_t$. We will make this clarification in the revised version.
>
> Further, the camera ready allows for one extra page which we will use to bring in the key lemmas (Lemma 19.9 and 20.10) in the analysis and details/plots on the real data experiments.

---

> > ### Comment · Reviewer_nCPY · 2023-08-11
> >
> > Thank you for your response. Based on your replies, I maintain my current score.

---

### Author Rebuttal · Authors · 2023-08-08

Here, we address a common questions asked by multiple reviewers -

***"What is the best known upper and lower bounds for the various settings of online estimation in the presence and absence of distribution shifts and corruptions?"***

We answer this in the table attached in the pdf, which we will add to the revised version.

To summarize the pdf, our work gives the first results for both upper and lower bounds in the presence of corruptions, whether or not there is distribution shifts. In the absence of corruptions, a lower bound was given in prior work of Besbes et.al. to handle distribution shifts, while ours is the first *high-probability* regret bound for heavy-tailed data under distribution shifts. Proving high-probability results for heavy-tailed data is technically non-trivial with emerging literature establishing bounds in the *stationary setting without drifts and corruptions*. (see [24, 34, 43, 52, 57, 60]). Our work contributes and extends this line of work by providing the first analysis in the presence of distribution shifts and corruptions, that is based on different martingale concentration arguments combined with induction arguments, which we believe are interesting in their own right.

---

### Decision · Program_Chairs · 2023-09-21

**Decision:**

Accept (poster)

**Comment:**

The authors give the first online estimation algorithm (for strongly convex losses) that is provably robust to heavy-tails, corruptions and distribution shift simultaneously, using clipped SGD with intermediate power-decay learning rate. The technical arguments are interesting. Though scores were split, a majority of reviewers were enthusiastic in recommending acceptance. The main issues raised were the problem formulation and novelty over previous work (stYK, 6VQW) and paper presentation (6VQW). From discussions, other reviewers agree that the problem goal is significant (e.g., obtaining high-probability bounds is technically challenging). I recommend acceptance, and remind the authors to incorporate the suggestions on presentation in the revision.